# MIRRORMARK: A DISTORTION-FREE MULTI-BIT WATERMARK FOR LARGE LANGUAGE MODELS

## ABSTRACT

As large language models (LLMs) become increasingly integral to broad applications such as question answering and content creation, reliable content attribution and accountability have grown increasingly urgent. Watermarking offers a promising approach to identifying AI-generated text. However, existing approaches either provide only a binary provenance signal or perturb the sampling distribution, degrading the text quality; approches that preserve text quality, in turn, often exhibit weak detectability and poor robustness. We propose MirrorMark, a multi-bit and distortion-free watermark for LLMs. By mirroring the sampling randomness in a measure-preserving way, MirrorMark embeds multi-bit messages without altering the token probability distribution during generation, and thus text quality is maintained by design. For robustness, we employ a content-based scheduler that partitions the messages into per-position symbols and allocates tokens to each symbol nearly uniformly, balancing token assignments across positions while maintaining robustness against desynchronization under insertions and deletions. We also present a theoretical analysis that models detection error versus the number of pseudorandom draws per generation step, offering interpretability to our empirical results and insights on the design of high-detectability multi-bit watermarks. In our comparisons with state-of-the-art multi-bit baselines, MirrorMark preserves the text quality comparable with non-watermarked text while delivering superior detectability: with 54 bits embedded in 300 tokens, it improves bit accuracy by 8–12% and correctly identifies up to 11% more watermarked texts when the false positive rate is fixed at 1%. These results show that MirrorMark enables practical attribution, offering a scalable path to provenance and accountability in LLM deployment.

## 1 INTRODUCTION

The rapid proliferation of large language models (LLMs) like ChatGPT (OpenAI (2022)) or the open-sourced LLaMA (Touvron et al. (2023)) and Gemini (Team et al. (2023)) has brought transformative advances to high-quality text generation (Jo (2023)), such as question answering, blog creation, and programming assistance (Austin et al. (2021); Perkins (2023)). Alongside their benefits, however, these models raise growing concerns over authenticity, ownership, and responsible use of generated content. In particular, as synthetic outputs become increasingly indistinguishable from human-produced material, content attribution has emerged as a crucial safeguard for mitigating misinformation, enforcing intellectual property rights, and enabling accountability in AI deployment.

Watermarking has become one of the most promising strategies for this purpose, embedding imperceptible signals into generated outputs that can later be detected to verify provenance. Existing papers suggest incorporating invisible watermarks into text to detect AI-generated content (Kirchenbauer et al. (2023); Aaronson & Kirchner (2022); Christ et al. (2024); Kuditipudi et al. (2024); Dathathri et al. (2024); Hu et al. (2024); Wu et al. (2024)). These schemes are designed only to answer a binary question: whether or not a piece of content is watermarked. Early research on watermarking for LLMs was pioneered by Kirchenbauer et al. (2023), who introduced a permutation-based reweighting strategy. Seeding a hash function with the previous context tokens, the vocabulary is partitioned into red and green lists. During generation, a small bias is added to the logits of tokens in the green list, increasing their likelihood of selection. Detection requires only knowledge of the hash function and can be performed via hypothesis testing, without model or API access. Hu et al.

(2024) and Wu et al. (2024) advanced this line of work by proposing unbiased (or stealthy) reweighting strategies. Although the proposed reweighting function introduces distortion at each generation step, it maintains the expected token distribution to ensure good text quality. A different paradigm, explored by Aaronson & Kirchner (2022); Christ et al. (2024); Kuditipudi et al. (2024); Dathathri et al. (2024), embeds the watermark during sampling stage to guarantee the probability distribution remains unchanged.

While such approaches are computationally efficient and have demonstrated robustness under benign transformations, they are inherently limited in expressiveness. They cannot encode meta information such as the model identity, generation time, or usage context, all of which could be valuable for auditing and forensic analysis. These limitations have motivated growing interest in multi-bit watermarking where the embedded signal conveys a payload of information rather than a binary message. A multi-bit watermark can encode a model identifier, generation timestamp, or application-specific metadata, significantly enhancing the utility of watermarking for real-world deployment. By enabling richer attribution, multi-bit schemes are able to support granular accountability. Similar to zero-bit watermarking, existing multi-bit watermarking schemes can be categorized into two groups: (i) watermarking with distortion (Wang et al. (2024); Fernandez et al. (2023); Yoo et al. (2024); Qu et al. (2024); Jiang et al. (2025)) and (ii) distortion-free watermarking (Zamir (2024); Kordi Boroujeny et al. (2024)). Building on the zero-bit watermarking framework of Kirchenbauer et al. (2023), subsequent methods (Wang et al. (2024); Fernandez et al. (2023); Yoo et al. (2024); Qu et al. (2024)) bias the sampling probability of selected tokens, deliberately shifting the model's output distribution away from its native distribution to embed the signal. This shift can degrade naturalness and readability, thereby diminishing the utility of the LLM for end users. Inspired by Wu et al. (2024) and Hu et al. (2024), Jiang et al. (2025) propose a multibit reweighting strategy to embed information during generation while maintaining unbiased distribution. However, the bit accuracy is still limited. Moreover, existing distortion-free schemes (Kordi Boroujeny et al. (2024); Zamir (2024)) design novel sampling strategies and score functions to embed multi-bit watermarks while preserving the original distribution of the LLM. However, since they build on Christ et al. (2024), where the resilience of the watermarking scheme remains an open issue, and focus solely on enabling multi-bit embedding, they do not incorporate any design for robustness.

In this work, we introduce MirrorMark, a framework that extends Aaronson & Kirchner (2022) and Dathathri et al. (2024) to enable multi-bit and distortion-free watermarking in LLM responses. MirrorMark preserves text quality while substantially improving detectability compared to state-of-the-art (SOTA) methods. To enhance robustness against editing attacks, we propose the Content-Anchored Balanced Scheduler (CABS), which anchors watermark scheduling to content-hashed frames and employs balanced token assignment within each frame. This design ensures near-uniform token allocation per bit and mitigates the effects of token insertions and deletions. In addition, we develop a theoretical framework that characterizes the relationship between equal error rate (EER) and the number of pseudorandom function (#PRF) draws per generation step, which offers interpretability to our empirical results and guides the design of high-detectability multi-bit watermarks. Besides, we conduct a comprehensive empirical study comparing MirrorMark with MPAC (Yoo et al. (2024)), RSBH (Qu et al. (2024)), and StealthInk (Jiang et al. (2025)). Our results show that with 36 bits embedded in 300 tokens, the multi-bit extension of Aaronson & Kirchner (2022) achieves a true positive rate at 1% false positive rate (TPR@1%FPR) of 99.8% and a bit accuracy of 98.19%, while maintaining text quality comparable to non-watermarked text. Similarly, the multi-bit extension of Dathathri et al. (2024) achieves a TPR@1%FPR of 99.6% and a bit accuracy of 96.14%, again with text quality on par with non-watermarked text. Notably, for the two extensions, these empirical findings are consistent with our theoretical analysis of the EER as a function of #PRF draws.

## 2 WARM UP: AARONSON & KIRCHNER (2022) AND DATHATHRI ET AL. (2024)[1]

The zero-bit watermarking methods that can be extended to multi-bit using MirrorMARK are those that generate the next token by sampling a random value. In this paper, we select two representa-

---

[1]In our paper, uppercase characters such as $G$, $U$ denote the random variable, while lowercase character such as $u$ denotes the realization of the random variables, and bold character such as $\boldsymbol{u}$ denotes vectors.

tives (i.e., Aaronson & Kirchner (2022) and Dathathri et al. (2024)) to apply MirrorMARK. In this section, we introduce their basic ideas, where the core idea is to select the next token using pseudorandom values, thereby embedding a statistical signal without altering the underlying probability distribution. Let $p(x_1), \ldots, p(x_V)$ denote the probability distribution over the $V$-tokens vocabulary at generation step $t$, given by the LLM as $p_{\text{LM}}(\cdot \mid x_{<t})$.

## 2.1 GUMBEL SAMPLING (AARONSON & KIRCHNER (2022))

The classical Gumbel trick (Gumbel (1954)) samples from this distribution by adding i.i.d. $\text{Gumbel}(0, 1)$ random variables $G_1, \ldots, G_V$ to the log-probabilities:

$$x^* := \arg \max_{1 \leq i \leq V} \left[ \log p(x_i) + G_i \right], \tag{1}$$

which guarantees that $\Pr(x^* = x_i) = p(x_i)$ for all $i$.

Since a $\text{Gumbel}(0, 1)$ random variable can be expressed as $-\log(-\log U)$ for $U \sim \text{Uniform}(0, 1)$, equation 1 is equivalent to drawing $U_1, \ldots, U_V \overset{\text{i.i.d.}}{\sim} \text{Uniform}(0, 1)$ and selecting

$$x^* = \arg \max_{1 \leq i \leq V} \left[ \log p(x_i) - \log(-\log U_i) \right] = \arg \max_{1 \leq i \leq V} U_i^{1/p(x_i)}. \tag{2}$$

To embed the watermark by Gumbel sampling, at step $t$, Aaronson & Kirchner (2022) use watermark key and the context tokens as the seed $r_t$ and set $u_i = g(x_i, r_t)$ for token $x_i$ where $g(\cdot, r_t)$ is a pseudorandom function (PRF) with range $\text{Uniform}(0, 1)$. This construction ensures that the watermark is embedded in the sampled token and can later be detected by reproducing these pseudorandom draws and designing an appropriate score function. Besides, due to the property of gumbel trick, the sampling process is distortion-free. However, as equation 2 shows, the token with the largest $U^{1/p(x)}$ is always selected. Therefore, the generated response is deterministic for the same prompt.

## 2.2 TOURNAMENT SAMPLING (DATHATHRI ET AL. (2024)

In tournament sampling which proceeds in $L$ layers, similarly, at layer $\ell$, a PRF $g^\ell(\cdot, r_t) : \mathcal{V} \to [0, 1]$ assigns each token a value $u^\ell$ using a seed $r_t$ from the watermark key and context tokens. Before the tournament starts, $n_0$ candidate tokens $\{c_1, \ldots, c_{n_0}\}$ are sampled from original probability distribution $p_{\text{LM}}(\cdot \mid x_{<t})$. In particular, with $L$ layers, $n_0 = 2^L$. For the first layer $\ell = 1$, the $n_0$ candidates are randomly paired. For each subsequent layer $\ell = 2, \ldots, L$, the $n_{\ell-1}$ surviving candidates are paired according to the tournament structure. In each match, the token with larger $g$-value wins. The winners form the candidate set for the next layer. After $L$ layers, the remaining single token $x_t$ is emitted as the output token. Compared to Aaronson & Kirchner (2022) which deterministically samples the token, the method in Dathathri et al. (2024) is a probabilistic scheme. Therefore, the responses generated by Dathathri et al. (2024) will show more diversity.

## 2.3 DETECTION

Given a text $x_1, \ldots, x_T$, in Aaronson & Kirchner (2022), the detector recomputes $u_t = g(x_t, r_t)$ for $t = 1, \ldots, T$. If the text is unwatermarked, the $u_t$ values follow $\text{Uniform}(0, 1)$ i.i.d. If watermarked, they are skewed toward larger values. Aaronson & Kirchner (2022) propose the following statistic to accumulate evidence for large $u_t$ values

$$\text{LogScore}(x) = -\sum_{t=1}^{T} \log(1 - u_t). \tag{3}$$

In Dathathri et al. (2024), let $u_{t,\ell} := g^\ell(x_t, r_t)$ and $\alpha_\ell$ be the weight of the $\ell$-th layer. They propose the weighted mean score as follows

$$\text{WeightedMeanScore} = \frac{1}{T} \sum_{t=1}^{T} \frac{1}{L} \sum_{\ell=1}^{L} \alpha_\ell u_{t,\ell}. \tag{4}$$

Furthermore, by accounting for the multi-layer structure, they leverage a Bayesian score that aggregates evidence across tokens and layers, i.e.,

$$\text{BayesianScore}(x) = P(w \mid \boldsymbol{u}) = \sigma \left( \log \frac{P(w \mid \boldsymbol{u})}{P(\neg w \mid \boldsymbol{u})} \right) = \sigma \left( \log \frac{P(\boldsymbol{u} \mid w)}{P(\boldsymbol{u} \mid \neg w)} + \log \frac{P(w)}{1 - P(w)} \right), \tag{5}$$

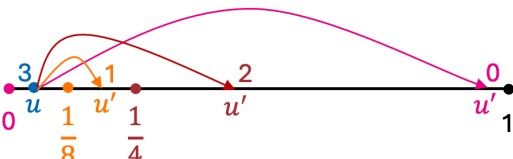

Figure 1: Overview of mod-1 mirroring. In this example, $m$=2, and $M \in \{0, 1, 2, 3\}$. blue $u$ is the original $u$ value and will be mirrored to $u' = \Psi(u; \psi_M)$ according to embedded message. To embed $M$=0, the pink $u'$ is obtained by mirroring $u$ against 0. To embed $M$=1, the orange $u'$ is obtained by mirroring $u$ against $\frac{1}{8}$. To embed $M$=2, the red $u'$ is obtained by mirroring $u$ against $\frac{1}{4}$. The message $M$=3 is the null symbol, and the original blue $u$ is applied to embed $M$=3.

where $P(w)$ and $P(\neg w)$ are respetively watermarked prior and nonwatermarked prior. $P(w \mid \boldsymbol{u})$ is the watermarked posterior, while $P(\boldsymbol{u} \mid w)$ and $P(\boldsymbol{u} \mid \neg w)$ are the likelihood for the watermarked and non-watermarked hypotheses, respectively. $\sigma(\cdot)$ is the logistic sigmoid. Since $u_{t,\ell}$ is a value generated by a PRF following $\text{Uniform}(0, 1)$, referring to equation A7 in Dathathri et al. (2024),

$$P(\boldsymbol{u} \mid \neg w) = \prod_{t=1}^{T} \prod_{\ell=1}^{L} P(u_{t,\ell} \mid \neg w) = \prod_{t=1}^{T} \prod_{\ell=1}^{L} 1 = 1, \tag{6}$$

Referring to equations A8, A9, and A10 in Dathathri et al. (2024),

$$P(\boldsymbol{u} \mid w) = \prod_{t=1}^{T} \prod_{\ell=1}^{L} P(u_{t,\ell} \mid w, u_{t,<\ell}) = \prod_{t=1}^{T} \prod_{\ell=1}^{L} \sum_{c=1}^{2} P(u_{t,\ell} \mid \pi_{t,\ell} = c) P(\pi_{t,\ell} = c \mid w, u_{t,<\ell}), \tag{7}$$

where $\pi_{t,\ell}$ denotes the number of distinct $u$ values in the pairwise tournament at layer $\ell$ for $t$-th token, and hence $\pi_{t,\ell} \in \{1, 2\}$. Basically, they derived $P(u_{t,\ell} \mid \pi_{t,\ell} = c)$ as in equation A9, the distribution of watermarked $u_{t,\ell}$ given $\pi_{t,\ell}$. Since the number of unique $u$ values is governed by the layer entropy and can be predicted from the preceding $u$ values, i.e., higher-entropy time steps typically produce larger $u$ values. Therefore, they used a logistic regression model to predict $P(\pi_{t,\ell} = c \mid w, u_{t,<\ell})$.

## 3 MirrorMark

In this paper, we propose a multi-bit and distortion-free watermarking framework, MirrorMark, which combines three complementary components to embed and recover multi-bit messages without altering the output distribution of LLMs. First, a mod-1 mirroring transformation encodes an $m$-bit symbol by reflecting each $u$ value around a message-specific pivot. Next, the Content-Anchored Balanced Scheduler (CABS) determines which symbol is embedded at each generation step by mapping tokens to message positions in a balanced and context-dependent manner. Finally, during decoding, CABS is replayed to recover token-to-position assignments, each symbol is decoded from the mirrored $u$ values using the appropriate score function, and all decoded values over the tokens are aggregated to detect the watermark.

### 3.1 Mod-1 Mirroring

To extend Aaronson & Kirchner (2022) and Dathathri et al. (2024) to multi-bit watermarking, we propose a mod-1 mirroring process and denote an $m$-bit watermark message as $M \in \mathbb{M} = \{0, 1, \dots, 2^m - 1\}$. Let $U \sim \text{Uniform}(0, 1)$ and let $u \in [0, 1)$ denote a realization of $U$, corresponding to the sampling procedures in Sections 2.1 and 2.2. For a message $M \in \{0, 1, \dots, 2^m - 2\}$, we define the mirroring point $\psi_M = \frac{M}{2^{m+1}}$ and reflect $u$ about $\psi_M$ as follows:

$$\Psi(u; \psi_M) = (2\psi_M - u) \bmod 1. \tag{8}$$

The overview of mod-1 mirroring is shown as Fig. 1. We reserve $M = 2^m - 1$ as a null symbol, which applies no mirroring, i.e., $\Psi(u; \psi_{\text{null}}) = u$, providing the multi-bit formulation with the flexibility to fall back to the original zero-bit scheme. The map $u \mapsto (2\psi - u) \bmod 1$ is a measure-preserving involution on $[0, 1)$ because it is a bijection that preserves local lengths and

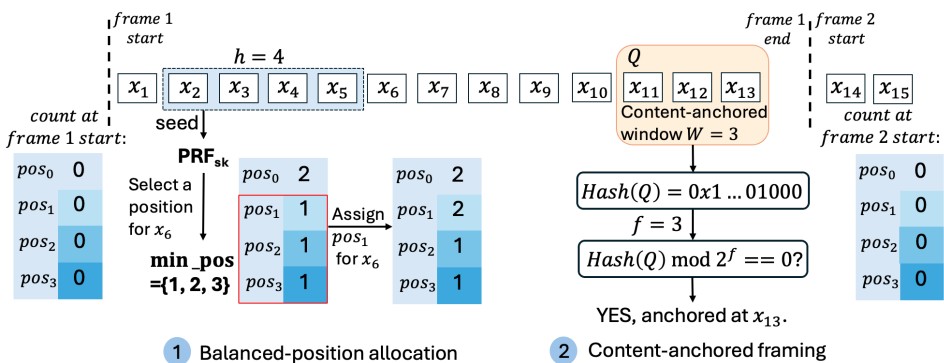

Figure 2: Overview of CABS, where the number of positions $H=4$

simply permutes symmetric pairs $\{(\psi_M - |u - \psi_M|) \bmod 1, (\psi_M + |u - \psi_M|) \bmod 1\}$. Hence, if $U \sim \text{Uniform}(0,1)$, then $\Psi(U; \psi) \sim \text{Uniform}(0,1)$. To embed the watermark, the encoder uses the mirrored $u$ corresponding to the chosen message $M$ to draw the next token.

## 3.2 CONTENT-ANCHORED BALANCED SCHEDULER (CABS)

To embed more bits in limited tokens, similar to Yoo et al. (2024); Jiang et al. (2025), we construct a message sequence MsgSeq with $H$ positions[2], where each position is intended to carry an $m$-bit symbol, and in this way we embed a payload of $b = m \cdot H$ bits. In their methods, during generation, an $m$-bit symbol to be embedded is assigned to the next token by pseudorandomly selecting a position based on the watermark key and the context tokens. However, such pseudorandom allocation does not guarantee uniform token distribution across positions when the number of tokens is limited. For instance, some positions may not receive any tokens at all, in which case the embedded message must be guessed at random during decoding. An intuitive alternative is to preferentially assign tokens to positions that are underrepresented. Yet, this strategy is fragile: even a few token insertions or deletions can desynchronize the assigned positions from those used at generation, thereby destroying the watermark.

To address these challenges, we propose a context-anchored balanced scheduler (CABS), which aims to balance token assignments across positions while maintaining robustness against desynchronization. We present CABS as in Fig. 2 and formally in Algorithm 3 in Appendix D. Specifically, CABS has two components: balanced-position allocation and context-anchored framing. To achieve balanced-position allcation, CABS maps token to position based on context $h$ tokens, ensuring that each position receives a sufficient number of tokens for reliable decoding, while avoiding reliance on fragile sequential assignments. Importantly, assignments are performed within frames, and each frame boundary is determined by a content-anchored window $Q$ of $W$ tokens. A new frame is anchored whenever the $f$ least significant bits of $\text{Hash}(Q)$ are all zero. At the start of each frame, the token counts for all positions are reset, and the allocation restarts from a synchronized state. This framing mechanism *prevents error propagation across the entire sequence and confines the impact of local insertions or deletions to the affected frame*. Besides, the parameters min_len and max_len shown in Algorithm 3 restrict the frame length between successive anchors, so that no frame is too short which avoids instability, or too long which prevents unbounded propagation of insertions or deletions. As a result, CABS not only reduces the risk of empty or highly imbalanced allocations but also improves resilience to editing operations such as insertion, deletion, and substitution. Furthermore, we show the encoding process as Algorithm 1.

## 3.3 DECODING AND DETECTION

Since we leverage CABS to allocate each token to a certain position of MsgSeq, each token carries the symbol at that position. We first decode the symbol at each position, and then perform the detection based on decoded MsgSeq.

---

[2]MsgSeq $\in \{0, \ldots, 2^m - 1\}^H$, which means each position of MsgSeq carries an $m$-bit symbol.

---

**Algorithm 1** CABS-based Encoder

---

**Input:** CABS parameters $\big(\mathsf{Elig}(\cdot), \mathsf{sk}, H, W, f, h, \mathtt{min\_len}, \mathtt{max\_len}\big)$, prompt $\boldsymbol{a}$, length $T$, message sequence with $H$ positions $\mathtt{MsgSeq} \in \{0, \dots, 2^m - 1\}^H$, original distribution $p_{LM}$, watermarked distribution $p_{wm}$

**Output:** Generated sequence $\boldsymbol{x}_{0:T-1}$

1: $\mathsf{cabs} \leftarrow \mathsf{CABS}\big(\mathsf{Elig}(\cdot), \mathsf{sk}, H, W, f, h, \mathtt{min\_len}, \mathtt{max\_len}\big)$
2: **for** $t = 0$ to $T - 1$ **do**
3:    **if** $t < h$ **then**
4:       Sample $x_t \sim p_{LM}\big(\cdot \mid \boldsymbol{a}, \boldsymbol{x}_{:t-1}\big)$
5:    **else**
6:       $pos \leftarrow \mathsf{cabs}(\boldsymbol{x}_{:t-1})$
7:       Sample $x_t \sim p_{wm}\big(\cdot \mid \boldsymbol{a}, \boldsymbol{x}_{:t-1}, \mathtt{MsgSeq}[pos]\big)$
8:    **end if**
9: **end for**

---

### 3.3.1 GUMBEL SAMPLING-BASED MIRRORMARK

For the Gumbel-based construction, we decode the symbol at each position by computing the LogScore for each $M \in \mathbb{M}$ and selecting the one with the maximum score. The symbol at a certain position with assigned $K$ tokens is

$$\hat{M} = \arg \max_{M \in \mathbb{M}} -\sum_{i=1}^{K} \log\big(1 - \Psi(u_i, \psi_M)\big). \tag{9}$$

### 3.3.2 TOURNAMENT SAMPLING-BASED MIRRORMARK

For the tournament-based construction, we can employ either the WeightedMeanScore or Bayesian decoder to decode the symbol at each position. By comparing the WeightedMeanScore for each $M \in \mathbb{M}$, we can decode the symbol with the maximum score at a certain position allocated with $K$ tokens,

$$\hat{M} = \arg \max_{M \in \mathbb{M}} \frac{1}{K} \sum_{t=1}^{K} \frac{1}{L} \sum_{\ell=1}^{L} \alpha_\ell \Psi(u_{t,\ell}, \psi_M). \tag{10}$$

For Bayesian decoder we train a decoder that evaluates the posterior

$$P(M \mid U, w) \propto P(M) \, P(U \mid M, w), \tag{11}$$

where $P(M)$ is the prior and $P(U \mid M, w)$ is the likelihood of the observed group $U = \{u_{i,j}\}_{i=1..K, j=1..L}$ at a position assigned with $K$ tokens. Similar to equation 7, the likelihood factorizes as

$$P(U \mid M, w) = \prod_{i=1}^{K} \prod_{j=1}^{L} \sum_{c=1}^{2} P\big(\Psi(u_{i,j}, \psi_M) \mid \pi_{i,j} = c\big) P\big(\pi_{i,j} = c \mid w, \Psi(u_{i,<j}, \psi_M)\big). \tag{12}$$

Then the Bayesian decision rule selects the symbol with the largest score

$$\hat{M} = \arg \max_{M \in \mathbb{M}} \left\{ \log P(M) + \prod_{i=1}^{K} \prod_{j=1}^{L} \sum_{c=1}^{2} P\big(\Psi(u_{i,j}, \psi_M) \mid \pi_{i,j} = c\big) P\big(\pi_{i,j} = c \mid w, \Psi(u_{i,<j}, \psi_M)\big) \right\}. \tag{13}$$

### 3.3.3 DETECTION PROCEDURE

After decoding, the $u$ value of each token is mirrored against the recovered symbol. For the symbols docoded by equation 9, equation 10, and equation 13, respectively, the mirrored values are then aggregated into a global score using LogScore (equation 3), WeightedMeanScore (equation 4), and BayesianScore (equation 5), repectively. The text is declared watermarked if this score exceeds a predefined threshold. Algorithm 2 summarizes the overall procedure: it first simulates the CABS scheduler to assign tokens to positions, applies the chosen decoder at each position, and finally performs global detection.

---

**Algorithm 2** CABS-based Decoding & Detection

---

**Input:** Sequence $\boldsymbol{x}_{0:T-1}$, secret key sk, message length $H$, context length $h$, CABS params $\big(\mathsf{Elig}(\cdot), W, f, \mathtt{min\_len}, \mathtt{max\_len}\big)$, decoder choice $\mathsf{DEC} \in \{\mathsf{gumbel}, \mathsf{wmean}, \mathsf{bayes}\}$, scorer choice $\mathsf{SCORER} \in \{\mathsf{gumbel}, \mathsf{wmean}, \mathsf{bayes}\}$, threshold thres

**Output:** Message sequence $\mathsf{MsgSeq} \in \{0, \ldots, 2^m - 1\}^H$ and a decision $\in \{\mathsf{true}, \mathsf{false}\}$ on whether $\boldsymbol{x}_{0:T-1}$ is watermarked

1: $\mathsf{cabs} \leftarrow \mathsf{CABS}\big(\mathsf{Elig}(\cdot), \mathsf{sk}, H, W, f, h, \mathtt{min\_len}, \mathtt{max\_len}\big)$
2: Initialize $\mathcal{U} \leftarrow \{\, pos : [\,] \mid pos = 1, \ldots, H \,\}, \quad \mathcal{U}_{\mathrm{mirror}} \leftarrow [\,]$
3: **for** $t = h, \ldots, T - 1$ **do**
4:     $pos \leftarrow \mathsf{cabs}(\boldsymbol{x}_{:t})$
5:     Generate random value $u_t$ seeding sk and $\boldsymbol{x}_{t-h:t}$
6:     $\mathcal{U}[pos].\mathsf{append}(u_t)$
7: **end for**
8: **for** $pos = 1, \ldots, H$ **do**
9:     $\mathsf{MsgSeq}[pos] \leftarrow \mathsf{SymbolDecoder}\big(\mathcal{U}[pos]; \mathsf{DEC}\big)$ %% If $\mathsf{DEC} = \mathsf{gumbel}$ use equation 9; if $\mathsf{DEC} = \mathsf{wmean}$ use equation 10; if $\mathsf{DEC} = \mathsf{bayes}$ use equation 13.
10:     **for** each $u \in \mathcal{U}[pos]$ **do**
11:         $u_{\mathrm{mir}} \leftarrow \Psi\big(u, \psi_{\mathsf{MsgSeq}[pos]}\big)\big)$
12:         $\mathcal{U}_{\mathrm{mirror}}.\mathsf{append}(u_{\mathrm{mir}})$
13:     **end for**
14: **end for**
15: $score \leftarrow \mathsf{Score}\big(\mathcal{U}_{\mathrm{mirror}}; \mathsf{SCORER}\big)$ %% If $\mathsf{SCORER} = \mathsf{gumbel}$ use equation 3; if $\mathsf{SCORER} = \mathsf{wmean}$ use equation 4; if $\mathsf{SCORER} = \mathsf{bayes}$ use equation 5.
16:
17: **return true** if $score > \mathtt{thres}$, else **false**

---

# 4 THEORETICAL EER FOR MIRRORMARK

In this section, we analyze the theoretical EER in the single-position setting ($H = 1$). MirrorMark uses multiple positions in practice, and the results reported in Section 5 correspond to the $H > 1$ setting. For completeness, Appendix F.5 presents the performance of MirrorMark under varying $m$ with $H = 1$, and the observed trends are fully consistent with the theoretical behavior analyzed in this section.

**Theorem 4.1** *Consider sequence-level detection over $T$ approximately independent tokens for (i) multi-bit watermarking based on Gumbel-max sampling and (ii) multi-bit watermarking based on tournament sampling. Let $\#\mathrm{PRF}$ denote the number of PRF draws per encoding step, we derive the theoretical equal error rate (EER) as follows. See proof in Appendix C.*

*(i) Gumbel-max sampling. In Gumbel-max, $\#\mathrm{PRF} = V$ where $V$ is the vocabulary size. Then,*
$$\log \mathrm{EER}_{\mathrm{Gumbel}} \sim -\Theta\big(T\,(\ln \#\mathrm{PRF})^2 - m\big), \tag{14}$$

*(ii) Tournament sampling. In tournament sampling, a full $L$-layer tournament costs $\#\mathrm{PRF} = 2^{L+1} - 2$ and $L = \log_2\big(\frac{\#\mathrm{PRF}}{2} + 1\big)$. Then,*
$$\log \mathrm{EER}_{tour} \sim -\Theta\big(\Gamma(c)^2 T \log \#\mathrm{PRF}\big). \tag{15}$$

*where $c \in [0, 1)$ represents the collision probability level, which increases with larger $C_{wm}^{\ell}$. $C_{wm}^{\ell}$ is defined in Definition 22 in Dathathri et al. (2024) and represents the collision probability at layer $\ell$, which is the probability that two samples drawn i.i.d. from the probability distribution of tokens at layer $l$ are the same. $\Gamma(c) \in (0, 0.694]$ is a decreasing function of $c$ whose explicit form is given in equation 66 in Appendix C.*

Fig. 3 theoretically confirms the EER varying with parameters stated in Theorem 4.1. The Gumbel-max curves follow the predicted quadratic decay in $\log V$ (equation 14), yielding excellent EER even at moderate $T$. Tournament sampling matches the linear decay predicted in equation 15, where the slope is proportional to $\Gamma(c)^2$. As the tournament depth increases from 20 to 30 layers, $\#\mathrm{PRF}$ grows, but the Uniform distribution leads to higher collision levels $c$ in deeper layers and reduces $\Gamma(c)$ as we analyzed in Appendix F.5. Therefore, the 20-layer curve outperforms the 30-layer one despite having fewer PRF draws.

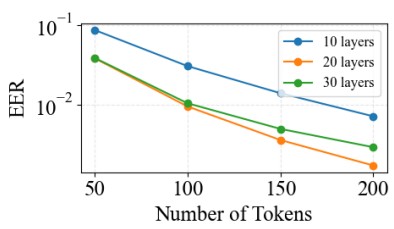

(a) Tournament sampling based
MirrorMark, $m$=1

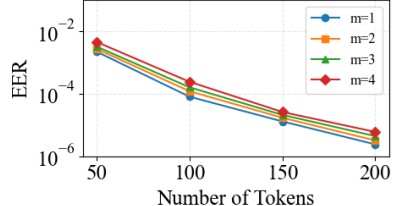

(b) Gumbel-max based MirrorMark

Figure 3: Comparison of the theoretical EER for MirrorMark.

# 5 EVALUATIONS

In this section, we compare MirrorMark with SOTA approches: MPAC (Yoo et al. (2024)), RSBH (Qu et al. (2024)), and StealthInk (Jiang et al. (2025)) from the aspects of detectability, text quality, and robustness. Basically, these three baselines are distortionary watermarking approaches. Specifically, StealthInk proposes an unbiased multi-bit watermarking, which maintains the output distribution in expectation. For MirrorMark, we compare the methods with three score functions, denoted as Gumbel-max (equation 3), Tour-Wmean (equation 4), and Tour-Bayes (equation 5). By default, we set the symbol bit length to $m = 3$ at each position for MirrorMark, unless stated otherwise. We use LLAMA2-7B (Touvron et al. (2023)) and 500 randomly selected texts from the RealNewsLike subset of C4 (Raffel et al. (2020)) as prompts. In particular, we use AUC and TPR@FPR=1% to evaluate the detection performance, where TPR@FPR=1% represents the true positive rate at a fixed false positive rate of 1%. We use the bit accuracy, the fraction of bits that are correctly decoded, to evaluate the decoding performance. Besides, we compare the perplexity, GPT4o[3] judge score, and repetition score of these approaches to evaluate their text quality. Please refer to Appendix D for detailed experimental setup.

## 5.1 TRADE-OFF BETWEEN TEXT QUALITY AND DETECTABILITY

Table 1: Mean perplexity and detectability for different approaches on 300 tokens. Each perplexity is given with a 90% confidence interval based on bootstrapping.

| Method | 36 Bits | | | | 54 Bits | | | |
|---|---|---|---|---|---|---|---|---|
| | AUC | TPR@1%FPR | Bit Acc. | Perplexity | AUC | TPR@1%FPR | Bit Acc. | Perplexity |
| Non Watermark | – | – | – | 7.2784 [7.1294, 7.4296] | – | – | – | 7.2784 [7.1294, 7.4296] |
| MPAC | 0.9949 | 0.9800 | 0.9347 | 9.1951 [9.0404, 9.3516] | 0.9962 | 0.9840 | 0.8928 | 9.3457 [9.1704, 9.5224] |
| RSBH | **0.9998** | **0.9980** | **1.0000** | 32.8955 [31.4973, 34.3369] | 0.9989 | 0.9980 | **0.9928** | 32.8184 [31.3574, 34.3446] |
| StealthInk | 0.9892 | 0.8520 | 0.8896 | 7.8241 [7.6260, 8.0223] | 0.9890 | 0.8900 | 0.8415 | 7.8950 [7.6933, 8.0974] |
| Gumbel-max | **0.9998** | **0.9980** | 0.9819 | **7.0486** [6.8991, 7.1997] | 0.9991 | 0.9960 | 0.9701 | **7.1751** [7.0195, 7.3383] |
| Tour-Wmean | 0.9994 | 0.9860 | 0.9518 | 7.3706 [7.2265, 7.5202] | **1.0** | **1.0** | 0.9110 | 7.3295 [7.1828, 7.4792] |
| Tour-Bayes | 0.9992 | 0.9960 | 0.9614 | 7.3706 [7.2265, 7.5202] | **1.0** | 0.9960 | 0.9276 | 7.3295 [7.1828, 7.4792] |

We first evaluate all approaches under moderate payload sizes ($b \in 36, 54$) with 300 generated tokens. The results, reported in Table 1, capture both detectability and text quality. Detectability is assessed using AUC, TPR@FPR=1%, and bit accuracy, while text quality is measured by perplexity. For completeness, we also provide results on shorter sequences of 200 tokens (Table 4) and longer sequences of 400 tokens (Table 5) in Appendix F.

---

[3] https://openai.com/index/hello-gpt-4o/

From these experiments, we observe that baseline methods exhibit a sharp trade-off between text quality and watermark detectability. Specifically, the baselines either suffer from significant perplexity degradation, making the text less natural, or show weakened detection power, confirming their limited applicability in realistic settings. By contrast, MirrorMark (including the Gumbel-max, Tour-Wmean, and Tour-Bayes variants) consistently achieves strong detection performance while maintaining perplexity at levels comparable to non-watermarked text, even for only 200 tokens, as demonstrated in Table 4. Besides, we evaluate the GPT4o judge score and repetition rate across these approaches as in Appendix F.5, which demonstrates the superior text quality of MirrorMark.

Given that MirrorMark maintains a favorable trade-off in these settings, we further stress-test it under larger payload sizes ($b \in \{72, 90\}$). The results in Figure 4 demonstrate that MirrorMark continues to provide competitive detectability while preserving text quality, highlighting its scalability beyond what baseline approaches can achieve. We show the comparison on AUC in Figure 7 in Appendix F. Overall, the Gumbel-max multi-bit watermarking achieves a TPR@1%FPR comparable to the tournament-sampling baseline, while yielding higher bit accuracy than both Tour-Bayes and Tour-Wmean. Tour-Bayes and Tour-Wmean achieve similar true positive rates, with Tour-Bayes offering superior bit accuracy. Besides, in Appendix F.5, we show that with single position $H = 1$, Gumbel-max is better than tournament sampling based MirrorMark, which is consistent with Theorem 4.1. Specifically, using LLaMA-2-7B with a vocabulary size of 32,000, we obtain $\log \text{EER}_{\text{Gumbel}} \sim -\Theta(107.6T)$. In contrast, setting $L = 30$ for tournament sampling gives $\log \text{EER}_{\text{tour}} \sim -\Theta(20.79T)$.

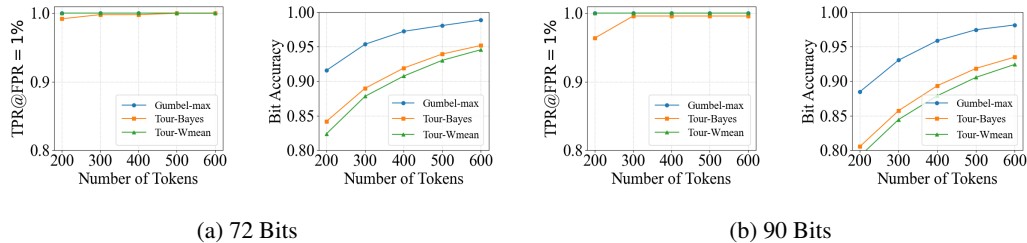

(a) 72 Bits          (b) 90 Bits

Figure 4: Detectability of MirrorMark across varying number of tokens repectively with 72 and 90 bits embedded.

## 5.2 ROBUSTNESS

Table 2: Detectability for different approaches on 400 tokens with 36 bits embedded after copy-paste attack, where the edit fraction $\epsilon \in \{0, 0.2, 0.4\}$.

| Method | $\epsilon = 0$ (No attack) | | | $\epsilon = 0.2$ | | | $\epsilon = 0.4$ | | |
|---|---|---|---|---|---|---|---|---|---|
| | AUC | TPR@1%FPR | Bit Acc. | AUC | TPR@1%FPR | Bit Acc. | AUC | TPR@1%FPR | Bit Acc. |
| MPAC | 0.9970 | 0.9820 | 0.9599 | 0.9753 | 0.8975 | 0.8997 | 0.9593 | 0.7675 | 0.8397 |
| RSBH | 0.9999 | **1.0** | **1.0** | 0.9697 | 0.0850 | 0.6138 | 0.8455 | 0.0050 | 0.6038 |
| StealthInk | 0.9941 | 0.9500 | 0.9204 | 0.9705 | 0.8175 | 0.8448 | 0.9172 | 0.4750 | 0.7716 |
| Tour-Wmean | 0.9997 | **1.0** | 0.9681 | 0.9981 | 0.9900 | 0.9106 | 0.9825 | 0.8980 | 0.8323 |
| Tour-Bayes | 0.9996 | **1.0** | 0.9681 | 0.9978 | 0.9840 | 0.9106 | 0.9900 | 0.9220 | 0.8323 |
| Gumbel-max | **1.0** | **1.0** | 0.9891 | **1.0** | **1.0** | **0.9690** | **1.0** | **1.0** | **0.9328** |

To evaluate the robustness of these approaches, we implement the copy-paste attack which involves mixing watermarked text with non-watermarked text and the paraphrasing attack that rewrites the watermarked text using another language model to preserve its meaning. For the copy-paste attack, we randomly mix a proportion $\epsilon$ of non-watermarked text into the watermarked text, maintaining the total length. In the paraphrasing attack, we leverage the paraphrasing model from Zhang et al. (2020). Table 2 compares the detectability against copy-paste attacks for different approaches on 400 tokens with 36 bits embedded. In particular, $\epsilon=0$ means no attack and the performance on the clean samples are reported. In particular, here, for MirrorMark, we present the results with symbol size $m=2$. Besides, we show the results with $\epsilon \in \{0.1, 0.3, 0.5\}$ in Table 9 in Appendix F. We observe that MirrorMark demonstrates greater robustness compared to other methods. For instance,

when $\epsilon = 0.4$, Tour-Wmean, Tour-Bayes, and Gumbel-max all achieve strong performance in terms of AUC, TPR@1%FPR, and bit accuracy, while Gumbel-max exhibits superior detectability even when mixed with a large portion of non-watermarked text. Furthermore, we examine the robustness under different symbol sizes $m \in \{2, 3, 4, 6\}$ in Figure 8 and Figure 9 in Appendix F.5. Setting $m = 2$ yields consistently strong performance. However, while Gumbel-max benefits from $m = 6$, the other two methods do not. This is likely because with $m = 6$, the number of positions becomes $H = 6$. Although this increases the number of tokens allocated per position, in tournament-sampling-based multi-bit watermarking, even without attack, the number of tokens per position is still insufficient to reliably decode 6 bits with high accuracy. In addition, to evaluate the robustness of CABS against token insertion, deletion, and substitution attack, we demonstrate the performance of Gumbel-max sampling based MirrorMark under different ratios of attacks as in Table 6, Table 7, and Table 8 in Appendix F.3.

Table 3 presents the performance of different approaches under paraphrasing attacks and the detectability of MirrorMark with 36 bits embedded in 400 tokens. The performance of MirrorMark is compared with varying symbol sizes $m$. Overall, MirrorMark maintains strong separability between watermarked and non-watermarked samples in terms of AUC, since paraphrasing changes the surface form of sentences but often preserves underlying semantic and statistical patterns that still carry weak watermark signals. In contrast, all methods exhibit poor decoding performance (i.e., low bit accuracy). This degradation arises because paraphrasing alters sentence structure and wording in ways that directly disrupt the precise token-level dependencies required for accurate bit recovery. Achieving robustness against paraphrasing attacks therefore remains an open problem in the design of multi-bit watermarking schemes. In particular, we observe that MirrorMark achieves better detection rate such as AUC and TPR@1%FPR than its zero-bit baselines (e.g., TB ($m$=4) vs. TB (0 bit)), and G-max ($m$=2) vs. G-max (0 bit)), which is because the multi-bit detector selects the message with the highest score, effectively amplifying the residual watermark bias that survives paraphrasing, whereas zero-bit detection lacks this amplification.

Table 3: Detectability against paraphrasing attack across different schemes, where TB represents Tour-Bayes, while G-max denotes Gumbel-max. For multi-bit watermarking approaches, 36-bits messages are embedded on 400 tokens of watermarked samples.

| | MPAC | RSBH | StealthInk | TB (0 bit) | TB ($m$=2) | TB ($m$=3) | TB ($m$=4) | TB ($m$=6) | G-max (0 bit) | G-max ($m$=2) | G-max ($m$=3) | G-max ($m$=4) | G-max ($m$=6) |
|---|---|---|---|---|---|---|---|---|---|---|---|---|---|
| AUC | 0.5743 | 0.3414 | 0.5188 | 0.7925 | 0.8139 | 0.9001 | 0.8938 | 0.8140 | 0.8245 | 0.9306 | 0.9091 | 0.9109 | 0.9025 |
| TPR@1%FPR | 0.0100 | 0.0000 | 0.0050 | 0.2630 | 0.2220 | 0.3200 | 0.3480 | 0.2300 | 0.2800 | 0.5780 | 0.4860 | 0.4620 | 0.3840 |
| Bit Accuracy | 0.5734 | 0.6220 | 0.5673 | – | 0.5216 | 0.5152 | 0.5152 | 0.5123 | – | 0.5434 | 0.5398 | 0.5378 | 0.5333 |

# 6 CONCLUSION

In this work, we propose MirrorMark, a multi-bit and distortion-free watermarking scheme for large language models. By leveraging a mod-1 mirroring process, our method encodes multi-bit messages through a measure-preserving transformation of the sampling randomness for two state-of-the-art (SOTA) distortion-free zero bit watermarking approaches. This design leaves the probability distribution of the model unchanged by the watermark, thereby maintaining text quality. To improve robustness, we introduce a content anchor-based scheduler (CABS) that distributes symbols across positions in a balanced manner, enabling reliable extraction under text distortions such as insertions and deletions. We further provide a theoretical analysis that characterizes detection error, which aligns with our empirical results. Although our experiments focus on extending two representative zero-bit watermarking schemes, we note that MirrorMark applies more generally: any zero-bit watermarking method that samples the next token via random values can be extended to multi-bit watermarking under the MirrorMark framework. Empirical evaluations show that MirrorMark achieves SOTA performance among multi-bit schemes, combining high bit accuracy with strong detectability while preserving the text quality of non-watermarked text. Overall, MirrorMark represents a practical step toward scalable provenance and accountability in LLM deployment. Future work could extend its applicability to additional zero-bit watermarking designs, further strengthen resilience against adversarial paraphrasing, develop adaptive schedulers for dynamic symbol allocation, and explore extensions of the mirroring principle to other modalities beyond text.

## 7 ETHICAL STATEMENT

This work focuses on developing watermarking techniques for LLMs to improve content attribution and accountability. It does not involve human subjects, sensitive personal data, or applications that directly pose risks to safety or security. Our method is designed to enhance transparency in AI-generated content and does not itself generate harmful or biased outputs beyond those already present in the underlying LLM. We adhere to the ICLR Code of Ethics. All research was conducted with integrity and without conflicts of interest or external sponsorship.

## 8 REPRODUCIBILITY STATEMENT

We have made extensive efforts to ensure the reproducibility of our results. The theoretical foundations of MirrorMark, including proofs of measure preservation and error analysis, are fully described in the main text and appendix. Detailed descriptions of the algorithms, scheduling mechanisms, and experimental setups are provided in the paper. Hyperparameters, evaluation metrics, and baselines are clearly specified to allow replication. To promote transparency and reproducibility, we will release our implementation and experimental scripts as open-source, aligned with the final version of this paper.

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

## A  THE USE OF LARGE LANGUAGE MODELS (LLMS)

In preparing this manuscript, we used LLMs as a writing assistant to polish the presentation of our text. Specifically, the LLM was solely employed to improve grammar, clarity, and flow of language in sections drafted by the authors.

## B  RELATED WORK

### B.1  ZERO-BIT WATERMARKING

Due to the discrete linguistic nature of text, designing effective watermarking schemes for digital text remains a challenging problem (Shih (2017)). Early approaches were primarily rule-based, including paraphrasing (Atallah et al. (2002)), syntactic restructuring (Atallah et al. (2001)), and synonym substitution (Topkara et al. (2006)). However, these methods relied on handcrafted transformations and were limited in scalability, naturalness, and robustness. The emergence of LLMs created new opportunities for watermarking because they are generative by nature, producing text token by token under probabilistic distributions. This generative process allows watermarking to be embedded directly in the sampling procedure rather than through post hoc text modifications. For example, Kirchenbauer et al. (2023) introduced the first watermarking scheme for LLMs and

highlighted a key property of reweighting-based watermarking: the watermark can be detected algorithmically without knowledge of the model parameters or access to the LLM API. Their method partitions the vocabulary into red and green token lists using a hash function seeded with the preceding context tokens, and then applies a small bias to the logits of green-list tokens. As a result, the watermarked LLM is more likely to generate green-list tokens. Detection is achieved by reconstructing the same lists and conducting hypothesis testing to evaluate whether a text was generated under the reweighted distribution. Subsequent works further strengthened this scheme to withstand distortion-bounded attacks such as insertion, deletion, and substitution (Kirchenbauer et al. (2024); Zhao et al. (2024)).Specifically, by retaining the configurations of red-green list, Hu et al. (2024) and Dipmark Wu et al. (2024) introduced an evolved family of permutation-based reweighting strategies for watermarking which maintains the expected distribution of the text; i.e., they proposed a stealthy or unbiased reweighting strategy for LLM watermarking. However, the detector in Hu et al. (2024) necessitates access to both the prompt and the output distribution provided by the LLM for a given prompt, which requires the detector possesses knowledge of the prompt used to generate the detected text.

In contrast to distortion-based watermarking, which embeds signals by perturbing token probability distributions, recent works have explored distortion-free approaches based on inverse sampling. For example, Christ et al. (2024) and Kuditipudi et al. (2024) proposed generating watermarked text without modifying the underlying distribution. However, the method of Christ et al. (2024) leaves open the challenge of resilience against text corruption. The scheme of Kuditipudi et al. (2024), although tailored for robust detection, it depends on hundreds of resampling steps during detection, which is computationally prohibitive for long texts. Beyond inverse sampling, other distortion-free techniques have also emerged. Aaronson & Kirchner (2022) introduced a Gumbel sampling–based watermark, while Dathathri et al. (2024) developed SynthIDText, which embeds watermarks through tournament sampling.

### B.2 MULTI-BIT WATERMARKING

Fernandez et al. (2023) extended the scheme of Kirchenbauer et al. (2023) by encoding multi-bit messages through message-specific green lists, obtained by shifting the vocabulary permutation according to the message. Similarly, Qu et al. (2024) cyclically shifted vocabulary permutations based on the message and biased tokens in the green list to enable efficient multi-bit decoding. They further incorporated error-correcting codes to strengthen robustness. However, overlapping shifts introduce interference that diminishes message distinctiveness and weakens statistical separation. To achieve reliable decoding, a stronger bias must be applied—at the cost of greater text distortion. MPAC Yoo et al. (2024) introduced a multi-color technique. In this scheme, the pseudorandom vocabulary permutation (seeded by prior tokens) is partitioned into multiple equal-length segments, each represented by a distinct color. Message bits are then encoded by selecting color segments. For example, dividing the vocabulary into four colors requires two binary bits to specify a segment. Thus, a 2-bits message corresponds to four binary bits in total, as each position requires two bits to indicate its color. During generation, the logits of tokens within the chosen color segment corresponding to the message are boosted by a fixed bias, steering the next token toward that segment.

Beyond color-based methods, other approaches focus on reducing or eliminating distortion. StealthInk( Jiang et al. (2025)) perturbs the distribution at each generation step but designs the watermark such that the overall distribution is preserved in expectation, maintaining fluency and text quality. However, its detectability remains limited. In contrast, Kordi Boroujeny et al. (2024) and Zamir (2024) proposed multi-bit schemes that are fully distortion-free, ensuring identical input–output distributions. Yet, because they build on the inverse-sampling framework of Christ et al. (2024), they inherit its unresolved weakness: resilience to text corruption remains an open challenge, with no practical solution to date.

## C PROOF FOR THEOREM 4.1: THEORETICAL EER OF MIRRORMARK

In the following, we analyze the theoretical EER of Gumbel-max and tournament sampling based MirrorMark with the number of positions $H = 1$.

## C.1 GUMBEL-MAX SAMPLING-BASED MULTIBIT WATERMARKING

Recall the sequence-level score of text $W$ for message $M$ is derived as follows, where sk is the watermark key and $u_t$ is the random value seeded by sk, and $h$ context tokens from $W_{t-h:t}$,

$$C_M(W, \text{sk}) = \frac{1}{T} \sum_{t=1}^{T} S_M(W_t, \text{sk}), \qquad S_M(W_t, \text{sk}) = \ln \frac{1}{1 - \Psi(u_t, \psi(M))}. \tag{16}$$

Under the null hypothesis $\mathcal{H}_0$, all $C_M$ share the same non-watermarked distribution. Under the alternative hypothesis $\mathcal{H}_1$, exactly one index $M^\star$ is "signal" while the remaining $2^m - 1$ are "null", where $M^\star$ represents the message embedded by the encoder.

Under $\mathcal{H}_0$, $\Psi \sim \text{Uniform}(0,1)$ and hence $S_M(W_t, \text{sk}) \overset{d}{=} \text{Exp}(1)$. Therefore,

$$\mathbb{E}[C_M(W, \text{sk}) \mid \mathcal{H}_0] = \mu_{\mathcal{H}_0} = 1, \qquad \text{Var}(C_M(W, \text{sk}) \mid \mathcal{H}_0) = \sigma_{\mathcal{H}_0}^2 = \frac{1}{T}. \tag{17}$$

Referring to equation (14) in Fernandez et al. (2023), under $\mathcal{H}_1$, for the $t$-th watermarked token with bias $p_t \in (0, 1]$, $\Psi(u_t, \psi(M^\star)) \sim \text{Beta}\left(\frac{1}{p_t}, 1\right)$ so that $1 - \Psi \sim \text{Beta}\left(1, \frac{1}{p_t}\right)$. According to the digamma function $\psi_0$ and trigamma function $\psi_1$ defined in Lemma C.3,

$$\mathbb{E}[S_{M^\star}(W_t, \text{sk})] = \psi_0\left(1 + \frac{1}{p_t}\right) - \psi_0(1) := H_{1/p_t}, \qquad \text{Var}\left(S_{M^\star}(W_t, \text{sk})\right) = \psi_1(1) - \psi_1\left(1 + \frac{1}{p_t}\right). \tag{18}$$

Therefore, for the true message $M^\star$,

$$\mathbb{E}[C_{M^\star} \mid \mathcal{H}_1] = \mu_{\mathcal{H}_1} = \frac{1}{T} \sum_{t=1}^{T} H_{1/p_t}, \qquad \text{Var}[C_{M^\star} \mid \mathcal{H}_1] = \sigma_{\mathcal{H}_1}^2 = \frac{1}{T^2} \sum_{t=1}^{T} \left[\psi_1(1) - \psi_1(1 + \frac{1}{p_t})\right]. \tag{19}$$

Let $Z = \max_{M \in \{0, \ldots, 2^m - 1\}} \{C_M\}$. Since the sequence-level score $C_M(W, \text{sk})$ averages over $T$ tokens, the Central Limit Theorem (CLT) suggests that, as $T$ grows, $C_M(W, \text{sk}) \sim \mathcal{N}(\mu_{\mathcal{H}_0}, \sigma_{\mathcal{H}_0}^2)$. Besides, although the statistics $\{C_M\}$ are not strictly independent since they are calculated on the same text, each $C_M$ is an average of $T$ per-token scores with variance $O(1/T)$. As $T$ grows, the variance of each $C_M$ shrinks. Therefore, the event $\{Z > \tau\}$ is potentially caused by one candidate $C_M$ exhibiting an unusually large deviation, rather than by simultaneous moderate deviations of many correlated $C_M$. Hence, we can approximate $\{C_M\}$ as independent. By Lemma C.1, we obtain

$$\text{FPR}(\tau) = \Pr\left(Z > \tau \mid \mathcal{H}_0\right) = 1 - \left[\Phi\left(\frac{\tau - \mu_{\mathcal{H}_0}}{\sigma_{\mathcal{H}_0}}\right)\right]^{2^m} \approx 2^m Q\left(\frac{\tau - \mu_{\mathcal{H}_0}}{\sigma_{\mathcal{H}_0}}\right), \tag{20}$$

where as defined in Lemma C.1, $\Phi(\cdot)$ denotes the cumulative distribution function of the standard normal distribution while $Q(\cdot)$ is the gaussian tail probability.

Similarly, under $\mathcal{H}_1$, we can approximate $C_{M^\star}(W, \text{sk}) \sim \mathcal{N}(\mu_{\mathcal{H}_1}, \sigma_{\mathcal{H}_1}^2)$ and calculate FNR as

$$\text{FNR}(\tau) = \Pr\left(Z < \tau \mid \mathcal{H}_1\right) = \Pr\left(C_{M^\star} < \tau \mid \mathcal{H}_1\right) = \Phi\left(\frac{\tau - \mu_{\mathcal{H}_1}}{\sigma_{\mathcal{H}_1}}\right) = Q\left(\frac{\mu_{\mathcal{H}_1} - \tau}{\sigma_{\mathcal{H}_1}}\right), \tag{21}$$

To solve the EER threshold, let $\text{FPR}(\tau^{\text{eer}}) = \text{FNR}(\tau^{\text{eer}})$. Let

$$z_0(\tau^{\text{eer}}) = \frac{\tau^{\text{eer}} - \mu_{\mathcal{H}_0}}{\sigma_{\mathcal{H}_0}},$$
$$z_1(\tau^{\text{eer}}) = \frac{\mu_{\mathcal{H}_1} - \tau^{\text{eer}}}{\sigma_{\mathcal{H}_1}}, \tag{22}$$

we write $z_0 = z_0(\tau^{\text{eer}})$ and $z_1 = z_1(\tau^{\text{eer}})$ for brevity, combining equation 20 and equation 21, then

$$z_1^2 = z_0^2 - 2m \ln 2 - 2 \ln\left(\frac{z_1}{z_0}\right). \tag{23}$$

Since $z_0$ and $z_1$ are of the same order as the EER operating points, $2\ln\left(\frac{z_1}{z_0}\right)$ is lower-order. Thus, we obtain

$$z_1^2 \approx z_0^2 - 2m\ln 2. \tag{24}$$

Let $\Delta\mu = \mu_{\mathcal{H}_1} - \mu_{\mathcal{H}_0}$, and thus,

$$\sigma_0 z_0 + \sigma_1 z_1 = \Delta\mu \tag{25}$$

We first take a baseline at $m = 0$. Therefore,

$$\tau_{\text{baseline}}^{\text{eer}} = \frac{\mu_{\mathcal{H}_0}\sigma_{\mathcal{H}_1} + \mu_{\mathcal{H}_1}\sigma_{\mathcal{H}_0}}{\sigma_{\mathcal{H}_0} + \sigma_{\mathcal{H}_1}} \tag{26}$$

Thus, plug equation 26 into equation 22,

$$z_0 = z_1 = z = \frac{\Delta\mu}{\sigma_{\mathcal{H}_0} + \sigma_{\mathcal{H}_1}}. \tag{27}$$

Now we take a first order perturbation for $m > 0$. Let

$$z_0 = z + \varepsilon_0, \qquad z_1 = z + \varepsilon_1, \tag{28}$$

since $z_1^2 - z_0^2 = (z + \varepsilon_1)^2 - (z + \varepsilon_0)^2 \approx 2z(\varepsilon_1 - \varepsilon_0)$, combining equation 24,

$$2z(\varepsilon_1 - \varepsilon_0) = -2m\ln 2. \tag{29}$$

Therefore,

$$\varepsilon_1 - \varepsilon_0 = -\frac{m\ln 2}{z} \tag{30}$$

Combining the identity $\mu_{\mathcal{H}_0} + \sigma_{\mathcal{H}_0}z_0 = \tau^{\text{eer}} = \mu_{\mathcal{H}_1} - \sigma_{\mathcal{H}_1}z_1$, we obtain

$$\sigma_{\mathcal{H}_0}\varepsilon_0 + \sigma_{\mathcal{H}_1}\varepsilon_1 = 0 \tag{31}$$

Furthermore, combining equation 30 and equation 31, we obatin

$$\varepsilon_1 = -\frac{\sigma_{\mathcal{H}_0}}{\Delta\mu} m\ln 2$$
$$\varepsilon_0 = \frac{\sigma_{\mathcal{H}_1}}{\Delta\mu} m\ln 2. \tag{32}$$

Therefore,

$$z_1 \approx z - \frac{\sigma_{\mathcal{H}_0}}{\Delta\mu} m\ln 2 \tag{33}$$

Substituting equation 27 gives the EER approximation

$$\text{EER}_{\text{Gumbel}} \approx Q\left(\frac{\Delta\mu}{\sigma_{\mathcal{H}_0} + \sigma_{\mathcal{H}_1}} - \frac{\sigma_{\mathcal{H}_0}}{\Delta\mu} m\ln 2\right). \tag{34}$$

For clarity, suppose that all tokens share the same bias $p_t \equiv p$. Then

$$\mu_{\mathcal{H}_1} = H_{1/p}, \quad \sigma_{\mathcal{H}_1}^2 = \frac{\psi_1(1) - \psi_1(1 + 1/p)}{T}, \quad \Delta\mu = H_{1/p} - 1. \tag{35}$$

Plugging these into equation 34, therefore,

$$\text{EER}_{\text{Gumbel}} \approx Q\left(\frac{(H_{1/p} - 1)\sqrt{T}}{1 + \sqrt{\psi_1(1) - \psi_1(1 + 1/p)}} - \frac{m\ln 2}{(H_{1/p} - 1)\sqrt{T}}\right). \tag{36}$$

**Relating $p$ to the vocabulary size.** In Gumbel-max sampling each of the $V$ vocabulary items is assigned an independent PRF value. Suppose $\kappa V$ candidates enter a uniform competition, which means each candidate receives an i.i.d. PRF value $U \sim \mathrm{Uniform}(0, 1)$ and the winner achieves $U_{(\kappa V)} = \max\{U_1, \ldots, U_{\kappa V}\} \sim \mathrm{Beta}(\kappa V, 1)$. Therefore, we can identify an effective pool size $\kappa V \simeq 1/p$. Furthermore, we set $\frac{1}{p} = \kappa V$ with $\kappa > 0$ estimated once in a development set.

Substituting $\frac{1}{p} = \kappa V$ into equation 36 gives

$$\mathrm{EER}_{\mathrm{Gumbel}} \approx Q\left(\frac{(H_{\kappa V} - 1)\sqrt{T}}{1 + \sqrt{\psi_1(1) - \psi_1(1 + \kappa V)}} - \frac{m \ln 2}{(H_{\kappa V} - 1)\sqrt{T}}\right). \tag{37}$$

For large $V$, using the expansions by Lemma C.3

$$H_{\kappa V} = \ln(\kappa V) + \gamma, \qquad \psi_1(1) - \psi_1(1 + \kappa V) = \frac{\pi^2}{6}, \tag{38}$$

where $\gamma$ is Euler's constant defined in Lemma C.3, we obtain the asymptotic form

$$\mathrm{EER}_{\mathrm{Gumbel}} \approx Q\left(\frac{(\ln V + \ln \kappa + \gamma - 1)\sqrt{T}}{1 + \pi/\sqrt{6}} - \frac{m \ln 2}{(\ln V + \ln \kappa + \gamma - 1)\sqrt{T}}\right). \tag{39}$$

Let

$$z_V = \frac{(\ln V + \ln \kappa + \gamma - 1)\sqrt{T}}{1 + \pi/\sqrt{6}} - \frac{m \ln 2}{(\ln V + \ln \kappa + \gamma - 1)\sqrt{T}}, \tag{40}$$

therefore,

$$\log \mathrm{EER}_{\mathrm{Gumbel}} = -(z_V)^2/2 - \log(z_V\sqrt{2\pi})$$

$$= -\frac{T}{2\left(1 + \frac{\pi}{\sqrt{6}}\right)^2}\left(\ln V + \ln \kappa + \gamma - 1\right)^2 + \frac{m \ln 2}{1 + \frac{\pi}{\sqrt{6}}} - \frac{(m \ln 2)^2}{2T\left(\ln V + \ln \kappa + \gamma - 1\right)^2} -$$

$$\log\left(z_V\sqrt{2\pi}\right). \tag{41}$$

Thus,

$$\log \mathrm{EER}_{\mathrm{Gumbel}} \sim -\Theta\left(T\left(\ln \#\mathrm{PRF}\right)^2 - m\right), \tag{42}$$

which means as the number of PRF draws grows, EER decreases as the rate of $exp(-\Theta(T(\ln \#\mathrm{PRF})^2))$. Furthermore, the increase in symbol size $m$ will lead to a higher EER.

## C.2 TOURNAMENT SAMPLING BASED MULTI-BIT WATERMARKING

Recall the score of $t$-th token for message $M$

$$S_M(\mathsf{sk}, W_t) = \frac{1}{L}\sum_{\ell=1}^{L}\alpha_\ell\,\Psi(u_{t,l}, \psi(M)), \tag{43}$$

and the sequence-level statistic for message $M$ as the per-token average

$$C_M(\mathsf{sk}, W) = \frac{1}{T}\sum_{t=1}^{T}S_M(\mathsf{sk}, W_t). \tag{44}$$

In this derivation, we treat $m = 1$, where the construction enforces $S_0(\mathsf{sk}, W_t) + S_1(\mathsf{sk}, W_t) \equiv 1$ per token from the property of mirroring. Define

$$Z_t = \max\{S_0(\mathsf{sk}, W_t), S_1(\mathsf{sk}, W_t)\} = \frac{1}{2} + \left|S_0(\mathsf{sk}, W_t) - \frac{1}{2}\right|, \tag{45}$$

and detect with $C_{\max} = \frac{1}{T} \sum_{t=1}^{T} Z_t$.

Under null hypothesis, at each layer $\ell$, $\Psi \sim \text{Uniform}(0, 1)$, hence

$$\mathbb{E}\big[S_0(\text{sk}, W_t) \mid \mathcal{H}_0\big] = \frac{1}{L} \sum_{\ell=1}^{L} \frac{\alpha_\ell}{2} = \frac{1}{2}, \qquad \text{Var}\big[S_0(\text{sk}, W_t) \mid \mathcal{H}_0\big] = \frac{1}{L^2} \sum_{\ell=1}^{L} \alpha_\ell^2 \, \text{Var}(\Psi) = \frac{A}{12 \, L^2}, \tag{46}$$

where $A = \sum_{\ell=1}^{L} \alpha_\ell^2$. Approximating $S_0(\text{sk}, W_t) - \frac{1}{2}$ by $\mathcal{N}(0, A/(12L^2))$ and using the Lemma C.2,

$$\mathbb{E}\big|S_0(\text{sk}, W_t) - \tfrac{1}{2}\big| = \sqrt{\frac{A}{6\pi \, L^2}}, \qquad \text{Var}\big|S_0(\text{sk}, W_t) - \tfrac{1}{2}\big| = \frac{A}{12L^2}\left(1 - \frac{2}{\pi}\right). \tag{47}$$

Therefore

$$\mathbb{E}[Z_t \mid \mathcal{H}_0] = \frac{1}{2} + \sqrt{\frac{A}{6\pi \, L^2}}, \qquad \text{Var}[Z_t \mid \mathcal{H}_0] = \frac{A}{12L^2}\left(1 - \frac{2}{\pi}\right), \tag{48}$$

and by the CLT for the average $C_{\max} = \frac{1}{T} \sum_t Z_t$,

$$\mu_{\mathcal{H}_0} = \mathbb{E}[C_{\max} \mid \mathcal{H}_0] = \frac{1}{2} + \sqrt{\frac{A}{6\pi \, L^2}}, \qquad \sigma_{\mathcal{H}_0}^2 = \text{Var}[C_{\max} \mid \mathcal{H}_0] = \frac{A}{12L^2 T}\left(1 - \frac{2}{\pi}\right). \tag{49}$$

We can derive the FPR as

$$\text{FPR} = \Pr[C_{max} > \tau | H_0] = Q\big(\frac{\tau - \mu_{\mathcal{H}_0}}{\sigma_{\mathcal{H}_0}}\big) \tag{50}$$

On the other hand, under the alternative hypothesis $\mathcal{H}_1$, at layer $\ell$, refer to Corollary 28 in SynthID-Text (Dathathri et al. (2024)), the mirrored random variable described by cdf and pdf as follows,

$$F_{\Psi_\ell}(x) = C_{wm}^\ell x + (1 - C_{wm}^\ell)x^2, \qquad f_{\Psi_\ell}(x) = C_{wm}^\ell + 2(1 - C_{wm}^\ell)x, \tag{51}$$

where $C_{wm}^\ell \in [0, 1)$ represents the collision probability at layer $\ell$ as defined in Definition 22 in SynthIDText, which is the probability that two samples drawn i.i.d. from the probability distribution of tokens at layer $l$ are the same. Hence,

$$\mathbb{E}[\Psi_\ell] = \frac{2}{3} - \frac{C_{wm}^\ell}{6}, \qquad \text{Var}(\Psi_\ell) = \frac{2 + 2C_{wm}^\ell - (C_{wm}^\ell)^2}{36}. \tag{52}$$

Hence the per-token $S_0(t) = \frac{1}{L} \sum_\ell \alpha_\ell \Psi_\ell$ has

$$\mu_S = \mathbb{E}\big[S_0(\text{sk}, W_t) \mid \mathcal{H}_1\big] = \frac{1}{L} \sum_{\ell=1}^{L} \alpha_\ell \left(\frac{2}{3} - \frac{C_{wm}^\ell}{6}\right),$$

$$v_S = \text{Var}\big[S_0(\text{sk}, W_t) \mid \mathcal{H}_1\big] = \frac{1}{L^2} \sum_{\ell=1}^{L} \alpha_\ell^2 \frac{2 + 2C_{wm}^\ell - (C_{wm}^\ell)^2}{36}. \tag{53}$$

Let $\mu_\Delta = \mu_S - \frac{1}{2}$. Using the Lemma C.2 again,

$$\mathbb{E}|S_0(\text{sk}, W_t) - \tfrac{1}{2}| = \sqrt{\frac{2}{\pi}} \, \sqrt{v_S} \, \exp\left(-\frac{\mu_\Delta^2}{2v_S}\right) + \mu_\Delta\left[1 - 2\Phi\left(-\frac{\mu_\Delta}{\sqrt{v_S}}\right)\right] =: \text{FNmean}(\mu_\Delta, v_S),$$

$$\text{Var}(|S_0(\text{sk}, W_t) - \tfrac{1}{2}|) = \mu_\Delta^2 + v_S - \big(\mathbb{E}|S_0(\text{sk}, W_t) - \tfrac{1}{2}|\big)^2 =: \text{FNvar}(\mu_\Delta, v_S). \tag{54}$$

Thus for $Z_t = \frac{1}{2} + |S_0(\text{sk}, W_t) - \frac{1}{2}|$,

$$\mathbb{E}[Z_t | \mathcal{H}_1] = \frac{1}{2} + \text{FNmean}(\mu_\Delta, v_S), \qquad \text{Var}[Z_t | \mathcal{H}_1] = \text{FNvar}(\mu_\Delta, v_S),$$

and

$$\mu_{\mathcal{H}_1} = \mathbb{E}[C_{\max} \mid \mathcal{H}_1] = \mathbb{E}[Z_t | \mathcal{H}_1], \qquad \sigma_{\mathcal{H}_1}^2 = \text{Var}[C_{\max} \mid \mathcal{H}_1] = \text{Var}[Z_t | \mathcal{H}_1]/T. \tag{55}$$

We can derive the FNR as

$$\text{FNR} = \Pr[C_{max} < \tau | H_1] = \Phi\left(\frac{\tau - \mu_{\mathcal{H}_1}}{\sigma_{\mathcal{H}_1}}\right) \tag{56}$$

Combining equation 50 and equation 56, let FPR = FNR, we can derive the EER is

$$\text{EER}_{\text{tour}} = \text{FPR}(\tau^{\text{eer}}) = \text{FNR}(\tau^{\text{eer}}) = Q\left(\frac{\mu_{\mathcal{H}_1} - \mu_{\mathcal{H}_0}}{\sigma_{\mathcal{H}_0} + \sigma_{\mathcal{H}_1}}\right) = \Phi\left(-\frac{\mu_{\mathcal{H}_1} - \mu_{\mathcal{H}_0}}{\sigma_{\mathcal{H}_0} + \sigma_{\mathcal{H}_1}}\right). \tag{57}$$

For easier analysis, we assume $\frac{1}{L}\sum_{\ell=1}^{L}\alpha_\ell = 1$, define $C_1 := \frac{1}{L}\sum_{\ell=1}^{L}\alpha_\ell C_{wm}^\ell$ and $C_2 := \frac{1}{L}\sum_{\ell=1}^{L}\alpha_\ell^2 \frac{2+2C_{wm}^\ell-(C_{wm}^\ell)^2}{36}$. Therefore,

$$\begin{aligned}
\mu_{\mathcal{H}_0} &= \frac{1}{2} + \sqrt{\frac{1}{6\pi L}} \\
\sigma_{\mathcal{H}_0} &= \sqrt{\frac{1}{12LT}\left(1 - \frac{2}{\pi}\right)}, \\
\mu_\Delta &= \frac{1 - C_1}{6}, \\
v_S &= \frac{C_2}{L}.
\end{aligned} \tag{58}$$

Since $v_S \to 0$ as $L \to \infty$, the folded-normal mean in equation 54 satisfies

$$\text{FNmean}(\mu_\Delta, v_S) = \sqrt{\frac{2}{\pi}}\sqrt{v_S}\, e^{-\mu_\Delta^2/(2v_S)} + \mu_\Delta\left[1 - 2\Phi\left(-\frac{\mu_\Delta}{\sqrt{v_S}}\right)\right] = \frac{1 - C_1}{6}. \tag{59}$$

Therefore,

$$\mu_{\mathcal{H}_1} = \frac{1}{2} + \text{FNmean}(\mu_\Delta, v_S) = \frac{4 - C_1}{6} \tag{60}$$

Meanwhile,

$$\sigma_{\mathcal{H}_1} = \sqrt{\mu_\Delta^2 + v_S - \left(\text{FNmean}(\mu_\Delta, v_S)\right)^2} = \sqrt{v_S} = \sqrt{\frac{C_2}{L}}. \tag{61}$$

Let

$$\kappa_0 = \sqrt{\frac{1 - \frac{2}{\pi}}{12}} \tag{62}$$

and

$$\kappa_1(C_2) = \sqrt{C_2}, \tag{63}$$

then

$$\sigma_{\mathcal{H}_0} = \frac{\kappa_0}{\sqrt{LT}}, \qquad \sigma_{\mathcal{H}_1} = \frac{\kappa_1(C_2)}{\sqrt{LT}}. \tag{64}$$

Hence, by equation 57,

$$\text{EER}_{\text{tour}} = Q\left(\frac{\mu_{\mathcal{H}_1} - \mu_{\mathcal{H}_0}}{\sigma_{\mathcal{H}_0} + \sigma_{\mathcal{H}_1}}\right) = Q\left(\frac{\frac{1-C_1}{6} - \frac{1}{\sqrt{6\pi L}}}{(\kappa_0 + \kappa_1(C_2))/\sqrt{LT}}\right). \tag{65}$$

Define

$$\Gamma(C_1, C_2) = \frac{\frac{1-C_1}{6}}{\kappa_0 + \kappa_1(C_2)}, \tag{66}$$

where $\Gamma(C_1, C_2) \in (0, \frac{1}{6\kappa_0}] \approx (0, 0.694]$,

and

$$\beta(C_1, C_2) = \frac{1}{\sqrt{6\pi}} \cdot \frac{1}{\kappa_0 + \kappa_1(C_2)}. \tag{67}$$

Then

$$\mathrm{EER}_{\mathrm{tour}} = Q\Big(\Gamma(C_1, C_2)\sqrt{LT} - \beta(C_1, C_2)\sqrt{T}\Big). \tag{68}$$

Using Lemma C.1, we obtain

$$\mathrm{EER}_{\mathrm{tour}} \approx \frac{\exp\Big(-\frac{1}{2}\big(\Gamma(C_1, C_2)\sqrt{LT} - \beta(C_1, C_2)\sqrt{T}\big)^2\Big)}{\sqrt{2\pi}\,\big(\Gamma(C_1, C_2)\sqrt{LT} - \beta(C_1, C_2)\sqrt{T}\big)}. \tag{69}$$

In a full $L$-layer tournament, the number of PRF evaluations per decoding step is

$$\#\mathrm{PRF} = 2^{L+1} - 2. \tag{70}$$

Hence

$$L = \log_2\Big(\tfrac{\#\mathrm{PRF}}{2} + 1\Big). \tag{71}$$

Substituting this into equation 69 gives

$$\mathrm{EER}_{\mathrm{tour}} \approx \frac{\exp\big(-\frac{T}{2}\,E(C_1, C_2, \#\mathrm{PRF})\big)}{\sqrt{2\pi T}\,\Big(\Gamma(C_1, C_2)\sqrt{\log_2(\tfrac{\#\mathrm{PRF}}{2} + 1)} - \beta(C_1, C_2)\Big)}, \tag{72}$$

where

$$E(C_1, C_2, \#\mathrm{PRF}) = \Gamma(C_1, C_2)^2 \, \log_2\Big(\tfrac{\#\mathrm{PRF}}{2} + 1\Big)$$
$$- 2\Gamma(C_1, C_2)\beta(C_1, C_2)\,\sqrt{\log_2\Big(\tfrac{\#\mathrm{PRF}}{2} + 1\Big)} + \beta(C_1, C_2)^2. \tag{73}$$

For large $\#\mathrm{PRF}$,

$$\log\mathrm{EER}_{\mathrm{tour}} = -\frac{\Gamma(C_1, C_2)^2}{2}\,T\,\log_2\Big(\tfrac{\#\mathrm{PRF}}{2} + 1\Big) - O\Big(T\sqrt{\log_2(\tfrac{\#\mathrm{PRF}}{2} + 1)}\Big). \tag{74}$$

Therefore,

$$\log\mathrm{EER}_{\mathrm{tour}} \sim -\Theta\Big(\Gamma(C_1, C_2)^2\,T\,\log\#\mathrm{PRF}\Big), \tag{75}$$

where $\Gamma(C_1, C_2) > 0$ decreases as the collision levels $C_1$ or $C_2$ increase. Since both aggregated quantities $C_1$ and $C_2$ are increasing functions of the layer-wise collision probabilities $C_\ell^{\mathrm{wm}}$, higher collision levels lead to larger $(C_1, C_2)$ and therefore a smaller value of $\Gamma(C_1, C_2)$. For notational simplicity, we denote the collision dependence using a single effective collision level $c \in [0, 1)$ and define $\Gamma(c) := \Gamma(C_1, C_2)$, where $c$ increases with larger $C_\ell^{\mathrm{wm}}$.

Under this unified notation, the equation 75 becomes

$$\log\mathrm{EER}_{\mathrm{tour}} \sim -\Theta\Big(T\Gamma(c)^2 \log\#\mathrm{PRF}\Big). \tag{76}$$

**Lemma C.1** *Let $X_1, \ldots, X_K \overset{\text{i.i.d.}}{\sim} \mathcal{N}(0, 1)$ and $G_K = \max_i X_i$. For large $z$,*

$$\Pr(G_K > z) = 1 - \big(1 - Q(z)\big)^K = K\,Q(z)\,\big(1 + o(1)\big), \tag{77}$$

*where $\Phi(z)$ denotes the cumulative distribution function of the standard normal distribution[4] and $Q(z) = 1 - \Phi(z)$ is its Gaussian tail probability.*

---

[4]https://en.wikipedia.org/wiki/Normal_distribution

**Lemma C.2** *Let $X \sim \mathcal{N}(\mu, \sigma^2)$ and $Y = |X|$. Then*

$$\mathbb{E}[Y] = \sigma\sqrt{\tfrac{2}{\pi}} \exp\left(-\tfrac{\mu^2}{2\sigma^2}\right) + \mu\left[1 - 2\Phi\left(-\tfrac{\mu}{\sigma}\right)\right], \tag{78}$$

*and*

$$\mathrm{Var}(Y) = \mu^2 + \sigma^2 - \left(\mathbb{E}[Y]\right)^2. \tag{79}$$

**Lemma C.3** *Let $T \sim \mathrm{Beta}(a, b)$. Then*

$$\mathbb{E}[\ln T] = \psi_0(a) - \psi_0(a+b), \qquad \mathrm{Var}(\ln T) = \psi_1(a) - \psi_1(a+b), \tag{80}$$

*where the digamma function $\psi_0(x)$ for $x > 0$ is defined as*

$$\psi_0(x) = -\gamma + \sum_{n=0}^{\infty}\left(\frac{1}{n+1} - \frac{1}{n+x}\right), \tag{81}$$

*with $\gamma$ the Euler's constant. The trigamma function $\psi_1(x)$ for $x > 0$ is defined as*

$$\psi_1(x) = \sum_{n=0}^{\infty}\frac{1}{(n+x)^2}. \tag{82}$$

*In particular, for $x > 0$, let the generalized harmonic number $H_x = \psi_0(x+1) - \psi_0(1)$. As $x \to \infty$,*

$$H_x = \ln x + \gamma. \tag{83}$$

# D ALGORITHM 3 OF CABS

---
**Algorithm 3** CABS Scheduling
---
**Input:** Eligibility function $\mathsf{Elig}(\cdot)$, secret key $\mathsf{sk}$, message length $H$, counter vector $\boldsymbol{c} \leftarrow \boldsymbol{0}^H$, queue $Q \leftarrow []$, window size $W$, $f$, count of tokens within a frame $\ell \leftarrow 0$, context length $h$, min_len, max_len, sequence $\boldsymbol{x}_{0:T-1}$
**Output:** Position assignment for each eligible token
 1: **for** $i = h, \dots, T-1$ **do**
 2:    **if** not $\mathsf{Elig}(\boldsymbol{x}_{i-h:i-1})$ **then**
 3:       **continue**
 4:    **else**
 5:       $F \leftarrow \mathsf{Hash}(Q)$
 6:       $Q.\mathsf{enqueue}(\boldsymbol{x}_i)$
 7:       **if** $|Q| > W$ **then**
 8:          $Q.\mathsf{dequeue}(\boldsymbol{x}_{i-W})$
 9:       **end if**
10:       $min\_pos = \arg\min(\boldsymbol{c})$ %% Select the positions with the fewest tokens from counter vector for tokens-to-positions mapping
11:       $pos \sim \mathsf{Unif}(min\_pos)$ %% Randomly select a position, seeded by $\mathsf{PRF}_{\mathsf{sk}}(\boldsymbol{x}_{i-h:i-1})$
12:       $\boldsymbol{c}_{pos} \leftarrow \boldsymbol{c}_{pos} + 1$ %% Increment the count for the assigned position
13:       $\ell \leftarrow \ell + 1$ %% Increment the count of tokens for the current frame
14:       $cut$ (true or false) $\leftarrow \left(\ell \geq \mathsf{min\_len} \wedge (F \bmod 2^f == 0)\right) \vee (\ell \geq \mathsf{max\_len})$ %% Whether to end the current frame and start a new one
15:       **if** $cut$ **then**
16:          $\boldsymbol{c} \leftarrow \boldsymbol{0}^H, \quad Q \leftarrow [], \quad \ell \leftarrow 0$
17:       **end if**
18:    **end if**
19: **end for**
---

# E   EXPERIMENTAL SETUP

Unless otherwise specified, all experiments use the Llama-2-7B model (Touvron et al. (2023)) on a text completion task. We construct prompts from the RealNewsLike subset of C4 (Raffel et al. (2020)). We randomly select 500 documents, truncate each document to obtain a prefix, and ask the model to generate a continuation conditioned on that prefix. Most results in the main paper are reported on this setting. To assess the generality of MirrorMark beyond this model and task, we additionally evaluate on the Gemma-7B-it (Team et al. (2024)) model on an instruction-following task. We randomly sample 500 prompts from the ELI5 dataset (Fan et al. (2019)), treat them as user instructions, and generate model responses.

Following Dathathri et al. (2024), we use top-100 sampling with temperature $T = 1.0$ for all evaluated watermarking approaches. For CABS, we use the same hyperparameters throughout the experiments, where $h = 4$, $f = 3$, $W = 4$, and max_len = max_factor $\cdot H$ with max_factor = 1.5 and $H$ denoting the number of positions in the context. Although as we evaluated in Fig. 5, the performance of Tournament sampling based MirrorMark with varying number of layers shows comparable performance, by following Dathathri et al. (2024), our experiments use a default of 30 tournament layers. For each combination of $m$ and base model in tournament-sampling–based MirrorMark, we train a separate Bayesian detector using 10,000 watermarked samples and 10,000 non-watermarked samples generated for that specific value of $m$. We randomly split the watermarked and non-watermarked feature files into an 80% training set and a 20% validation set. The detector is trained with the Adam optimizer using a learning rate of $3 \times 10^{-3}$, a batch size of 64, and up to 100 epochs. We select the model that achieves the highest validation TPR at 1% FPR, and report its performance in the main paper.

For all baseline comparisons, we follow the default symbol sizes $m$ specified in the original papers. In particular, MPAC uses $m = 2$, StealthInk uses $m = 1$, and RSBH uses $m = 6$.

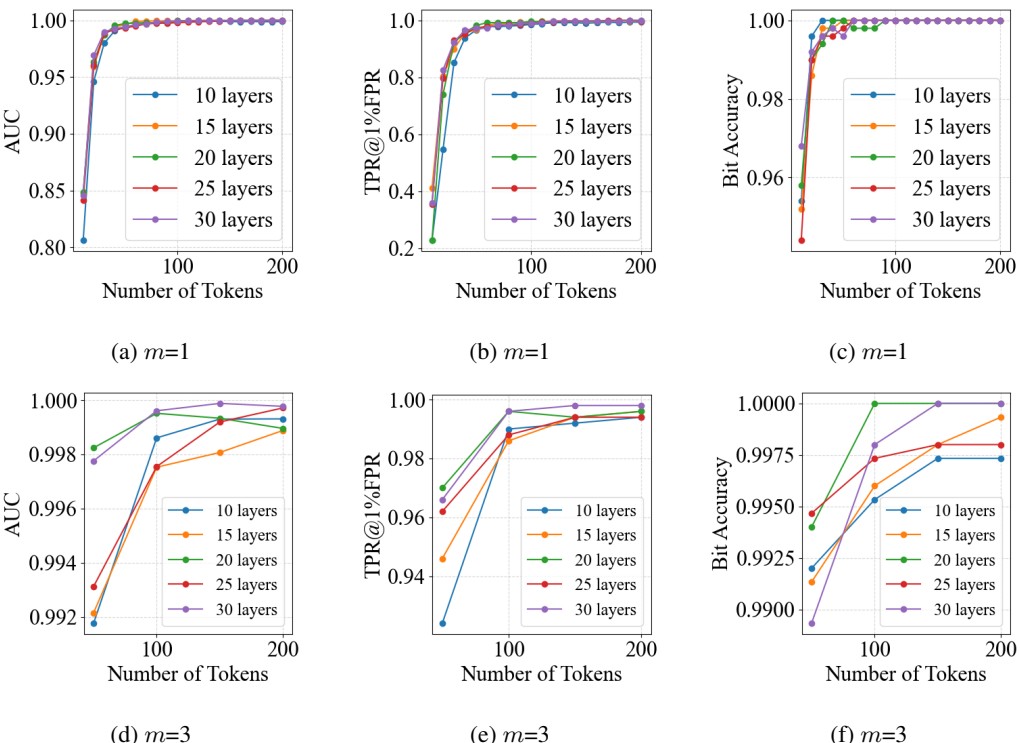

Figure 5: Detectability of tournament sampling based MirrorMark (Tour-Bayes) compared between varying number of layers with $m \in \{1, 3\}$ and $H$=1.

# F  ADDITIONAL RESULTS

## F.1  DISCUSSION ON RESERVING A NULL SYMBOL

In our mod-1 mirroring scheme, each message $M$ is associated with a center $\psi(M) = \frac{M}{2^{m+1}}$, and mirroring is applied as $\Psi(u; \psi(M)) = (2\psi(M) - u) \bmod 1$. The only difference between the reserve and non-reserve designs lies in whether the last symbol $M = 2^m - 1$ is mirrored. In the reserve version this symbol is designated as a null symbol and is not mirrored during generation, while in the non-reserve version all symbols are mirrored, including $M = 2^m - 1$.

The motivation for introducing the null symbol is to allow the flexibility of the multi-bit extension to revert to the original zero-bit case when desired. Besides, When $m = 1$, only two payload symbols ($M = \{0, 1\}$) are used for embedding, and both designs apply exactly the same mirroring transformations to these two messages. Thus, for $m = 1$, the reserve and non-reserve designs are theoretically expected to yield identical performance.

For larger message sizes ($m \in \{2, 3, 4\}$), we empirically compare the two designs in Fig. 6. Across AUC, bit accuracy, and TPR@1%FPR, the performance of reserve and non-reserve closely follow each other across all token lengths. While small discrepancies appear at low token counts, they fall within natural sampling variability, and empirical results show no consistent performance gap between the two designs.

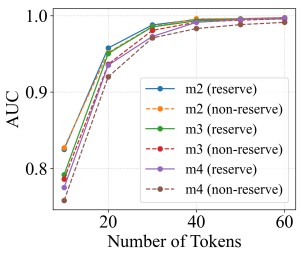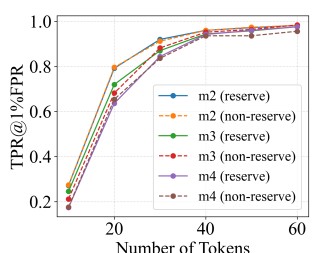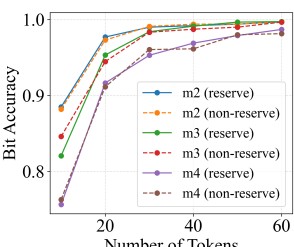

Figure 6: Performance impact of reserving a null symbol in tournament sampling based MirrorMark across varying number of $m$ and number of tokens, $H = 1$.

## F.2  PERFORMANCE COMPARISON OVER 200 AND 400 TOKENS

We present the performance comparison across different approaches over 200 and 400 tokens, respectively as in Table 4 and Table 4, where the watermarked text generated by each approach is embedded with 36 bits and 54 bits, respectively.

## F.3  AUC OF MIRRORMARK IN 72 BITS AND 90 BITS

Fig. 7 demonstrates the AUC of MirrorMark in 72 bits and 90 bits across varying number of tokens, respectively.

## F.4  ABLATION STUDY FOR CABS

To analyze parameter sensitivity, Tables 6, Table 7 and Table 8 report ablations over three CABS parameters without attack and under insertion, deletion, and substitution attacks, repectively. For each attack, we have the edit ratio $\epsilon \in \{0, 0.2, 0.4\}$, which represents the proportion of tokens that are modified in the text. Specifically, $\epsilon = 0$ means there is no attack. The results show clear quantitative trends.

Table 4: Mean perplexity and detectability for different approaches on 200 tokens. Each perplexity is given with a 90% confidence interval based on bootstrapping.

| Method | 36 Bits | | | | 54 Bits | | | |
|---|---|---|---|---|---|---|---|---|
| | AUC | TPR@1%FPR | Bit Acc. | Perplexity | AUC | TPR@1%FPR | Bit Acc. | Perplexity |
| Non Watermark | – | – | – | 7.7836 [7.6024, 7.9665] | – | – | – | 7.7836 [7.6024, 7.9665] |
| MPAC | 0.9903 | 0.9400 | 0.8893 | 9.8604 [9.6782, 10.0450] | 0.9913 | 0.9180 | 0.8394 | 10.1388 [9.9353, 10.3464] |
| RSBH | 0.9983 | **0.9980** | **0.9992** | 32.6466 [31.2956, 34.0539] | 0.9979 | **0.9980** | **0.9928** | 32.6994 [31.2430, 34.2013] |
| StealthInk | 0.9787 | 0.6540 | 0.8423 | **7.3038** [7.0626, 7.5421] | 0.9654 | 0.4420 | 0.7896 | **7.2339** [6.9976, 7.4662] |
| Gumbel-max | **0.9992** | 0.9920 | 0.9596 | 7.5709 [7.3951, 7.7503] | 0.9991 | 0.9960 | 0.9347 | 7.6708 [7.4902, 7.8644] |
| Tour-Wmean | 0.9955 | 0.9780 | 0.9191 | 7.7710 [7.6014, 7.9373] | **0.9999** | **0.9980** | 0.8657 | 7.7592 [7.5870, 7.9293] |
| Tour-Bayes | 0.9961 | 0.9800 | 0.9236 | 7.7710 [7.6014, 7.9373] | 0.9993 | 0.9780 | 0.8780 | 7.7592 [7.5870, 7.9293] |

Table 5: Mean perplexity and detectability for different approaches on 400 tokens. Each perplexity is given with a 90% confidence interval based on bootstrapping.

| Method | 36 Bits | | | | 54 Bits | | | |
|---|---|---|---|---|---|---|---|---|
| | AUC | TPR@1%FPR | Bit Acc. | Perplexity | AUC | TPR@1%FPR | Bit Acc. | Perplexity |
| Non Watermark | – | – | – | 7.0513 [6.9156, 7.1849] | – | – | – | 7.0513 [6.9156, 7.1849] |
| MPAC | 0.9970 | 0.9820 | 0.9599 | 8.8160 [8.6754, 8.9583] | 0.9960 | 0.9940 | 0.9227 | 8.8811 [8.7232, 9.0393] |
| RSBH | 0.9999 | **1.0** | **1.0** | 32.5108 [32.1111, 34.9533] | 0.9990 | **1.0** | **0.9972** | 33.6699 [32.1541, 35.2284] |
| StealthInk | 0.9941 | 0.9500 | 0.9204 | **6.5826** [6.4060, 6.7593] | 0.9952 | 0.9400 | 0.8748 | **6.5893** [6.4053, 6.7813] |
| Gumbel-max | **1.0** | **1.0** | 0.9912 | 6.8081 [6.6618, 6.9545] | 0.9998 | 0.9980 | 0.9831 | **6.8855** [6.7453, 7.0332] |
| Tour-Wmean | 0.9998 | 0.9960 | 0.9672 | 7.1759 [7.0406, 7.3120] | **1.0** | **1.0** | 0.9366 | 7.0888 [6.9534, 7.2243] |
| Tour-Bayes | 0.9999 | 0.9980 | 0.9667 | 7.1759 [7.0406, 7.3120] | 0.9999 | 0.9980 | 0.9529 | 7.0888 [6.9534, 7.2243] |

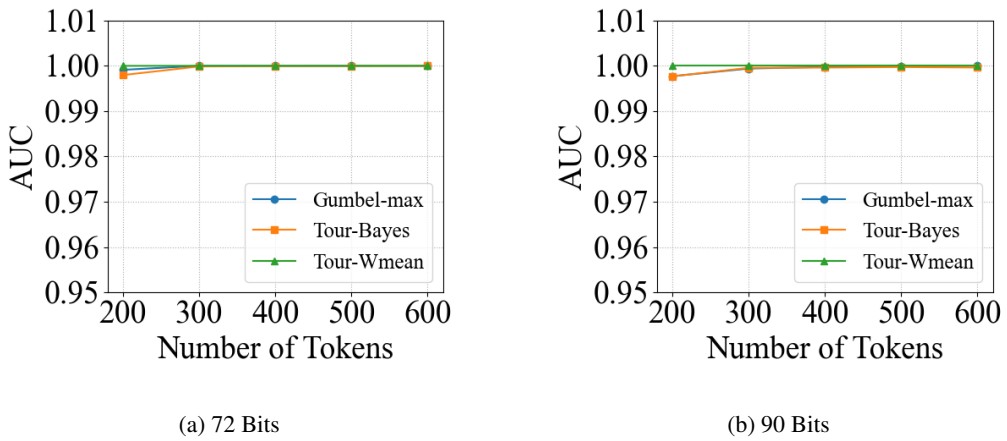

(a) 72 Bits                           (b) 90 Bits

Figure 7: AUC of MirrorMark across varying number of tokens respectively with 72 and 90 bits embedded.

$f = 3$ consistently provides the highest bit accuracy and strong TPR@1%FPR across insertion, deletion, and substitution, indicating that it offers the best trade-off between robustness and token coverage. $W = 4$ performs best or near-best for all edit ratios, capturing sufficient contextual information without overfitting to local perturbations. max_factor $= 1.5$ achieves the strongest robustness across edit rates, balancing frame-size flexibility and stability. Overall, the ablations demonstrate that the configuration used in the main paper ($f = 3$, $W = 4$, max_factor $= 1.5$) is empirically optimal among the tested settings. Specifically, Across these attacks, insertion primarily shifts tokens forward. Even at $\epsilon = 0.4$, MirrorMark maintains high detectability (AUC $= 0.999$, TPR@1%FPR $= 0.992$), with bit accuracy reduced to $0.790$, indicating that insertion mainly impairs bit recovery rather than WM/Non-WM separation. Deletion is the most adversarial, reducing available tokens. At $\epsilon = 0.4$, AUC remains above chance ($0.939$) and TPR@1%FPR reaches $0.604$. This degradation arises not only from the desynchronization of the token-to-position mapping but also from reduced detectability due to the smaller number of surviving tokens. Substitution preserves length and is the least destructive. At $\epsilon = 0.4$, MirrorMark sustains strong detectability (AUC $\approx 0.998$, TPR@1%FPR $\approx 0.992$) and relatively high bit accuracy (about $0.75$–$0.78$), confirming that CABS effectively absorbs localized perturbations.

### F.5 DETETABILITY COMPARISON OVER DIFFERENT $m$ FOR MIRRORMARK AFTER COPY-PASTE ATTACK

Table 9 compares the detectability of different approaches under copy-paste attacks, where each watermarked text contains 36 embedded bits within a 400-token sequence. Besides, Fig. 8 and Fig. 9 shows the detectability of MirrorMark against copy-paste attacks with 36 bits embedded in 400 tokens, where different $m$ are compared. $m \in \{2, 3, 4, 6\}$ is corresponding to $H \in \{18, 12, 9, 6\}$ respectively.

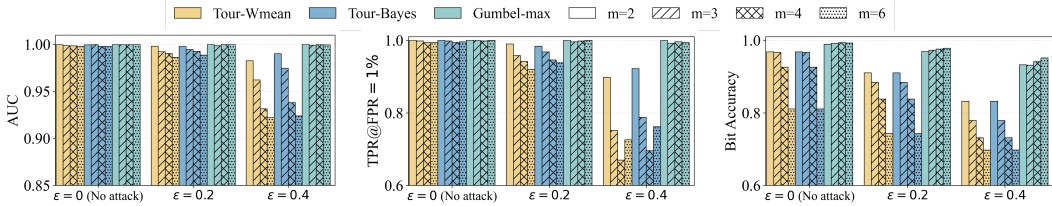

Figure 8: Detectability of MirrorMark against copy-paste attacks with 36 bits embedded in 400 tokens, where the edit fraction $\epsilon \in \{0, 0.2, 0.4\}$. To embed 36 bits, different $m$ applies for various number of positions $H$, i.e., $m \in \{2, 3, 4, 6\}$ is respectively corresponding to $H \in \{18, 12, 9, 6\}$.

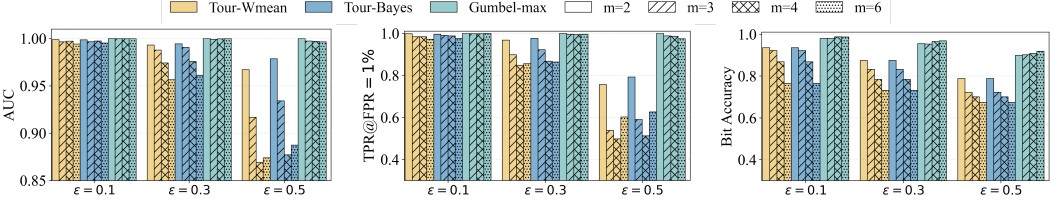

Figure 9: Detectability of MirrorMark against copy-paste attacks with 36 bits embedded in 400 tokens, where the edit fraction $\epsilon \in \{0.1, 0.3, 0.5\}$. To embed 36 bits, different $m$ applies for various number of positions, i.e., the number of positions for $m \in \{2, 3, 4, 6\}$ is respectively, 18, 12, 9, 6.

### F.6 THE EFFECT OF POSITION ALLOCATION SCHEDULER ON WATERMARKING SCHEMES

To disentangle the contributions of mod-1 mirroring and CABS, we perform an evaluation that systematically combines different position scheduler with different watermarking schemes. In particular, we incorporate the position schedulers used in MPAC (Yoo et al. (2024)) and RSBH (Qu et al. (2024)), which we denote as *NaiveHash* and *DPHash*, respectively. NaiveHash (MPAC, Section 3.2) seeds a PRF using the previous $h$ tokens to randomly select a position, whereas DPHash

| | Setting | AUC | TPR@1%FPR | Bit Accuracy |
|---|---|---|---|---|
| **Varying $f$ (with $W = 4$, max_factor=1.5)** | | | | |
| $\epsilon = 0.0$ | $f = 1$ | **1.000** | 0.998 | 0.939 |
| | $f = 2$ | **1.000** | 0.998 | 0.952 |
| | $f = 3$ | **1.000** | **1.000** | **0.985** |
| | $f = 4$ | **1.000** | 0.998 | 0.957 |
| $\epsilon = 0.2$ | $f = 1$ | 0.999 | 0.996 | 0.828 |
| | $f = 2$ | 0.999 | 0.996 | 0.838 |
| | $f = 3$ | **1.000** | **0.998** | **0.852** |
| | $f = 4$ | **1.000** | 0.996 | 0.847 |
| $\epsilon = 0.4$ | $f = 1$ | 0.998 | 0.984 | 0.772 |
| | $f = 2$ | **0.999** | 0.988 | 0.766 |
| | $f = 3$ | **0.999** | **0.992** | **0.790** |
| | $f = 4$ | **0.999** | **0.992** | 0.785 |
| **Varying $W$ (with $f = 3$, max_factor=1.5)** | | | | |
| $\epsilon = 0.0$ | $W = 1$ | **1.000** | 0.998 | 0.945 |
| | $W = 2$ | **1.000** | 0.998 | 0.943 |
| | $W = 3$ | **1.000** | 0.998 | 0.946 |
| | $W = 4$ | **1.000** | **1.000** | **0.985** |
| | $W = 5$ | **1.000** | **1.000** | 0.944 |
| $\epsilon = 0.2$ | $W = 1$ | 0.999 | 0.994 | 0.841 |
| | $W = 2$ | 0.999 | **0.998** | 0.843 |
| | $W = 3$ | **1.000** | **0.998** | 0.841 |
| | $W = 4$ | **1.000** | **0.998** | **0.852** |
| | $W = 5$ | **1.000** | 0.996 | 0.835 |
| $\epsilon = 0.4$ | $W = 1$ | 0.998 | 0.990 | 0.769 |
| | $W = 2$ | 0.999 | 0.992 | 0.770 |
| | $W = 3$ | **1.000** | **0.994** | 0.769 |
| | $W = 4$ | 0.999 | 0.992 | **0.790** |
| | $W = 5$ | 0.998 | 0.982 | 0.763 |
| **Varying max_factor (with $f = 3$, $W = 4$)** | | | | |
| $\epsilon = 0.0$ | max_factor $= 1.25$ | **1.000** | **1.000** | 0.948 |
| | max_factor $= 1.50$ | **1.000** | **1.000** | **0.985** |
| | max_factor $= 2.00$ | **1.000** | **1.000** | 0.955 |
| $\epsilon = 0.2$ | max_factor $= 1.25$ | **1.000** | 0.996 | 0.850 |
| | max_factor $= 1.50$ | **1.000** | 0.998 | **0.852** |
| | max_factor $= 2.00$ | **1.000** | **1.000** | 0.839 |
| $\epsilon = 0.4$ | max_factor $= 1.25$ | **0.999** | 0.988 | 0.768 |
| | max_factor $= 1.50$ | **0.999** | **0.992** | **0.790** |
| | max_factor $= 2.00$ | 0.998 | 0.990 | 0.778 |

Table 6: Robustness of MirrorMark under insertion attacks with different CABS parameters, where $m = 2$, $H = 12$, and the number of tokens is 300.

| Setting | | AUC | TPR@1%FPR | Bit Accuracy |
|---|---|---|---|---|
| **Varying $f$ (with $W = 4$, max_factor=1.5)** | | | | |
| $\epsilon = 0.0$ | $f = 1$ | **1.000** | 0.998 | 0.939 |
| | $f = 2$ | **1.000** | 0.998 | 0.952 |
| | $f = 3$ | **1.000** | **1.000** | **0.985** |
| | $f = 4$ | **1.000** | 0.998 | 0.957 |
| $\epsilon = 0.2$ | $f = 1$ | 0.998 | **0.988** | 0.683 |
| | $f = 2$ | 0.997 | 0.984 | **0.702** |
| | $f = 3$ | **0.999** | **0.988** | 0.700 |
| | $f = 4$ | 0.998 | 0.982 | 0.700 |
| $\epsilon = 0.4$ | $f = 1$ | 0.939 | **0.604** | 0.464 |
| | $f = 2$ | 0.942 | 0.566 | 0.471 |
| | $f = 3$ | **0.946** | 0.566 | 0.472 |
| | $f = 4$ | 0.945 | 0.584 | **0.474** |
| **Varying $W$ (with $f = 3$, max_factor=1.5)** | | | | |
| $\epsilon = 0.0$ | $W = 1$ | **1.000** | 0.998 | 0.945 |
| | $W = 2$ | **1.000** | 0.998 | 0.943 |
| | $W = 3$ | **1.000** | 0.998 | 0.946 |
| | $W = 4$ | **1.000** | **1.000** | **0.985** |
| | $W = 5$ | **1.000** | **1.000** | 0.944 |
| $\epsilon = 0.2$ | $W = 1$ | 0.998 | 0.980 | 0.686 |
| | $W = 2$ | 0.999 | **0.990** | 0.689 |
| | $W = 3$ | **1.000** | 0.986 | **0.707** |
| | $W = 4$ | 0.999 | 0.988 | 0.700 |
| | $W = 5$ | 0.999 | 0.988 | 0.700 |
| $\epsilon = 0.4$ | $W = 1$ | **0.956** | 0.580 | 0.476 |
| | $W = 2$ | 0.948 | 0.564 | 0.465 |
| | $W = 3$ | 0.947 | **0.600** | **0.481** |
| | $W = 4$ | 0.946 | 0.566 | 0.472 |
| | $W = 5$ | 0.949 | 0.592 | 0.453 |
| **Varying max_factor (with $f = 3$, $W = 4$)** | | | | |
| $\epsilon = 0.0$ | max_factor = 1.25 | **1.000** | **1.000** | 0.948 |
| | max_factor = 1.50 | **1.000** | **1.000** | **0.985** |
| | max_factor = 2.00 | **1.000** | **1.000** | 0.955 |
| $\epsilon = 0.2$ | max_factor = 1.25 | **0.999** | 0.986 | 0.682 |
| | max_factor = 1.50 | **0.999** | 0.988 | **0.700** |
| | max_factor = 2.00 | **0.999** | **0.990** | 0.693 |
| $\epsilon = 0.4$ | max_factor = 1.25 | **0.954** | **0.606** | 0.464 |
| | max_factor = 1.50 | 0.946 | 0.566 | **0.472** |
| | max_factor = 2.00 | 0.950 | 0.558 | 0.464 |

Table 7: Robustness of MirrorMark under deletion attacks with different CABS parameters, where $m = 2$, $H = 12$, and the number of tokens is 300.

| | Setting | AUC | TPR@1%FPR | Bit Accuracy |
|---|---|---|---|---|
| **Varying $f$ (with $W = 4$, max_factor=1.5)** | | | | |
| $\epsilon = 0.0$ | $f = 1$ | **1.000** | 0.998 | 0.939 |
| | $f = 2$ | **1.000** | 0.998 | 0.952 |
| | $f = 3$ | **1.000** | **1.000** | **0.985** |
| | $f = 4$ | **1.000** | 0.998 | 0.957 |
| $\epsilon = 0.2$ | $f = 1$ | **0.998** | 0.970 | 0.684 |
| | $f = 2$ | 0.997 | **0.984** | 0.710 |
| | $f = 3$ | **0.998** | **0.984** | **0.723** |
| | $f = 4$ | 0.997 | **0.984** | 0.709 |
| $\epsilon = 0.4$ | $f = 1$ | **0.960** | 0.646 | 0.492 |
| | $f = 2$ | 0.945 | 0.644 | **0.504** |
| | $f = 3$ | 0.948 | **0.648** | 0.499 |
| | $f = 4$ | 0.957 | 0.622 | 0.486 |
| **Varying $W$ (with $f = 3$, max_factor=1.5)** | | | | |
| $\epsilon = 0.0$ | $W = 1$ | **1.000** | 0.998 | 0.945 |
| | $W = 2$ | **1.000** | 0.998 | 0.943 |
| | $W = 3$ | **1.000** | 0.998 | 0.946 |
| | $W = 4$ | **1.000** | **1.000** | **0.985** |
| | $W = 5$ | **1.000** | **1.000** | 0.944 |
| $\epsilon = 0.2$ | $W = 1$ | 0.998 | 0.968 | 0.703 |
| | $W = 2$ | **0.999** | **0.990** | 0.709 |
| | $W = 3$ | 0.997 | 0.984 | 0.709 |
| | $W = 4$ | 0.998 | 0.984 | **0.723** |
| | $W = 5$ | 0.997 | 0.984 | 0.690 |
| $\epsilon = 0.4$ | $W = 1$ | 0.942 | 0.638 | **0.503** |
| | $W = 2$ | 0.933 | 0.574 | 0.494 |
| | $W = 3$ | **0.951** | 0.632 | 0.490 |
| | $W = 4$ | 0.948 | **0.648** | 0.499 |
| | $W = 5$ | 0.947 | 0.590 | 0.489 |
| **Varying max_factor (with $f = 3$, $W = 4$)** | | | | |
| $\epsilon = 0.0$ | max_factor $= 1.25$ | **1.000** | **1.000** | 0.948 |
| | max_factor $= 1.50$ | **1.000** | **1.000** | **0.985** |
| | max_factor $= 2.00$ | **1.000** | **1.000** | 0.955 |
| $\epsilon = 0.2$ | max_factor $= 1.25$ | **0.998** | 0.978 | 0.708 |
| | max_factor $= 1.50$ | **0.998** | **0.984** | **0.723** |
| | max_factor $= 2.00$ | **0.998** | 0.982 | 0.709 |
| $\epsilon = 0.4$ | max_factor $= 1.25$ | 0.949 | 0.610 | 0.486 |
| | max_factor $= 1.50$ | 0.948 | **0.648** | **0.499** |
| | max_factor $= 2.00$ | **0.951** | 0.624 | 0.482 |

Table 8: Robustness of MirrorMark under substitution attacks with different CABS parameters, where $m = 2$, $H = 12$, and the number of tokens is 300.

Table 9: Detectability for different approaches on 400 tokens with 36 bits embedded after copy-paste attack, where the edit fraction $\epsilon \in \{0.1, 0.3, 0.5\}$.

| Method | $\epsilon = 0.1$ | | | $\epsilon = 0.3$ | | | $\epsilon = 0.5$ | | |
|---|---|---|---|---|---|---|---|---|---|
| | AUC | TPR@1%FPR | Bit Acc. | AUC | TPR@1%FPR | Bit Acc. | AUC | TPR@1%FPR | Bit Acc. |
| MPAC | 0.9847 | 0.9025 | 0.9263 | 0.9729 | 0.8650 | 0.8725 | 0.9290 | 0.6075 | 0.7959 |
| RSBH | 0.9840 | 0.4275 | 0.6156 | 0.9386 | 0.0150 | 0.6181 | 0.7243 | 0.01 | 0.5825 |
| StealthInk | 0.9901 | 0.9100 | 0.8870 | 0.9636 | 0.6675 | 0.8213 | 0.8374 | 0.2575 | 0.7419 |
| Tour-Wmean | 0.9989 | 0.9980 | 0.9357 | 0.9932 | 0.9680 | 0.8750 | 0.9671 | 0.7560 | 0.7880 |
| Tour-Bayes | 0.9987 | 0.9960 | 0.9357 | 0.9944 | 0.9760 | 0.8750 | 0.9787 | 0.7920 | 0.7880 |
| Gumbel-max | **1.0** | **1.0** | **0.9801** | **1.0** | **1.0** | **0.9549** | **1.0** | **1.0** | **0.8986** |

(RSBH, Section 4.2) constructs a balanced token-to-segment mapping through a secret-key shuffle followed by a dynamic programming procedure.

Because the DPHash table released in the official implementation of Qu et al. (2024) is constructed with $h = 1$, we evaluate performance under this setting in Fig. 11. Additionally, since our main experiments use $h = 4$ by default unless otherwise noted, we also report results under $h = 4$ in Fig. 12. For both setting, we show the Gini score[5] in Fig. 10, which quantifies how balanced the token allocation is across positions. The lower Gini score represents more balanced allocation. We observe that CABS consistently shows significant low Gini score that is near to 0, and outperforms NaiveHash and DPHash on MirrorMark (both Tour-Bayes and Gumbel-max) across all detectability metrics. In contrast, for MPAC, the AUC and TPR@1%FPR of CABS are comparable to those of NaiveHash and DPHash, while the bit accuracy of CABS is only slightly higher. This is expected because balanced allocation helps decode messages more reliably. In contrast, positions receive few or no tokens can provide only weak or random evidence. CABS is designed precisely to mitigate such imbalance by distributing tokens more evenly across positions.

The difference between MirrorMark and MPAC arises from how each method uses positional evidence. MirrorMark aggregates evidence from all positions, making it highly sensitive to positional imbalance. For example, consider a text with 100 tokens distributed across four positions as 85–5–5–5. In watermark text, the position with 85 tokens provides a strong signal for the correct message, but the remaining three positions—with only five tokens each—contribute mostly noise. When combined in the final score, this noisy evidence dilutes the strong signal, making watermark and non-watermark score distributions more difficult to separate.

MPAC, however, is robust under the same allocation. The 85 tokens in the dominant position overwhelmingly vote for the correct message in watermark text, while non-watermark text remains roughly balanced across message candidates. Because MPAC keeps only the largest vote per position and sums these maxima, lightly populated positions add very little and do not introduce harmful noise. Consequently, the detectability of MPAC remains stable even under highly uneven token allocation.

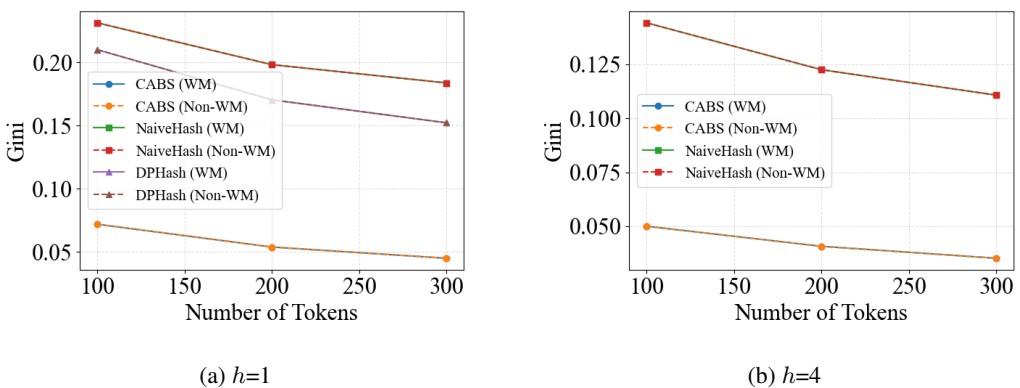

(a) $h=1$            (b) $h=4$

Figure 10: Comparison of token-allocation balance between different position scheduler. The setting is $m = 2$, $H = 12$, and the number of tokens is 300. The Gini coefficient is significantly lower (more balanced allocation) when using CABS, showing that CABS reduces position-allocation skew and improves uniformity.

### F.7 REPEATITION SCORE AND LLM-AS-JUDGE SCORE OF THE TEXT GENERATED WITH WATERMARKING SCHEME

In addition to the perplexity results reported in Table 4, Table 1, and Table 5, we further evaluate the linguistic quality of MirrorMark using two complementary metrics: (1) an LLM-as-a-judge assessment with GPT-4o (Fig. 13), and (2) a repetition-based analysis using distinct-2 and repetition rate (Table 10).

---

[5]https://en.wikipedia.org/wiki/Gini_coefficient

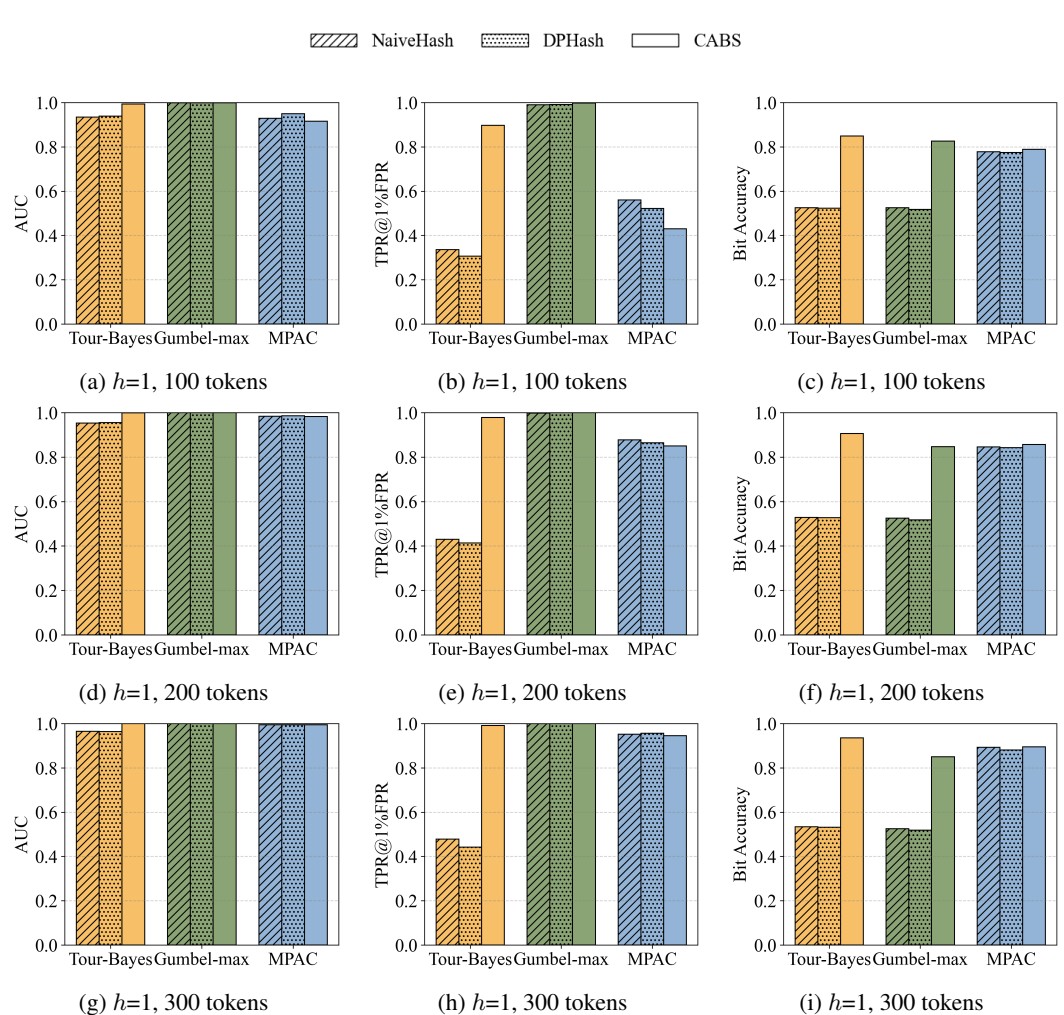

Figure 11: Detectability of MirrorMark with length of n-gram $h$=1. The setting is $m = 2$, $H = 12$, and the number of tokens is 300.

Table 10: Text quality scored with distinct-2 and repetition rate across watermarking schemes, 36 bits are embedded in 300 tokens.

| | Non-watermarked | MPAC | RSBH | StealthInk | TB (m=2) | TB (m=3) | TB (m=4) | TB (m=6) | G-max (m=2) | G-max (m=3) | G-max (m=4) | G-max (m=6) |
|---|---|---|---|---|---|---|---|---|---|---|---|---|
| Distinct-2 | 0.9471 | 0.9624 | 0.9648 | 0.9498 | 0.9452 | 0.9494 | 0.9475 | 0.9451 | 0.9277 | 0.9269 | 0.9209 | 0.9292 |
| Repetition Rate | 0.4542 | 0.4183 | 0.3528 | 0.4410 | 0.4538 | 0.4504 | 0.4509 | 0.4561 | 0.4733 | 0.4761 | 0.4849 | 0.4752 |

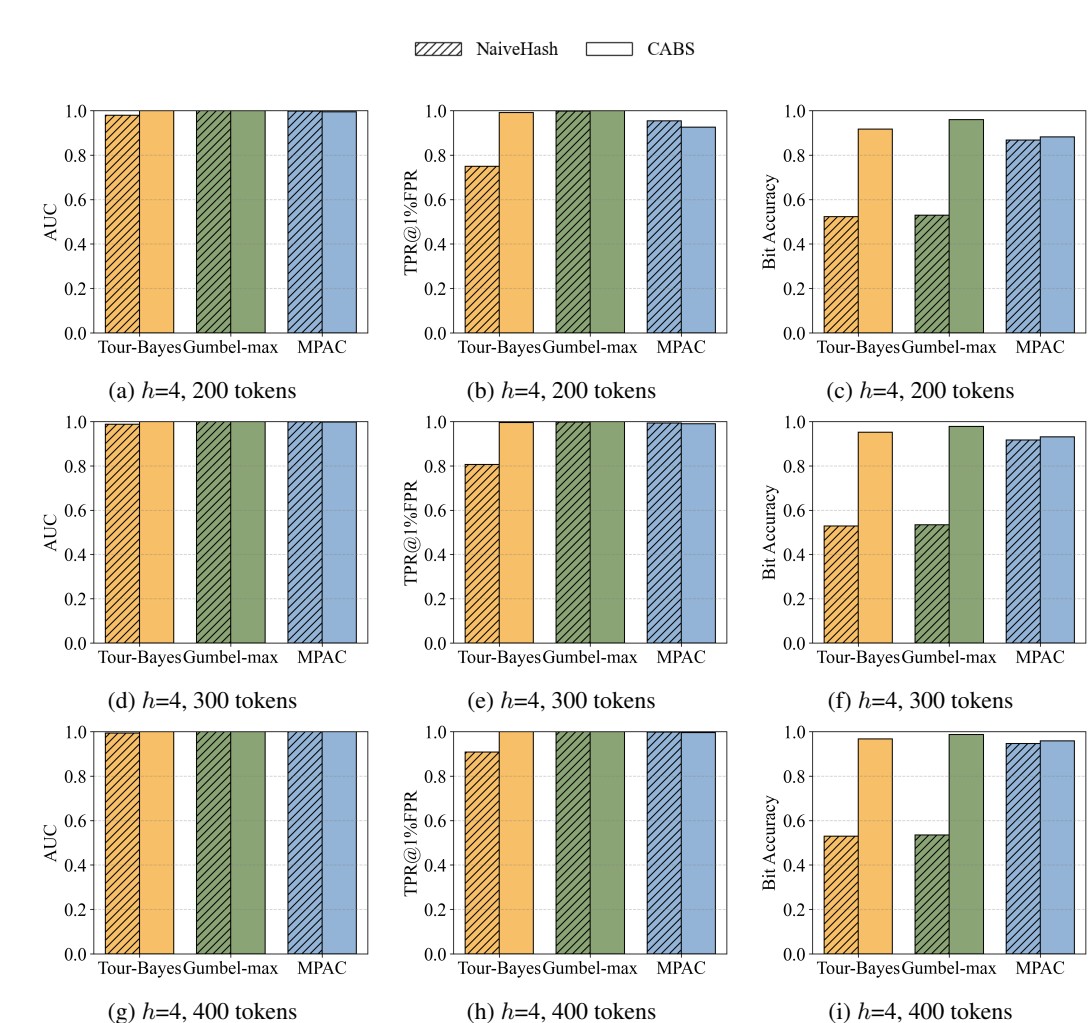

Figure 12: Detectability of MirrorMark across $h$=4. The setting is $m = 2$, $H = 12$, and the number of tokens is 300.

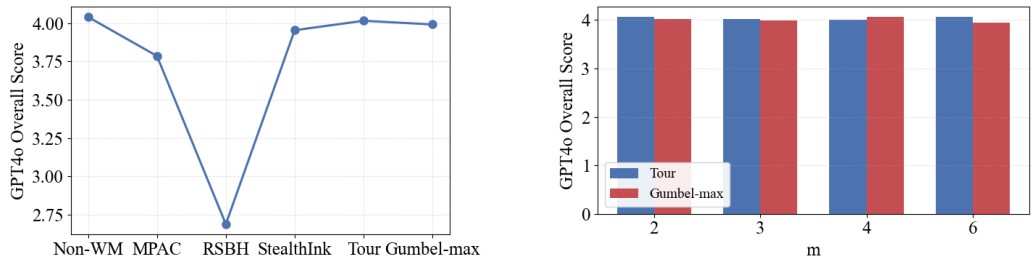

Figure 13: Text quality scored by GPT4o over 300 tokens, where $m$=3 and $H$=12

For the LLM-as-a-judge study, GPT-4o scored each text along four dimensions: coherence, clarity, naturalness, and overall quality. Following Jovanović et al. (2024), we design the following GPT4o Judge prompt explicitly to ignore truncation effects and focus solely on linguistic fluency.

---

**GPT-4o Judge Prompt**

You are an impartial expert evaluator of linguistic text quality.
The given text is a continuation generated from a truncated C4 sample (15–20 words). The text may start or end abruptly because the generation length is fixed (e.g., 300 tokens). Do **NOT** penalize truncation or incompleteness.
Evaluate **ONLY** linguistic quality:

- Coherence — logical flow of ideas

- Clarity — easy to understand

- Naturalness — how fluent / human-written the text appears

Rate each from 1 to 5. Compute "overall" as the average of the three.
Return only a JSON object in exactly the following structure:

```
{
    "coherence": float,
    "clarity": float,
    "naturalness": float,
    "overall": float
}
```

Text: <<<TEXT>>>

---

Across all configurations, MirrorMark achieves GPT-4o scores that are statistically indistinguishable from the non-watermarked baseline. The overall score difference consistently stays within 0.05–0.10, well inside the natural variance of GPT-4o evaluations. These results confirm that mod-1 mirroring does not degrade linguistic quality, aligning with our theoretical guarantee that Mirror-Mark is distribution-preserving. In contrast, distortion-based baselines such as MPAC and RSBH exhibit noticeably lower GPT-4o scores, consistent with their higher perplexity and the known side effects of their logit-biasing mechanisms.

The diversity analysis further reinforces these findings. Although MPAC and RSBH report high distinct-2 and low repetition rates, this behavior is driven by artificially skewing the token distribution away from natural language usage, which corresponds to their lower GPT-4o scores. In comparison, MirrorMark, especially the tournament-sampling variant, achieves distinct-2 and repetition rates nearly identical to non-watermarked text, demonstrating that it preserves natural linguistic diversity. While Gumbel-max is inherently more deterministic under top-$k$ sampling and thus yields slightly lower diversity, GPT-4o evaluations confirm that this does not harm fluency or naturalness, as the generated sentences remain coherent and well-structured.

## F.8 COMPARISON WITH MULTI-KEY BASED MULTI-BIT WATERMARKING FERNANDEZ ET AL. (2023)

We further compare our Gumbel-max based MirrorMark with the naive multi-key multi-bit extension proposed by Fernandez et al. (2023), which also builds upon Gumbel-max zero-bit watermarking. As shown in Fig. 14, the multi-bit watermarking algorithm of Fernandez et al. consistently underperforms MirrorMark across all message lengths. In the multi-key design, when the decoder assumes an incorrect message, it reconstructs an entirely independent PRF sequence, and the resulting score behaves indistinguishably from non-watermarked text. Because the wrong-message hypotheses receive no penalty, the separation between messages remains limited.

In contrast, MirrorMark employs a single PRF and introduces message-dependent mirroring. When the decoder assumes an incorrect message, it evaluates the correlation using an incorrect mirroring center, which systematically reduces the score and effectively imposes a penalty on wrong hypotheses. This mechanism, absent in the multi-key approach, creates markedly larger score gaps between correct and incorrect messages, yielding substantially higher bit accuracy. Finally, both Fig. 14 and

Fig. 15 confirm the expected trend under MirrorMark, where the zero-bit ($m = 0$) setting yields the strongest signal, and performance degrades smoothly as $m$ increases.

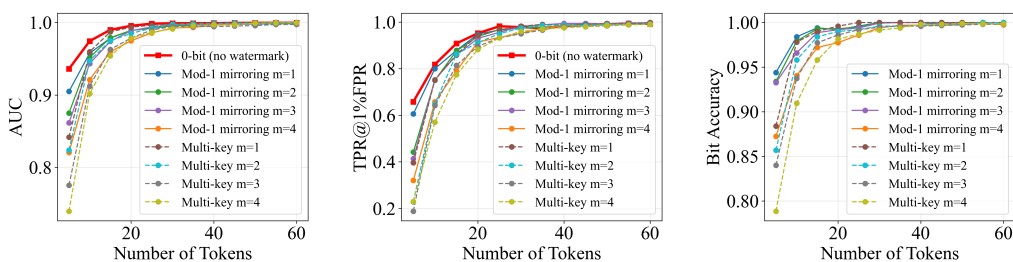

Figure 14: Performance comparison between Gumbel-max based MirrorMark and multi-key based multi-bit watermarking, $H = 1$.

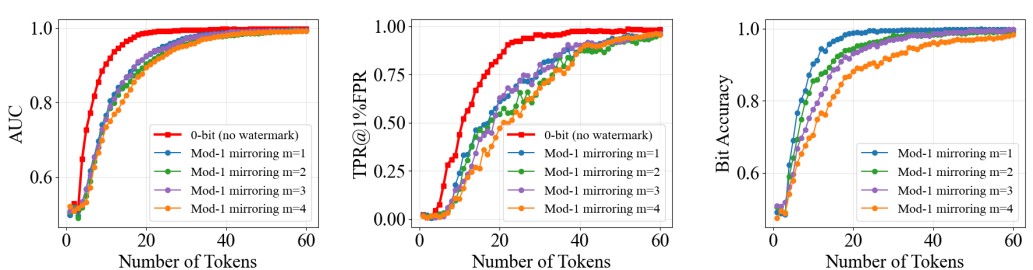

Figure 15: Performance comparison between zero-bit tournament sampling based watermark and tournament sampling based MirrorMark (Tour-Bayes), $m = 1$.

## F.9  CROSS-LANGUAGE ADAPTATION

To evaluate whether MirrorMark is tied to a specific language or can be reliably applied across languages, we conduct a cross-language experiment using the multilingual XL-Sum dataset Hasan et al. (2021) on Gemma-7B-it Team et al. (2024). For each language (English, Chinese, Russian), we sample summaries from XL-Sum and prompt the model to generate full news articles in the corresponding language. During generation, we apply exactly the same MirrorMark watermarking rule as in our main experiments. For each language, we generate 500 paired watermarked and non-watermarked samples of length 200 tokens, and then evaluate both the Bayesian detector for tournament sampling (Tour-Bayes) and the analytic detector for Gumbel-max.

Fig. 16 shows that a threshold $\tau$ calibrated in one language does not perfectly transfer to another: when a threshold learned on English is applied to Chinese, the empirical FPR on Chinese increases, whereas the same threshold applied to Russian remains largely unchanged; conversely, a threshold learned on Chinese becomes overly conservative when applied to English or Russian. This behavior aligns with an important empirical fact documented in prior work Montemurro & Zanette (2011): Chinese text consistently exhibits lower next-token entropy than English and Russian, while English and Russian have similar entropy profiles. Fig. 17 and Fig. 18 further support this explanation— Chinese WM and NWM score distributions are shifted to the right compared to English and Russian, although the separation between the two hypotheses remains similar. As a result, a threshold $\tau$ calibrated on English (where the NWM distribution is farther left) becomes slightly too permissive for Chinese, increasing FPR, whereas a threshold $\tau$ calibrated on Chinese becomes too strict for English and Russian, lowering both FPR and TPR. Thus, the cross-language FPR drift observed in Fig. 16 is fully explained by entropy-driven shifts in score distributions, rather than by any language dependence of the watermarking method itself.

Overall, Fig. 16, Fig. 17, and Fig. 18 demonstrate that MirrorMark is not tied to English or any particular dataset. Both the tournament-sampling (Tour-Bayes) and Gumbel-max variants show similar ROC curves and clearly separated score distributions across all three languages. The only differences are small score-scale shifts arising from language-specific model entropy, which can be handled with simple threshold recalibration. This supports our claim that MirrorMark is a data-agnostic generative watermark whose detectability depends primarily on sequence length and entropy, rather than on the specific language or domain.

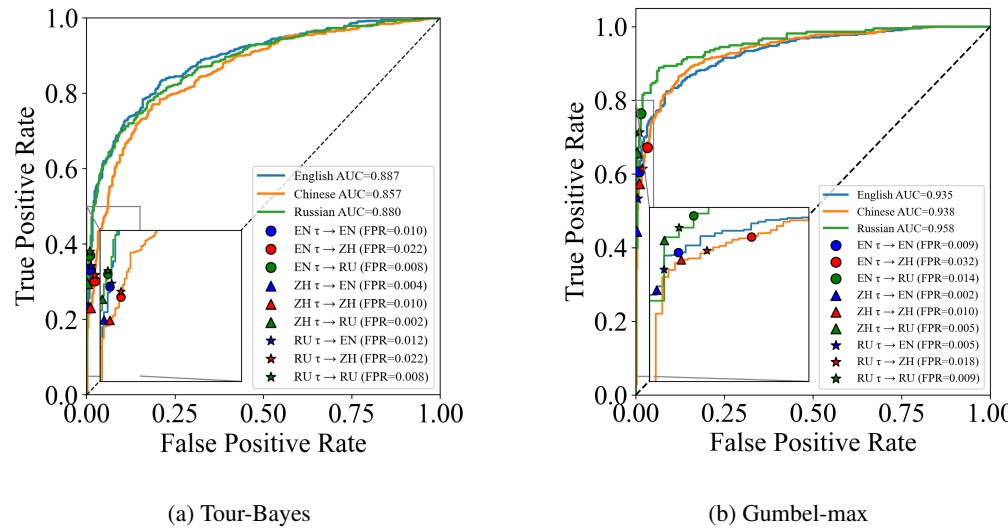

(a) Tour-Bayes

(b) Gumbel-max

Figure 16: ROC across three languages over 200 tokens, where $m$=3 and the number of positions $H$=12

### F.10 DETECTABILITY ON INSTRUCTION TASK

Fig. 19 reports the detectability of MirrorMark on instruction task, where 500 randomly selected ELI5 prompts Fan et al. (2019) are evaluated with Gemma-7B-it model Team et al. (2024). In particular, we apply the default setting stated in Section D. Compared with the completion results on C4 in Fig. 14 where even at 60 tokens the TPR@1%FPR already exceeds 95% for both m = 1 and m = 3, the detection performance on instruction tasks is weaker. For example, In Fig. 19 (b), the Gumbel-max–based MirrorMark reaches about 80% TPR@1%FPR at 100 tokens, while in Fig. 19 (e), the tournament-sampling–based MirrorMark reaches about 65% TPR@1%FPR at 100 tokens. This gap is consistent with the nature of instruction-following generation because the instruction-tuned model interpreting a structured prompt leads to more deterministic token selection, reducing the effective randomness available for watermark perturbation, which further limits the magnitude of step-wise perturbations in $u$ values, making it harder to accumulate statistical evidence compared to the text completion task.

To further analyze the effect of the $u$ value distribution on tournament sampling, we additionally evaluate a variant where u is drawn from a Bernoulli distribution instead of the $\mathrm{Uniform}(0,1)$. As shown in Fig. 19(k), at 100 tokens, the TPR@1%FPR improves by around 5 percentage points. The reason is that Bernoulli sampling produces the maximum possible diversity, since $u$ value takes only values in $\{0,1\}$. This effectively pushes scores all the way from 0 to 1 to make the watermark signal easier to detect, instead of, for example, fluctuating only from 0.4 to 0.6. However, because Bernoulli $u$ values are inherently binary, the scheme can embed at most 1 bit, limiting its applicability to multi-bit settings.

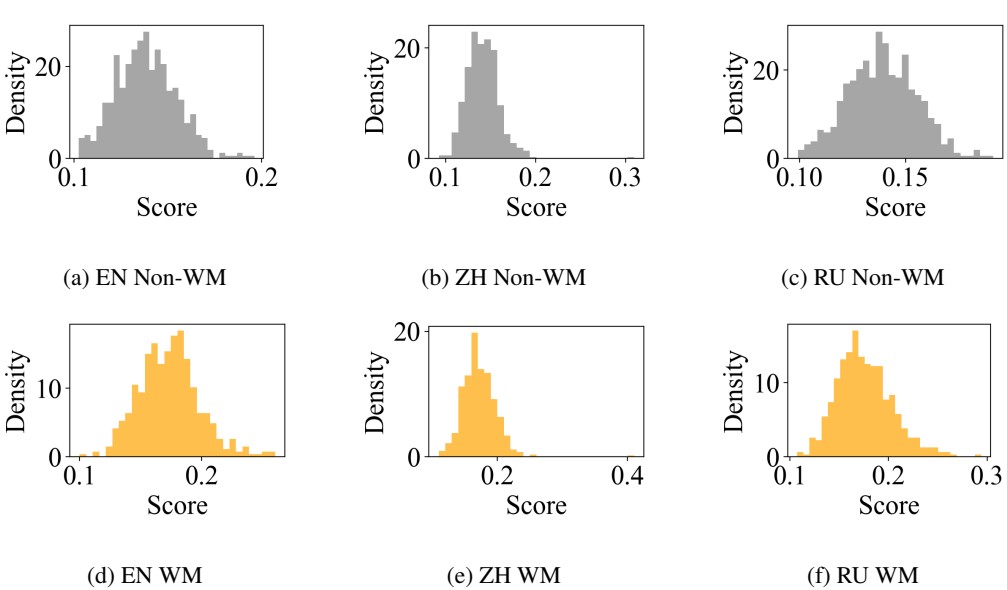

Figure 17: Score distributions with Tour-Bayes for English (EN), Chinese (ZH), and Russian (RU) under non-watermarked (top row) and watermarked (bottom row) text at 200 tokens, and the message with $m$=3 and $H$=12 is embedded in each watermarked sample.

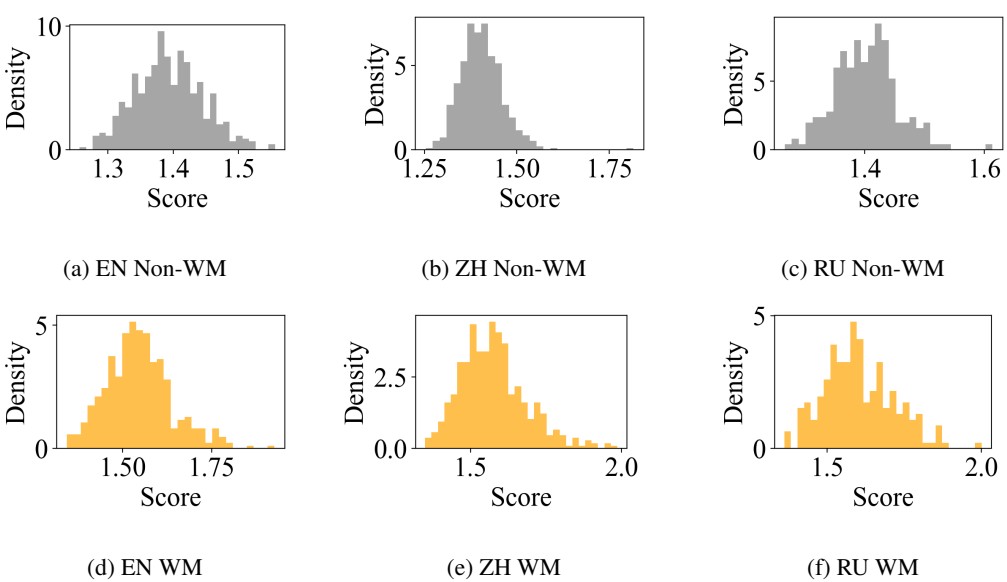

Figure 18: Score distributions with Gumbel-max for English (EN), Chinese (ZH), and Russian (RU) under non-watermarked (top row) and watermarked (bottom row) text at 200 tokens, and the message with $m$=3 and $H$=12 is embedded in each watermarked sample.

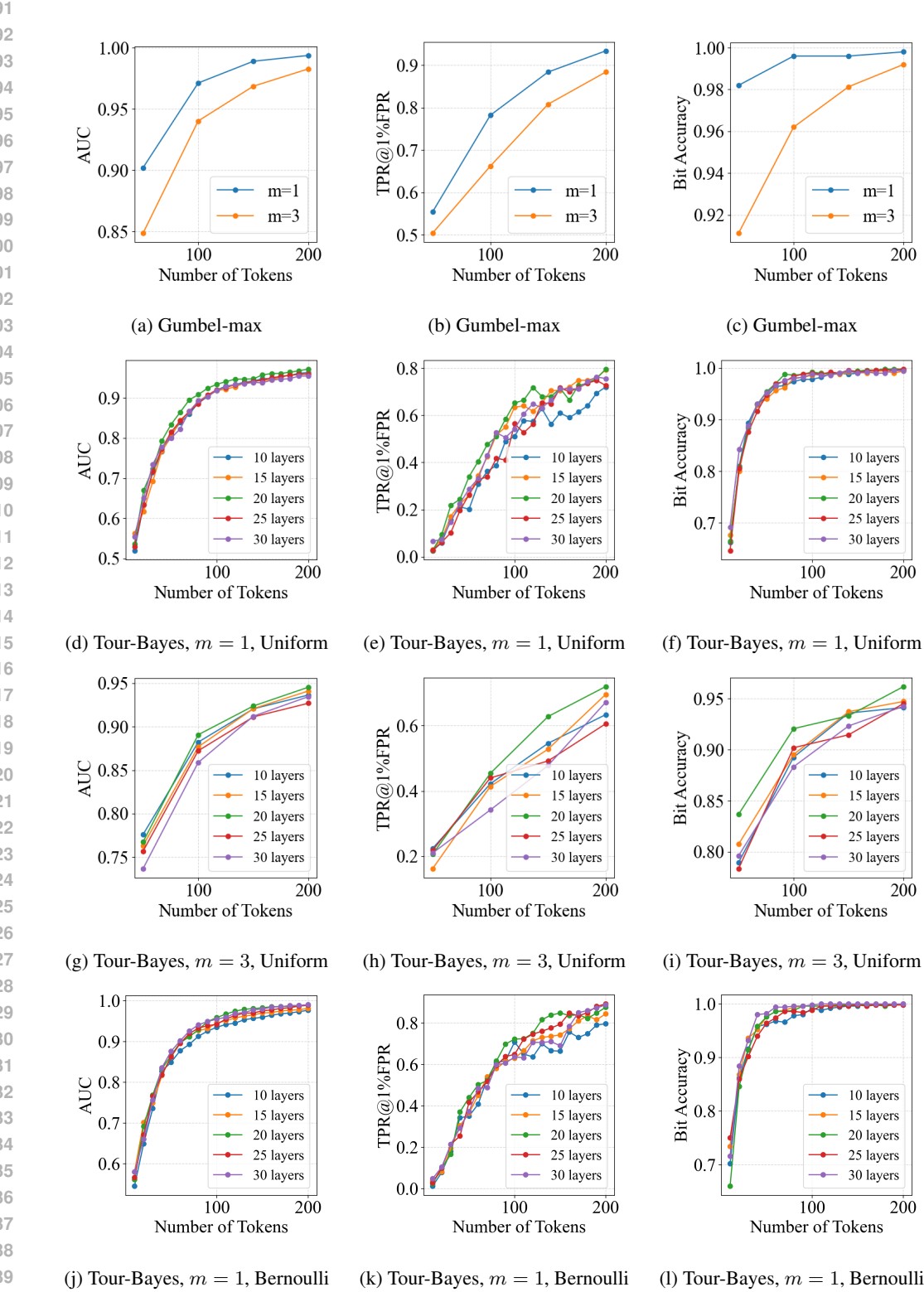

Figure 19: Detectability of MirrorMark on Gemma-7B-it and ELI5 prompts, with watermark of $m \in \{1, 3\}$ and $H = 1$ embedded in each response.

