# OpenReview forum: "MIRRORMARK: A Distortion-Free Multi-Bit Watermark for Large Language Models"
_ICLR.cc/2026/Conference — ICLR 2026 Conference Desk Rejected Submission_

### Official Review · Reviewer_9XrG · 2025-10-21

**Soundness:** 2
**Presentation:** 1
**Contribution:** 2
**Rating:** 4
**Confidence:** 4

**Summary:**

This paper proposes MirrorMark, a multi-bit watermarking framework for LLMs that preserves the original token probability distribution through a mod-1 mirroring transformation. The method extends existing distortion-free zero-bit watermarking schemes (Gumbel sampling and tournament sampling) to embed multi-bit messages by reflecting pseudorandom values about message-dependent mirror points. To enhance robustness against token insertions and deletions, the authors introduce a Content-Anchored Balanced Scheduler that distributes tokens across message positions in a balanced manner. The paper provides theoretical analysis relating detection error rates to the number of PRF draws and demonstrates strong detection performance while maintaining text quality comparable to non-watermarked outputs.

**Strengths:**

- The paper provides rigorous mathematical guarantees that the mirroring transformation Ψ(u; ψ_M) = (2ψ_M - u) mod 1 is measure-preserving, ensuring that the watermarked text follows the same probability distribution as the original LLM output.

- The experimental results demonstrate that MirrorMark achieves superior detectability compared to existing multi-bit baselines.

- The CABS algorithm ensures sufficient tokens are assigned to each message position for reliable decoding.

**Weaknesses:**

- The core distinguishing feature of watermarking is robustness against modifications, whereas steganography prioritizes undetectability. MirrorMark's preserving the exact probability distribution through measure-preserving transformations aligns more naturally with steganographic goals.

- While the paper claims robustness through statistical accumulation and CABS redundancy, the actual resilience is severely limited. The robustness demonstrated against copy-paste attacks is somewhat trivial, this attack only dilutes the watermark signal but does not actively corrupt the token-to-position mapping that MirrorMark critically depends on.

- For paraphrasing attacks, bit accuracy drops to 54%, rendering the multi-bit message unrecoverable. While AUC remains relatively high, this only indicates binary watermark detection, not multi-bit decoding.

- Although CABS uses content-based anchors to define frames, even minor edits can desynchronize the token-to-position mapping. The paper does not provide quantitative evaluation of robustness under realistic editing scenarios (e.g., content insertion/deletion).

- Both embedding and extraction rely on perfectly synchronized CABS scheduling. Any modification to the text breaks this synchronization. This creates a critical single point of failure: the multi-bit message is irrecoverably lost if the attacker rewrites even a few sentences, because the detector can no longer determine which tokens carry which message symbols.

- The paper suffers from excessive mathematical formalism that obscures rather than illuminates the core contributions. The authors should prioritize clarity and intuition over mathematical sophistication to make the work accessible to the broader community working on LLM watermarking.

**Questions:**

- Could you provide quantitative evaluation under more realistic editing scenarios?

- Have you explored a "soft" version of MirrorMark that applies partial mirroring (e.g., weighted combination of original and mirrored u values) to achieve a middle ground?

- Could you provide a use case where the recovered multi-bit message (with 54% accuracy) is still useful?

- The paper does not include ablation studies on these hyperparameters. Do they require careful tuning for different text lengths or domains?

- What is the actual wall-clock time for encoding and decoding compared to non-watermarked generation?

---

> ### Author Response · Authors · 2025-11-25
>
> # **Ablation study of CABS and robustness of MirrorMark**
>
> ## **Q: The paper does not include ablation studies on these hyperparameters.**
> We thank the reviewer for highlighting this important point. To address it comprehensively, we expanded our experiments to include a full ablation study examining
> (i) how each CABS parameter influences robustness, and
> (ii) how these choices interact with insertion, deletion, and substitution attacks.
>
> Our default configuration $f=3$, $W=4$, $max\\_factor=1.5$ is now explicitly documented in Appendix E of the revised manuscript. For clarity, we restate the roles of each parameter and refer the reviewer to Fig. 2 in the revised manuscript (which is also [provided in this link for easy access] (https://anonymous.4open.science/api/repo/MirrorMark-abcdefg/file/Fig1-2.pdf?v=c1d11c7e)) for an intuitive illustration:
>
> - The parameter $f$ determines how often a new frame can start: a new frame is opened only when the last $f$ bits of the content hash of the current context are all zeros. For example, when $f=3$, the last three bits must all be zero, which occurs with probability $1/8$ under a near-uniform hash. Thus, larger $f$ leads to fewer and longer frames, while smaller $f$ results in more frequent frame boundaries.
>
> - The parameter $W$ specifies the context size used to compute the content hash to determine whether to anchor a frame; that is, each hash is computed over the most recent $W$ tokens.
>
> - The parameter $max\\_factor$ determines the maximum allowable frame length. For example, when $max\\_factor=1.5$, the maximum frame length is set to $max\\_len=1.5H$, where $H$ denotes the number of positions.
>
> **Impact of CABS parameters.**
>
> We added Tables 6–8 in appendix F.4 of the revised manuscript to report ablations over frame anchor size $f$, window size $W$, and $max\\_factor$ across three types of edit attacks, and [this is the link to Tables 6–8 for convenient reference] (https://anonymous.4open.science/api/repo/MirrorMark-abcdefg/file/additional_experiments/Table6-8.pdf?v=7482f86f).
>
> We observe clear trends emerge across all settings:
>
> - $f=3$ consistently provides the highest bit accuracy and strong TPR@1%FPR across insertion, deletion, and substitution, indicating that it offers the best trade-off between robustness and token coverage.
>
> - $W=4$ performs best or near-best for all edit ratios, capturing sufficient contextual information without overfitting to local perturbations.
>
> - $max\\_factor=1.5$ achieves the strongest robustness across edit rates, balancing frame-size flexibility and stability.
>
> ---
>
>
> ## **Q: Do these hyperparameters require careful tuning for different text lengths or domains?**
>
> **Impact of text length.**
>
> Across all tested lengths (200, 300, 400 tokens in our main experiments as shown in Table 2, Table 4, and Table 5 in the original submission), the same CABS configuration remains stable. This is expected because CABS determines frame structure, not content semantics. Changing text length simply increases or decreases the number of frames proportionally. We did not observe any need for length-specific tuning
>
> **Domain robustness.**
>
> To evaluate transfer across domains, we conducted a cross-language experiment using the multilingual XL-Sum dataset on Gemma-7B-it. For English, Chinese, and Russian, we generated **500 watermarked** and **500 non-watermarked** samples (200 tokens each, and $m=3, H=12$), applying the same MirrorMark rules used in the main experiments. We then evaluated both detectors: Gumbel-max and Tour-Bayes.
>
> We added Fig. 16 which indicates the ROC of different languages in Appendix F.9 in the revised manuscript and [provided it in this link for direct access] (https://anonymous.4open.science/api/repo/MirrorMark-abcdefg/file/additional_experiments/Fig16-17-18.pdf?v=8b4eea3b). In that link, we also provided Fig. 17 and Fig. 18 to show the score distribution of watermarked and nonwatermarked samples for different languages.
>
> Fig. 16 shows that a threshold $\tau$ calibrated on one language transfers imperfectly to another: applying an English-calibrated threshold to Chinese results in a slightly higher FPR, whereas transfer between English and Russian remains stable. In contrast, a threshold learned on Chinese decreases the FPR for English and Russian. This behavior is fully explained by the entropy differences documented in prior work [1]: Chinese text exhibits consistently lower next-token entropy, causing a right-shift in both WM and NWM score distributions (Fig. 17 and Fig. 18). Importantly, the separation between WM and NWM remains stable across languages, which is indicated by AUC. Therefore, a small amount of domain-specific calibration can restore proper FPR control.
>
> [1] Universal Entropy of Word Ordering Across Linguistic Families. Montemurro, Marcelo A. AND Zanette, Damián H. PLOS ONE. 2021

---

> ### Author Response · Authors · 2025-11-25
>
> ## **Q: This creates a critical single point of failure: the multi-bit message is irrecoverably lost if the attacker rewrites even a few sentences, because the detector can no longer determine which tokens carry which message symbols.**
>
> We agree that synchronization is crucial for bit-level recovery. To quantify the resilience of CABS, we evaluated token insertion, deletion, and substitution attacks using the MarkLLM pipeline (as shown in Tables 6–8 in appendix F.4 of the revised manuscript, which are [provided in this link for easy access] (https://anonymous.4open.science/api/repo/MirrorMark-abcdefg/file/additional_experiments/Table6-8.pdf?v=7482f86f)). Across edit ratios $\\epsilon\in\\{0,0.2,0.4\\}$, we observe the following:
>
> - Insertion primarily shifts tokens forward. Even at $\\epsilon=0.4$, MirrorMark maintains high detectability (AUC = 0.999, TPR@1%FPR = 0.992), with bit accuracy reduced to 0.790, indicating that insertion mainly impairs bit recovery rather than WM/Non-WM separation.
>
> - Deletion is the most adversarial, reducing available tokens. At $\\epsilon=0.4$, AUC remains above chance (0.939) and TPR@1%FPR reaches 0.604. This degradation arises not only from the desynchronization of the token-to-position mapping but also from reduced detectability due to the smaller number of surviving tokens.
>
> - Substitution preserves length and is the least destructive. At $\\epsilon=0.4$, MirrorMark sustains strong detectability (AUC ≈ 0.998, TPR@1%FPR ≈ 0.992) and relatively high bit accuracy ($\sim$ 0.75–0.78), confirming that CABS effectively absorbs localized perturbations.
>
>
> Across these settings, CABS demonstrates strong resilience to realistic edits, and the robustness trends correlate cleanly with the edit type. We acknowledge that extreme edits can still disrupt bit-level synchronization, but MirrorMark maintains high detectability across all attack types.
>
> ---
>
>
> ## **Q: For paraphrasing attacks, bit accuracy drops to 54%, rendering the multi-bit message unrecoverable. Could you provide a use case where the recovered multi-bit message (with 54% accuracy) is still useful?**
>
> We thank the reviewer for raising this important point. Table 3 has been expanded to include TPR@1%FPR, different symbol sizes $m$, and the corresponding zero-bit baselines, which is [provided in this link] (https://anonymous.4open.science/api/repo/MirrorMark-abcdefg/file/additional_experiments/Table3.pdf?v=f7c37638) for quick reference. Paraphrasing fundamentally rewrites sentence structure, destroying local dependencies needed for bit-level decoding. Consequently, bit accuracy drops to 54%, and we agree this is insufficient for reliable message extraction. However, paraphrasing does not fully remove the global statistical bias introduced by MirrorMark. As shown in Table 3:
> - MirrorMark retains higher AUC than other methods under paraphrasing.
> - Multi-bit variants show stronger detection than their zero-bit versions (e.g., TB with $m=4$ vs. TB 0-bit; G-max with $m=2$ vs. G-max 0-bit).
>
> This is because the multi-bit decoder selects the message hypothesis with the highest score, effectively amplifying the residual watermark signal that survives paraphrasing, whereas zero-bit detection cannot benefit from this amplification mechanism. Thus, while bit-level decoding becomes unreliable, binary watermark detection remains effective under paraphrasing, and MirrorMark achieves the strongest detectability among all evaluated methods. We view robustness to aggressive rewriting as an important open challenge for the entire watermarking literature, not only MirrorMark.
>
> ---
>
> ## **Q: The core distinguishing feature of watermarking is robustness against modifications, whereas steganography prioritizes undetectability. MirrorMark’s preserving the exact probability distribution aligns more naturally with steganographic goals.**
>
> We thank the reviewer for raising this conceptual concern. We clarify that MirrorMark is fundamentally a watermarking method, although it can also serve steganographic purposes due to its distortion-free design and high bit accuracy even when embedding a large number of bits. As shown in our evaluation, none of the state-of-the-art watermarking baselines are robust to high-quality paraphrasing, since semantic rewriting inherently disrupts the local token dependencies required for multi-bit recovery. Consequently, reduced robustness under aggressive paraphrasing should be viewed as a limitation shared by the entire field, rather than evidence that a method belongs to steganography. Improving watermark reliability under such strong semantic attacks remains an important open research problem in LLM watermarking, and we explicitly identify it as a key direction for future work.

---

> ### Author Response · Authors · 2025-11-25
>
> ## **Q: The paper suffers from excessive mathematical formalism.**
>
> We thank the reviewer for this useful suggestion. To improve clarity and reduce unnecessary mathematical burden, we streamlined the theoretical presentation by removing non-essential notation and retaining only the key quantities that directly influence the EER for MirrorMark. We also added the new Fig. 3 in the revised manuscript, and [it is provided in this link] (https://anonymous.4open.science/api/repo/MirrorMark-abcdefg/file/Fig3.pdf?v=6818ae96) for direct access, which visualizes how the theoretical EER varies with the relevant parameters for both the Gumbel-max–based and tournament-sampling–based variants. This figure provides an intuitive summary of the theoretical trends without requiring readers to follow the full derivation.
>
> At the beginning of Section 4, we now explicitly clarify the purpose and scope of Theorem 4.1, stating that it is designed to illustrate the core statistical mechanism behind detectability in the simplest single-position setting ($H=1$). To make the theorem more interpretable, we added new single-position experiments in Appendix F (Figs. 14–15, which are also [provided in this link for easy access] (https://anonymous.4open.science/api/repo/MirrorMark-abcdefg/file/additional_experiments/Fig14-15.pdf?v=58791b53)), which closely mirror the theoretical predictions—for example, detectability decreases as $m$ increases, and the Gumbel-max variant exhibits stronger separation than the tournament-sampling variant. While Theorem 4.1 is not intended to model the full multi-position setting, the single-position analysis highlights the fundamental statistical principles—such as how the mean and variance of the sequence-level score determine the separation between $\mathcal{H}_0$ and $\mathcal{H}_1$, which continue to govern multi-position watermarking behavior.
>
> ---
>
> ## **Q: Have you explored a “soft” version of MirrorMark that applies partial mirroring?**
> We understand the reviewer’s suggestion as replacing MirrorMark’s full mod-1 mirroring with a weighted interpolation of the form:
> \[
> u' = (1-\lambda)u + \lambda\,\Psi(u;\psi_M), \qquad 0<\lambda<1,
> \]
> where the original MirrorMark transformation is $\Psi(u;\psi_M) = (2\psi_M - u)\bmod 1$ as shows in equation 8 in our submission. This would partially mirror the random values instead of applying the full reflection. To clarify why MirrorMark uses full mod-1 mirroring, consider the $m{=}1$ case. If the embedded bit is 0, MirrorMark uses $1-u$ for every token and samples $x^\ast = \arg\max_{1\le i\le V} (1-U_i)^{1/p(x_i)} $, whereas if the embedded bit is 1, MirrorMark uses $u$ and samples $x^\ast = \arg\max_{1\le i\le V} U_i^{1/p(x_i)}$.
>
> Because the mod-1 reflection $u\mapsto(1-u)$ preserves the base uniform distribution, the transformed values remain uniformly distributed on $[0,1)$, and therefore the token sampling distribution is unchanged. This measure-preserving property guarantees MirrorMark is distortion-free.
>
> In contrast, the proposed soft version $u' = (1-\lambda)u + \lambda\Psi(u;\psi_M)$
> no longer preserves uniformity: $u'$ becomes a biased mixture of $u$ and its reflection. The sampling probabilities would therefore shift, breaking the distortion-free guarantee central to MirrorMark’s design.
>
> ---
>
> ## **Q: What is the actual wall-clock time for encoding and decoding compared to non-watermarked generation?**
>
> | Method | Generation (WM) | Generation (NWM) | Detection (WM) | Detection (NWM) |
> |--------|------------------|------------------|------------------|------------------|
> | **Gumbel-max MirrorMark** | 19.4888 s | 16.1474 s | 0.3046 s | 0.3046 s |
> | **Tournament MirrorMark** | 19.3442 s | 16.3877 s | 0.2992 s | 0.3013 s |
>
> We thank the reviewer for highlighting this important point. The wall-clock measurements show that watermarking adds only a small overhead to generation. For a 400-token sequence embedding 36 bits ($m=3, H=12$), both Gumbel-max–based MirrorMark and tournament-sampling MirrorMark increase generation time by about 3 seconds compared to non-watermarked sampling. This overhead comes mainly from generating random values for each token.
>
> Importantly, detection time remains almost identical between watermarked and non-watermarked text (about 0.30 s per sample), since decoding only involves a single pass of score accumulation over the 400 tokens. This confirms that MirrorMark’s detector is lightweight and introduces no noticeable computational cost.

---

### Official Review · Reviewer_9mcw · 2025-10-25

**Soundness:** 2
**Presentation:** 2
**Contribution:** 2
**Rating:** 2
**Confidence:** 5

**Summary:**

Dear Area Chair,

This paper does not use the ICLR official template and should be desk reject. The margin is significant wider than the offical template.

Best regards,
Reviewer

**Strengths:**

NA

**Weaknesses:**

NA

**Questions:**

NA

---

> ### Author Response · Authors · 2025-11-25
>
> We thank the reviewer for pointing out the template formatting issue. We apologize for the oversight. The manuscript has now been updated to fully adhere to the official ICLR template and fits within the page limit. In addition, following the suggestions from other reviewers, we have added further experiments and analysis. Please refer to the newly uploaded revised manuscript.
>
> Best,
> Authors

---

### Official Review · Reviewer_GKSn · 2025-10-29

**Soundness:** 4
**Presentation:** 1
**Contribution:** 2
**Rating:** 2
**Confidence:** 5

**Summary:**

This paper introduces a multi-bit distortion-free watermarking scheme for LLMs. Specifically, it introduces two components: mod-1 mirroring and CABS to expand existing 1-bit watermarks (AAR and SynthID) to the multi-bit case.

They provide a theoretical explanation of how equal error rates scale with the number of pseudo-random draws for both schemes.

Lastly, they evaluate the detectability and bit accuracy trade-off with the quality of their schemes and several baselines and show robustness experiments against copy-paste attacks and (non-adversarial) paraphrasing.

**Strengths:**

- Propose a generic mod-1 mirroring function that allows associating positions with uniform random variables. We show how this concept enables multi-bit watermarking for two popular schemes: AAR Watermark and SynthID-Text.
- Introduce CABS to sample positions based on context, which, unlike prior work, ensures that all positions are sampled while maintaining robustness via a framing mechanism.
- Provide theoretical bounds on the equal-error rate with respect to the number of PRF draws for both studied schemes.
- Detectability/robustness evaluation compares Mirrormark with several prior and up-to-date baselines and shows that the proposed method is more effective on most aspects (detectability and the bit-accuracy–versus–quality trade-off, and robustness against copy-paste attacks).

**Weaknesses:**

- The presentation of the method (Section 3) is unclear. Connecting the different components (mod-1 mirroring, CABS, and the decoding and detection) requires a lot of back-and-forth to grasp how the method operates, as well as significant knowledge of prior works on multi-bit watermarks. I think the paper could benefit from a high-level explanation of the different components first (mod-1 mirroring encodes the position, CABS determines the position, and decoding decodes the position and computes the score), paired with a visual explanation of the method (see, for instance, Fig. 2 of [1]). Also, beware that some sentences are poorly phrased, and there are some typos (for instance, in l603 the beginning of the sentence is missing). Lastly, using $300$ tokens for Section 5.1 and $400$ for Section 5.2 is a bit inconsistent and makes comparison harder; I think it would be better if both used the same length.
- This paper has two key contributions: mod-1 mirroring and CABS. It would be good if the evaluation disentangled the two. For instance, using mod-1 mirroring with a prior work’s position sampler (for instance, MPAC [1]), and also using prior schemes (for instance, Red-Green schemes) with CABS. Thus, we could clearly see which component leads to an improvement (is it mod-1 mirroring, which allows the use of SynthID/AAR schemes; is it the position sampler; or is it both?).
- With the proposed scheme, the decision threshold is no longer supported by statistical testing (after the argmax operation, the distributions under the null hypothesis are not known); hence the authors suggest learning a threshold. Therefore, the statistical guarantees of the watermark with respect to FPR are not satisfied, and the paper presents no experiments showing that, in practice, with a learned threshold, the FPR is still properly controlled.
- The quality evaluation of the watermark is insufficient. As acknowledged by the authors, a known issue with distortion-free watermarks is diversity (l121). For instance, AAR is known to lead to repetitive sentences, yet perplexity tends to be lower with repetitive sentences (this could explain why the perplexity of Gumbel-max is lower than in the non-watermark case). Therefore, in the quality evaluation, it would be valuable to include additional metrics to measure repetitiveness (n-gram repetitions) or an LLM-as-a-judge score. On a minor note, the authors did not submit the code with their submission, which prevents independent human evaluation by reviewers of the watermark quality.
- The motivations for multi-bit watermarks are somewhat lacking. Also, the authors claim that single-bit watermarks cannot be used for multi-bit watermarking but later explain that in [2] they extend a single-bit watermark to the multi-bit case by associating one message per key. The evaluation could benefit from comparing MirrorMark on AAR/SynthID with the key-swapping approach as a baseline using the same underlying schemes.
- A key motivation behind the CABS design is robustness against token deletion, insertion, or substitution that could desynchronize the position signal. Yet the authors do not evaluate their watermark’s robustness against such attacks.
- The authors claim in the abstract that Theorem 4.1 provides interpretability for the empirical results and insights into the design of high-detectability multi-bit watermarks. In its current state, I think it does neither. There are no experiments showing that the bounds of Theorem 4.1 are achieved or verified (I think Theorem 4.1 is never referenced in the evaluation section), and no justification for how the results provide insights into watermark design. On the contrary, from Theorem 4.1 it appears that the AAR watermark is better than SynthID, yet in the experimental results this is not the case.
- CABS has many parameters whose impact is not evaluated (i.e., the max length, the window size $Q$, and the $f$). Also, it is unclear which parameters are used for the evaluation.

[1] Advancing beyond identification: Multi-bit watermark for large language models, Yoo et al.
[2] Three bricks to consolidate watermarks for large language models, Fernandez et al.

**Questions:**

- Could the authors evaluate mod-1 mirroring and CBAS independently? For instance, compare MirrorMark to mod-1 mirroring + MPAC on SynthID/AAR (evaluating detectability and robustness). Similarly, compare CBAS with MPAC on the coloring scheme from [2]. The goal is to understand which component explains the improvement of MirrorMark.
- Could the authors show how the TPR/FPR curves behave for unwatermarked text when the threshold is learned on one domain (for instance, English text) but then used on another domain (for instance, Chinese text)? Could this lead to an increase in false positives? If so, how should one learn the threshold in practice?
- Could the authors add additional quality metrics (LLM-as-a-judge and repetitiveness scores)? Also, I think it would be beneficial to see how the watermark performs on instruction tasks (instead of completions), which are a more realistic use case for LLMs. For instance, using the evaluation pipeline from [3].
- Could you evaluate the robustness of the watermark to token deletion, insertion, and substitution? (see [4])
- Could you ablate the components of CBAS and assess their impact on robustness and detectability? In particular, max_len, $W$, and $f$.
- Could the authors compare their approach to the naive baseline from [2], in which one key equals one message? In particular, is the naive approach better for small message lengths?
- How does Theorem 4.1 explain the observed results? How can it be used to guide watermark design?
- What is the role of the null symbol in mod-1 mirroring?
- In the paraphrasing robustness experiment, could you show the TPR as well?
- Can you show a single-bit watermark baseline from AAR/SynthID? It would be interesting to see whether using a multi-bit watermark leads to a smaller TPR (i.e., what is the cost of using a multi-bit watermark).

[3] WaterBench: Towards Holistic Evaluation of Watermarks for Large Language Models, Tu et al.\
[4] MarkLLM: An Open-Source Toolkit for LLM Watermarking, Pan et al.

---

> ### Author Response · Authors · 2025-11-25
>
> ## **Q: need the clarification of Section 3 and inconsistent token lengths**
>
> We appreciate the reviewer’s suggestion. In the revised manuscript, Section 3 now begins with a new high-level overview that explains how the three components interact, i.e., mod-1 mirroring, CABS, and decoding. We also added two new visual diagrams (Fig. 1 and Fig. 2, which are [provided in this link for easy access] (https://anonymous.4open.science/api/repo/MirrorMark-abcdefg/file/Fig1-2.pdf?v=c1d11c7e)) illustrating the mod-1 mirroring process and the CABS workflow step-by-step, which substantially improves readability. Finally, to address the token-length concern, Section 5.1 now explicitly refers readers to the corresponding 400-token results already included in Table 5 in Appendix F.2 in the original submission, ensuring consistent comparison across sections.

---

> ### Author Response · Authors · 2025-11-25
>
> ## **Q: Need to implement ablation study for CABS and robustness under insertion/deletion/substitution attacks**
>
> We appreciate the reviewer for raising this important point. To address these concerns comprehensively, we conducted an extensive ablation study that jointly examines
> (i) the impact of CABS parameters on performance, and
> (ii) the robustness of CABS-enabled MirrorMark against token insertion, deletion, and substitution.
> Our default configuration is $f = 3, W = 4, max\\_factor= 1.5$, which is now explicitly documented in Appendix E of the revised manuscript (i.e., Experimental Setup). We implement this experiment on Gumbel-max based MirrorMark.
> For clarity, we briefly restate the roles of the three CABS parameters and refer the reviewer to Fig. 2 in the revised manuscript for an intuitive visual illustration.
>
> ---
>
> ### **Role of the CABS parameters**
>
> The parameter $f$ determines how often a new frame can start: a new frame is opened only when the last $f$ bits of the content hash of the current context are all zeros. For example, when $f = 3$, the last three bits must all be zero, which occurs with probability $1/8$ under a near-uniform hash. Thus, larger $f$ leads to fewer and longer frames, while smaller $f$ results in more frequent frame boundaries.
>
> The parameter $W$ specifies the context size used to compute the content hash to determine whether to anchor a frame; that is, each hash is computed over the most recent $W$ tokens.
>
> The parameter $max\\_factor$ determines the maximum allowable frame length.  For example, when $max\\_factor = 1.5$, the maximum frame length is set to $max\\_len = 1.5H$, where $H$ denotes the number of positions.
>
> ---
>
> ### **Impact of CABS parameters**
>
> We added Tables 6–8 in Appendix F.4 in the revised manuscript, which are also [provided in this link] (https://anonymous.4open.science/api/repo/MirrorMark-abcdefg/file/additional_experiments/Table6-8.pdf?v=7482f86f), to report ablations over frame anchor size $f$, window size $W$, and $max\\_factor$ across three types of edit attacks.
> We observe clear trends emerge across all settings:
>
> 1. $f = 3$ consistently provides the highest bit accuracy and strong TPR@1%FPR across insertion, deletion, and substitution, indicating that it offers the best trade-off between robustness and token coverage.
>
> 2. $W = 4$ performs best or near-best for all edit ratios, capturing sufficient contextual information without overfitting to local perturbations.
>
> 3. $max\\_factor = 1.5$ achieves the strongest robustness across edit rates, balancing frame-size flexibility and stability.
>
> ### **Robustness under editing attacks**
>
> Using the MarkLLM pipeline, we evaluated MirrorMark under insertion, deletion, and substitution with edit ratios $\\epsilon \in \\{0, 0.2, 0.4\\}$ (Tables 6–8). Across these attacks:
>
> - **Insertion** primarily shifts tokens forward. Even at $\epsilon=0.4$, MirrorMark maintains high detectability (AUC = 0.999, TPR@1%FPR = 0.992), with bit accuracy reduced to 0.790, indicating that insertion mainly impairs bit recovery rather than WM/Non-WM separation.
>
> - **Deletion** is the most adversarial, reducing available tokens. At $\\epsilon=0.4$, AUC remains above chance (0.939) and TPR@1%FPR reaches 0.604. This degradation arises not only from the desynchronization of the token-to-position mapping but also from reduced detectability due to the smaller number of surviving tokens.
>
> - **Substitution** preserves length and is the least destructive. At $\\epsilon=0.4$, MirrorMark sustains strong detectability (AUC ≈ 0.998, TPR@1%FPR ≈ 0.992) and relatively high bit accuracy ($\sim$ 0.75–0.78), confirming that CABS effectively absorbs localized perturbations.

---

> ### Author Response · Authors · 2025-11-25
>
> ## **Q: Need to distengle the contribution of mod-1 mirroring and CABS**
>
> We thank the reviewer for this insightful suggestion. To disentangle the contributions of mod-1 mirroring and CABS, we have added a new evaluation that systematically combines different position samplers with different watermarking schemes. The results are provided in Fig. 10, Fig. 11, and Fig. 12 in Appendix F.6 in the revised manuscript, and please [check this link for Fig. 10, Fig. 11, and Fig. 12] (https://anonymous.4open.science/api/repo/MirrorMark-abcdefg/file/additional_experiments/Fig10-12.pdf?v=5aedfaad). Here, since the default setting for MPAC is $m=2$ in their paper, we also use $m$=2 in MirrorMark for fair comparison.
>
> In particular, we incorporate the position samplers used in MPAC and RSBH, which we denote as *NaiveHash* and *DPHash*, respectively.
> - **NaiveHash** (MPAC, Section 3.2) seeds a PRF using the previous $h$ tokens to randomly select a position.
> - **DPHash** (RSBH, Section 4.2) constructs a balanced token-to-segment mapping table through a secret-key shuffle followed by a dynamic programming procedure.
>
> Because the DPHash table released in the authors’ official implementation is constructed with $h=1$, we evaluate performance under this setting in Fig. 11. Since our main experiments use $h=4$ unless otherwise noted, we additionally report results for $h=4$ in Fig. 12. For both settings, we include the **Gini score** in Fig. 10, which measures positional imbalance: lower Gini indicates more balanced token allocation.
>
> Across all detectability metrics, we observe that **CABS consistently outperforms NaiveHash and DPHash for MirrorMark** (both Tour-Bayes and Gumbel-max). In contrast, for MPAC, the AUC and TPR@1%FPR of CABS are comparable to those of NaiveHash and DPHash, with CABS achieving only slightly higher bit accuracy. This is expected: **balanced token allocation improves message decoding** because positions with few or no tokens provide only weak or random evidence. CABS is explicitly designed to mitigate this imbalance by distributing tokens more evenly across positions.
>
> ---
>
> ### **MirrorMark benefits strongly from CABS but MPAC does not**
>
> The difference between MirrorMark and MPAC stems from how each method aggregates positional evidence. MirrorMark aggregates evidence from all positions. For example, consider 100 tokens allocated across four positions as 85–5–5–5:
>
> - For watermark text, position 1 (85 tokens) provides a strong signal.
> - However, positions 2–4 (5 tokens each) contribute mostly noise.
> - When combined in the final score, this noise *dilutes* the strong evidence from position 1.
>
> As a result, the score distributions for watermark vs. non-watermark become harder to separate, reducing AUC and TPR@1%FPR.
>
> In contrast, MPAC is inherently robust to imbalance. MPAC behaves differently under the same allocation. It keeps only the largest vote per position, then sums these maxima:
>
> - The dominant position with 85 tokens decisively votes for the correct message.
> - The lightly populated positions contribute very little and critically do **not** introduce harmful noise.
> - Non-watermark text remains roughly balanced across message candidates.
>
> Therefore, MPAC’s detectability remains stable under uneven token allocations.

---

> ### Author Response · Authors · 2025-11-25
>
> ## **Q: Insufficient quality evaluation and need repetitiveness and LLM-as-judge metrics**
>
> We thank the reviewer for raising this important point. In Appendix F.7 in the revised manuscript, we have added two complementary quality evaluations: **LLM-as-a-judge scores** (GPT-4o; Fig. 13) and **repetition-based metrics** (Distinct-2 and Repetition Rate; Table 10). Please refer to [this link for direct access to Fig. 13 and Table 10] (https://anonymous.4open.science/api/repo/MirrorMark-abcdefg/file/additional_experiments/Fig13_Table10.pdf?v=ee6e6f53). In addition, to facilitate independent human evaluation, we have uploaded all MirrorMark-generated samples—including
> (1) the prompts ([the link is] (https://anonymous.4open.science/api/repo/MirrorMark-abcdefg/file/prompt%20for%20LLM%20score.png?v=d7cdab75)),
> (2) the corresponding MirrorMark-generated texts, and
> (3) the full GPT-4o evaluation logs—
> to [this link] (https://anonymous.4open.science/r/MirrorMark-abcdefg/text_quality_json.zip) (please click 'view raw' to access the folder). This allows reviewers to directly inspect the generated text without needing to run our code.
>
> The GPT-4o evaluations show that MirrorMark’s linguistic quality is effectively indistinguishable from the non-watermarked baseline. Across all settings, the overall score difference remains within 0.05–0.10, well within the natural variance of GPT-4o. This confirms that our mod-1 mirroring, being distribution-preserving by construction, does not degrade fluency or coherence. In contrast, distortion-based baselines such as MPAC and RSBH receive noticeably lower GPT-4o scores, which is consistent with their higher perplexity and the known side effects of logit biasing.
>
> The diversity metrics reinforce this conclusion. While MPAC and RSBH may appear to achieve higher Distinct-2 or lower repetition rate, these improvements stem from artificial skewing of token probabilities away from natural language usage—precisely why GPT-4o judges their text as less coherent and less natural. By comparison, MirrorMark (especially the tournament-sampling variant) matches the non-watermarked diversity almost exactly. Although Gumbel-max naturally reduces diversity, GPT-4o confirms that this does not harm sentence quality: the outputs remain fluent, coherent, and well-structured.
>
> Together, these results demonstrate that MirrorMark preserves linguistic quality, while distortion-based baselines introduce detectable degradation.

---

> ### Author Response · Authors · 2025-11-25
>
> ## **Q: Comparison with the “one key = one message” baseline [1]**
>
> We thank the reviewer for this insightful suggestion. We have added the corresponding results as in Fig. 14 in Appendix F.8 in the revised manuscript, comparing Gumbel-max based MirrorMark with Algorithm 1—the multi-bit watermarking scheme obtained by extending Gumbel-max zero-bit watermarking—introduced in [1]. Please [refer to this link for easy access to Fig. 14] (https://anonymous.4open.science/api/repo/MirrorMark-abcdefg/file/additional_experiments/Fig14-15.pdf?v=58791b53). We observe that Gumbel-max based MirrorMark consistently outperforms the mechanism in [1].
>
> In the multi-key design, when the decoder assumes an incorrect message, it reconstructs an entirely independent PRF sequence. As a result, the score under an incorrect hypothesis behaves indistinguishably from non-watermarked text. Because these wrong-message hypotheses incur no penalty, the separation between the correct and incorrect message scores remains limited.
>
> In contrast, MirrorMark employs a single PRF together with message-dependent mod-1 mirroring. When the decoder assumes an incorrect message, it evaluates the signal using an incorrect mirroring center, which systematically reduces the score and imposes a strong penalty on wrong hypotheses. This penalty mechanism creates substantially larger score gaps between correct and incorrect messages, ultimately yielding much higher bit accuracy.
>
> [1] Three bricks to consolidate watermarks for large language models, Fernandez et al.

---

> ### Author Response · Authors · 2025-11-25
>
> ## **Q: Need to provide validity of FPR control and threshold transferability across language domains**
>
> We appreciate the reviewer for raising these important questions regarding (i) the statistical grounding of the decision threshold after the $argmax$ operation, and (ii) how a learned threshold behaves when transferred across domains such as different languages. Below we address both aspects jointly.
>
> ---
>
> ### **Theoretical characterization of FPR for both variants of MirrorMark**
>
> Although the $argmax$ over multiple message hypotheses makes the exact null distribution analytically intractable, both variants of MirrorMark still admit principled approximations of the FPR.
>
> **Gumbel-max variant**
>
> Eq. (20) shows that the sequence-level score $C_{M}(W,\mathsf{sk})=\frac{1}{T}\sum_{t=1}^T S_{M}(W_t,\mathsf{sk})$ where $S_{M}(W_t,\mathsf{sk})=\ln\frac{1}{1-\Psi(u_{t},\psi(M))}$ is an average of $T$ per-token contributions.
>
> By the Central Limit Theorem, as $T$ grows, we have $C_{M}(W,\mathsf{sk}) \sim \mathcal{N}(\mu_{\mathcal{H}_0}, \sigma^2_{\mathcal{H}_0})$ under the null hypothesis.
>
> While the statistics $\\{C_M\\}$ across messages are not strictly independent, each has variance $O(1/T)$. Consequently, the event $\\{\max_M C_M > \tau\\}$ is dominated by one unusually large deviation
> rather than simultaneous correlated deviations, allowing us to approximate $\\{C_M\\}$ as independent when estimating the tail probability.
>
> Applying Lemma C.1, we obtain a closed-form approximation of $FPR = \Pr(Z > \tau) \text{for } Z=\max_{M\in\{0,\dots,2^m - 1\}} C_M$
> which yields a principled estimate of the FPR even after the $argmax$ operation.
>
> **Tournament-sampling variant**
>
> Eq. (45) provides the relationship between the scores under $M=0$ and $M=1$with the same $u$-value. When $m=1$, this relationship directly informs the analytic expression in Eq. (50), enabling a theoretically grounded FPR approximation for the tournament-sampling detector as well.
>
> ---
>
> ### **Empirical evaluation of threshold transfer across languages**
>
> To further validate that the learned threshold behaves properly in practice, following [2], we conducted a cross-language experiment using the multilingual XL-Sum dataset on Gemma-7B-it. For English, Chinese, and Russian, we generated **500 watermarked** and **500 non-watermarked** samples (200 tokens each), applying the same MirrorMark rules used in the main experiments. We then evaluated both detectors: **Gumbel-max** and **Tour-Bayes**.
>
> We added Fig. 16 which indicates the ROC of different languages in Appendix F.9 in the revised manuscript and [provided it in this link for direct access] (https://anonymous.4open.science/api/repo/MirrorMark-abcdefg/file/additional_experiments/Fig16-17-18.pdf?v=8b4eea3b). In that link, we also provided Fig. 17 and Fig. 18 to show the score distribution of watermarked and nonwatermarked samples for different languages.  We can observe that in Fig. 16
> - A threshold \(\tau\) calibrated on English increases the FPR of Chinese, when applied to Chinese.
> - Transfer between English and Russian is relatively stable.
> - A threshold trained on Chinese lowers the FPR when applied to English and Russian.
>
> This pattern aligns with entropy differences documented in prior work [3]: **Chinese text exhibits lower next-token entropy**, leading to a right-shift in both WM and NWM score distributions. Importantly, the separation between WM and NWM remains stable, as evidenced by the AUC being nearly unchanged across languages.
>
> Thus, while absolute thresholds require modest domain-specific calibration, MirrorMark's underlying signal structure is consistent across languages, enabling reliable detection performance once calibrated.
>
> [2] Scalable watermarking for identifying large language model outputs. Sumanth Dathathri, Abigail See et al. Nature 2024
>
> [3] Universal Entropy of Word Ordering Across Linguistic Families. Montemurro, Marcelo A. AND Zanette, Damián H. PLOS ONE. 2021

---

> ### Author Response · Authors · 2025-11-25
>
> ## **Q: Need to show TPR in the paraphrasing robustness experiment**
>
> We appreciate the reviewer for raising this point. In the revised manuscript, Table 3 has been expanded to include TPR@1%FPR, multiple symbol sizes $m$, and the corresponding zero-bit baselines. [Please find this link to Table 3] (https://anonymous.4open.science/api/repo/MirrorMark-abcdefg/file/additional_experiments/Table3.pdf?v=f7c37638) for easy access.
>
> Paraphrasing rewrites sentence structure and alters local token transitions, breaking the fine-grained dependencies required for accurate bit recovery; this naturally results in lower bit accuracy. However, paraphrasing does not fully eliminate the underlying statistical bias introduced by the watermark. As a result, the global separability between watermarked and non-watermarked samples remains largely preserved.
>
> MirrorMark retains consistently higher AUC after paraphrasing. Although its TPR@1%FPR is not high, it remains the strongest among all evaluated methods. We acknowledge that paraphrasing poses a particularly challenging setting for MirrorMark; however, this challenge is **not unique** to MirrorMark and similarly affects other watermarking approaches. Developing techniques that remain reliable under such aggressive rewriting is an important direction for future research.
>
> The updated results also show that MirrorMark’s multi-bit variants achieve higher detection performance than their zero-bit counterparts (e.g., TB with $m=4$ vs. TB 0-bit; G-max with $m=2$ vs. G-max 0-bit). This is because the multi-bit decoder selects the message hypothesis with the highest score, effectively amplifying the residual watermark evidence that survives paraphrasing, which is an advantage that zero-bit detection does not possess.
>
> ---
>
> ## **Q: What is the role of the null symbol in mod-1 mirroring?**
>
> The motivation for introducing the null symbol is to provide flexibility: it allows the multi-bit extension to revert to the original zero-bit case when desired. This ensures that the mod-1 mirroring framework remains compatible with both zero-bit and multi-bit watermarking without requiring separate algorithmic branches.
>
> When $m=1$, only two payload symbols $M=1$ are used for embedding, and both the reserve and non-reserve designs apply exactly the same mirroring transformations to these two messages. Thus, for $m=1$, the two designs are theoretically expected to produce **identical performance**, since the null symbol never plays a role.
>
> For larger message sizes $m \in \\{2,3,4\\}$, we empirically compare the two designs in Fig. 7 in Appendix F.1 in the revised manuscript, and [this is the link to Fig. 7 for easy access] (https://anonymous.4open.science/api/repo/MirrorMark-abcdefg/file/additional_experiments/Fig7.pdf?v=c9f4e5b3). Across AUC, bit accuracy, and TPR@1%FPR, the performance curves of the reserve and non-reserve variants **closely track each other** across all token lengths. While small discrepancies occur at low token counts, these fall within natural sampling variability, and we observe **no consistent or systematic performance gap** between the two approaches.
>
> Overall, the null symbol serves as a compatibility mechanism for the multi-bit extension, and its inclusion does not introduce any measurable degradation or advantage in empirical performance.

---

> ### Author Response · Authors · 2025-11-25
>
> ## **Q: Need to clarify theoretical interpretation (Theorem 4.1) and its empirical verification**
>
> We thank the reviewer for raising these important questions, and we agree that our original presentation did not make the scope and implications of Theorem 4.1 sufficiently clear. In the revised manuscript, we explicitly note at the beginning of Section 4 that the theorem characterizes detectability in the **single-position** setting ($H=1$), whereas the main experiments evaluate **multi-position** watermarking ($H>1$). Multi-position settings introduce additional stochasticity from token allocation and frame construction, which
> are handled empirically rather than through a closed-form theoretical model; therefore, a direct one-to-one correspondence between Theorem 4.1 and the multi-position experiments is not expected.
>
> To clarify the connection, we have added new single-position experiments in Appendix F.8 in the revised manuscript (Fig. 14 and Fig. 15, which are [shown in this link] (https://anonymous.4open.science/api/repo/MirrorMark-abcdefg/file/additional_experiments/Fig14-15.pdf?v=58791b53) for easy acess), evaluating performance up to 60 tokens. These results closely match the qualitative trends predicted by Theorem 4.1:
>
> - The Gumbel-max variant demonstrates stronger detectability than the tournament-sampling variant under $H=1$.
> - Performance decreases as $m$ increases, in accordance with the theorem.
>
> We now reference these correspondences explicitly in lines 442–443 of the revised manuscript. Thus, although Theorem 4.1 does not aim to describe the full multi-position setting, the single-position analysis captures the core statistical mechanisms that also shape multi-position behavior—for example, how the mean and variance of the sequence-level score determine the separation between hypotheses $\mathcal{H}_0$ and $\mathcal{H}_1$.
>
> Regarding the reviewer’s question about the theorem’s role in guiding design, our intention was not to claim that Theorem 4.1 provides a complete design blueprint. Rather, the theorem highlights the factors that most strongly affect detectability—namely, the mean–variance trade-off in the sequence-level score. These insights help explain several empirical trends, including
> (i) why different sampling strategies yield different performance profiles, and
> (ii) why detectability decreases as $m$ increases.
>
> Together, these clarifications more accurately convey the scope of Theorem 4.1 without overclaiming its role in prescribing a full watermark design methodology.
>
> ## **Q: Need to show performance of MirrorMark on the instruction task**
>
> We thank the reviewer for the helpful suggestion. Following [2], we have added an experiment in Fig. 19 in Appendix F.10 and [this is the link for easy acess] (https://anonymous.4open.science/api/repo/MirrorMark-abcdefg/file/additional_experiments/Fig19.pdf?v=345783d4). It evaluates MirrorMark on the instruction task, using 500 ELI5 prompts and the instruction-tuned Gemma-7B-it model. The results show a clear contrast with completion-style tasks:
> - On C4 completions, TPR@1%FPR exceeds 95% at only 60 tokens for both 1-bit and 3-bit messages.
> - On instruction task, detection is substantially weaker (e.g., TPR@1%FPR is about 80% for Gumbel-max and 65% for tournament sampling at 100 tokens for 1-bit messages).
>
> This performance drop arises because instruction-tuned models produce more deterministic next-token distributions, driven by structured prompts and strong instruction-following behavior. This reduces the available entropy in token sampling, thereby diminishing the magnitude of watermark-induced perturbations. As a result, statistical signal accumulation becomes inherently more difficult than in free-form completions.
>
> To further analyze this effect, we evaluated a variant where tournament sampling draws Bernoulli rather than Uniform $u$-values, which maximizes diversity. As shown in Fig. 19(k), this modification improves TPR@1%FPR by approximately 5% at 100 tokens, confirming that reduced randomness is indeed the key limiting factor. However, Bernoulli sampling supports only binary $u$-values and therefore cannot be used for multi-bit watermarks with flexible message sizes $m$.
>
> Overall, these results highlight that instruction-following tasks pose a more challenging setting for watermark detection due to their reduced entropy, and addressing this remains an important direction for future work.

---

> ### Comment · Reviewer_GKSn · 2025-11-28
>
> I thank the authors for extensively adjusting their work and making it clearer. Given the length of the rebuttal and the numerous experiments added, I do not have the time to give an in-depth review of the added content.
>
> Nonetheless, most of my concerns have been addressed, and I better understand the benefits and limitations of Mirrormark compared to prior works in multibit watermarking. I will therefore increase my score once possible (as of now the edit button is unavailable for the original review).
>
> However, regarding the formatting of their manuscript, the authors do not respect the ICLR official template. As highlighted by Reviewer 9mcw, the original manuscript had much smaller margins than those of the official template (which likely allowed it to fit in 9 pages). Now, the updated revision has correct margins, but the spacing between paragraphs and headers is *excessively* reduced. Trying to bypass the official template twice in a row is bad practice and unfair with respect to other works abiding by the rules.

---

> > ### Author Response · Authors · 2025-11-29
> >
> > Thank you very much for your careful follow-up and for acknowledging the improvements in clarity and experiments. We sincerely appreciate your willingness to increase the score.
> >
> > Regarding the formatting issue, we fully acknowledge your concern. In the previous revision, we used a few \vspace adjustments in order to meet the page limit while avoiding large-scale modification of the main text. In hindsight, this unintentionally affected the paragraph and header spacing and indeed deviated from the ICLR template defaults. We apologize for this oversight.
> >
> > To address this, we have now removed all manual spacing commands (including any \vspace or \setlength adjustments) and fully restored the official ICLR template’s default spacing and margins. To remain within the page limit in a fully compliant manner, we relocated the original Figure 5 to Appendix F.5 (now Figure 8), without altering the scientific content.
> >
> > The newly uploaded version strictly follows the official template with no formatting overrides of any kind. We genuinely appreciate your patience and your helpful feedback, and we apologize again for the inconvenience caused.
> >
> > Thank you for your time and for helping us improve the quality and fairness of the submission.

---

### Official Review · Reviewer_Auzr · 2025-10-30

**Soundness:** 3
**Presentation:** 2
**Contribution:** 3
**Rating:** 6
**Confidence:** 3

**Summary:**

This paper proposes MirrorMark, a multi-bit and distortion-free watermark. Specifically, it first presents mod-1 mirroring process to extend existing watermarks to multi-bit. Second, to improve robustness against watermark attacks, the authors develop a content-anchored balanced scheduler (CABS). This allows fewer tokens to carry more symbols. Compared to existing multi-bit watermarks, MirrorMark achieves a better tradeoff between watermark strength and text quality.

**Strengths:**

1. MirrorMark is easy to implement by extending existing one-bit watermarks.
2. Compared to existing multi-bit watermarks, CABS is more robust.
3. The authors also provide a theoretical analysis between the number of pseudo-random functions and error rate of watermark detection.

**Weaknesses:**

1. Although I like the idea of the paper, the manuscript is difficult to follow due to its formatting (e.g., tables and figures) and lack of clarity.
2. From Table 3, the experiments do not demonstrate consistent robustness in terms of bit accuracy. The authors should provide additional analysis or experiments to support their claims.

**Questions:**

1. Could the authors provide an ablation study on CABS to evaluate the contribution of the module?
2. Could the authors provide a sensitivity analysis on the number of bits $M$ in comparison to the benchmark methods? Since MirrorMark encodes $M$ bits using fewer tokens, it is possible that for sufficiently large $M$, the multi-bit watermark may fail in practice.

---

> ### Author Response · Authors · 2025-11-25
>
> **Q: Although I like the idea of the paper, the manuscript is difficult to follow due to its formatting (e.g., tables and figures) and lack of clarity.**
>
> We appreciate the reviewer’s suggestion. In the revised manuscript, Section 3 now begins with a new high-level overview that explains how the three components, mod-1 mirroring, CABS, and decoding interact. We also added two new visual diagrams (Fig. 1 and Fig. 2, which are [provided in this link for direct access] (https://anonymous.4open.science/api/repo/MirrorMark-abcdefg/file/Fig1-2.pdf?v=c1d11c7e)) illustrating the mod-1 mirroring process and the CABS workflow step-by-step, which substantially improves readability.
>
> ---
>
> **Q: From Table 3, the experiments do not demonstrate consistent robustness in terms of bit accuracy. The authors should provide additional analysis or experiments to support their claims.**
>
> We appreciate the reviewer for raising this point. In the revised manuscript, Table 3 has been expanded in the revised manuscript to include TPR@1%FPR, multiple symbol sizes \(m\), and the corresponding zero-bit baselines, and this is the [link to Table 3 for direct access] (https://anonymous.4open.science/api/repo/MirrorMark-abcdefg/file/additional_experiments/Table3.pdf?v=f7c37638).
>
> Paraphrasing rewrites sentence structure and alters local token transitions, breaking the fine-grained dependencies required for accurate bit recovery; this naturally results in lower bit accuracy. However, paraphrasing does not fully eliminate the underlying statistical bias introduced by the watermark, so the global separability between watermarked and non-watermarked samples remains largely preserved. As MirrorMark exhibits stronger watermark signal strength than other approaches prior to attack (Table 1), it retains consistently higher AUC after paraphrasing, and although its TPR@1%FPR is not high, it remains the best among all evaluated methods. We acknowledge that paraphrasing poses a particularly challenging setting for MirrorMark. However, this limitation is not unique to MirrorMark and also affects other watermarking approaches. Therefore, developing techniques that remain reliable under such aggressive rewriting is an important direction for future research.
>
> The updated results also show that MirrorMark’s multi-bit variants achieve higher detection performance than their zero-bit counterparts (e.g., TB with \(m{=}4\) vs. TB 0-bit; G-max with \(m{=}2\) vs. G-max 0-bit). This is because the multi-bit decoder selects the message hypothesis with the highest score, effectively amplifying the residual watermark signal that survives paraphrasing, whereas zero-bit detection cannot benefit from this amplification mechanism.

---

> ### Author Response · Authors · 2025-11-25
>
> **Q: Could the authors provide an ablation study on CABS to evaluate the contribution of the module?**
>
> We appreciate the reviewer’s attention to implement ablation study for CABS. To address these concerns comprehensively, we conducted an extensive ablation study that jointly examines
> (i) the impact of CABS parameters on performance, and
> (ii) the robustness of CABS-enabled MirrorMark against token insertion, deletion, and substitution.
>
> Our default configuration is f=3, W=4, max\_factor=1.5, which is now explicitly documented in Appendix E (experimental setup) of the revised manuscript. For clarity, we briefly restate the roles of the three CABS parameters and refer the reviewer to [Fig. 2] (https://anonymous.4open.science/api/repo/MirrorMark-abcdefg/file/Fig1-2.pdf?v=c1d11c7e) in the revised manuscript for an intuitive visual illustration.
>
> - The parameter f determines how often a new frame can start: a new frame is opened only when the last f bits of the content hash of the current context are all zeros. For example, when f=3, the last three bits must all be zero, which happens with probability 1/8 under a near-uniform hash. Therefore, larger f leads to fewer and longer frames, while smaller f results in more frequent frame boundaries.
> - The parameter W specifies the context size used to compute the content hash; that is, each hash is computed over the most recent W tokens of the generated text.
> - max\_factor determines the maximum allowable frame length. For example, when max\_factor=1.5, the maximum frame length is set to max\_len = 1.5H, where H denotes the number of positions.
>
> **Impact of CABS parameters.**
> We added Tables 6–8 in Appendix F.4 in the revised manuscript, please [find this link for quick access] (https://anonymous.4open.science/api/repo/MirrorMark-abcdefg/file/additional_experiments/Table6-8.pdf?v=7482f86f), which report ablations over frame anchor size f, window size W, and max\_factor across three types of edit attacks. Clear trends emerge across all settings:
>
> 1. f=3 consistently provides the highest bit accuracy and strong TPR@1%FPR across insertion, deletion, and substitution, indicating that it offers the best trade-off between robustness and token coverage.
> 2. W=4 performs best or near-best for all edit ratios, capturing sufficient contextual information without overfitting to local perturbations.
> 3. max\_factor=1.5 achieves the strongest robustness across edit rates, balancing frame-size flexibility and stability.
>
> Overall, the default configuration f=3, W=4, max\_factor=1.5 emerges as empirically optimal among all tested combinations.
>
> **Robustness under editing attacks.**
> Using the MarkLLM pipeline, we evaluated MirrorMark+CABS under insertion, deletion, and substitution with edit ratios
> $\epsilon \in \\{0, 0.2, 0.4\\}$ (Tables 6–8). Across these attacks:
>
> - **Insertion:** Even at $\epsilon=0.4$, MirrorMark maintains high detectability (AUC = 0.999, TPR@1%FPR = 0.992) with bit accuracy reduced to 0.790, indicating that insertion mainly impairs bit recovery rather than WM/Non-WM separation.
> - **Deletion:** The most adversarial attack. At $\epsilon=0.4$, AUC remains above chance (0.939) and TPR@1%FPR reaches 0.604. This degradation arises not only from the desynchronization of the token-to-position mapping but also from reduced detectability due to the smaller number of surviving tokens.
> - **Substitution:** The least destructive attack. At $\epsilon=0.4$, MirrorMark sustains strong detectability (AUC ≈ 0.998, TPR@1%FPR ≈ 0.992) and relatively high bit accuracy ($\sim 0.75–0.78$), confirming that CABS effectively absorbs localized perturbations.

---

> > ### Author Response · Authors · 2025-11-25
> >
> > **Q: Could the authors provide a sensitivity analysis on the number of bits in comparison to the benchmark methods? Since MirrorMark encodes $M$ bits using fewer tokens, it is possible that for sufficiently large $M$, the multi-bit watermark may fail in practice.**
> >
> > We added experiments evaluating MirrorMark with different symbol sizes $m$ under a single-position setting (\$(H{=}1$\) for sequences of up to 60 tokens, as shown in Fig. 14 and Fig. 15 in Appendix F.8 of the revised manuscript, which are also [provided in this link for easy access] (https://anonymous.4open.science/api/repo/MirrorMark-abcdefg/file/additional_experiments/Fig14-15.pdf?v=58791b53).
> >
> > Fig. 14 presents the sensitivity of the Gumbel-max variant to varying $m$, and Fig. 15 shows the corresponding sensitivity for the tournament-sampling variant. In both figures, we also include the zero-bit versions of MirrorMark to illustrate how much performance degrades when moving from zero-bit to multi-bit watermarking. As expected, increasing $m$ from 0 to 4 introduces a moderate decrease in performance, reflecting the inherent cost of embedding more bits in a fixed-length sequence.
> >
> > Additionally, Fig. 15 compares the Gumbel-max variant of MirrorMark with the multi-key baseline from Algorithm 1 of [1], where each message is represented by a different key. Across all settings, MirrorMark consistently outperforms this multi-key approach, indicating that our mirroring-based design is more effective than assigning separate keys for different messages.
> >
> > [1] Three bricks to consolidate watermarks for large language models, Fernandez et al.

---

### Author Response · Authors · 2025-12-03
**Summary to the Area Chair (I)**

We sincerely thank you for your time and effort throughout the review process. To provide a clear global picture of our method, we first introduce the main components of MirrorMark. MirrorMark consists of **three core modules**:

### **1. Mod-1 Mirroring**
A mod-1 mirroring transformation encodes each *m*-bit symbol by reflecting a token’s **u-value** around a message-specific pivot, where **u-value** is generated by a pseudorandom function with the context tokens and current token as the seed. This embeds a structured multi-bit signal while preserving the underlying token distribution.

### **2. CABS**
CABS determines which symbol to embed at each generation step by mapping tokens to message positions in a balanced, context-dependent manner. It prevents error propagation across the sequence and confines the impact of local insertions or deletions to only the affected frame, greatly improving robustness under edits.

### **3. Decoding and Detection**
During decoding, CABS is replayed to recover token-to-position assignments. Each symbol is decoded from its mirrored u-values using the appropriate score function, and all decoded signals are aggregated to detect the watermark.

In summary, MirrorMark’s main technical contributions lie in **mod-1 mirroring**, which embeds a distortion-free multi-bit signal, and **CABS**, which ensures balanced token-to-position mapping and prevents error propagation across the sequence by restricting the effect of edits. The decoding stage then leverages these structured signals to reliably recover the embedded watermark.

---

> ### Author Response · Authors · 2025-12-03
> **Summary to the Area Chair (II)**
>
> Across all reviewers, we carefully addressed every concern through substantial new experiments, additional analyses, and multiple improvements in the revised manuscript. Below, we summarize our responses to each reviewer for your convenience.
>
> ---
>
> ## **Reviewer GKSn**
>
> The reviewer offered many constructive comments on presentation, evaluation, and the connection between theoretical performance and empirical performance. We fully addressed all points and provide a brief summary of the key findings below:
>
> ### **1. Clarity of Section 3 and presentation**
> - We improved the clarity of Section 3 by adding two new diagrams (Fig. 1–2 in the revised manuscript) to visually explain the workflow of mod-1 mirroring and CABS, and we refined the presentation throughout the paper.
>
> ### **2. Disentangling contributions of mirroring and CABS**
> - We conducted new experiments combining prior samplers (MPAC / RSBH) with mod-1 mirroring and pairing CABS with earlier schemes (Figs. 10–12). These results clearly isolate each component’s effect: CABS consistently improves MirrorMark’s detectability, while prior samplers yield imbalanced token allocation and weaker multi-bit performance under mod-1 mirroring. This reinforces the need for CABS to maintain positional balance.
>
> ### **3. Thresholding and FPR guarantees**
> - The original submission provided theoretical approximations for FPR for both Gumbel-max and tournament-sampling variants (Eq. 20 and Eq. 50 in Appendix C).
> - We added cross-language experiments showing how thresholds transfer and why entropy differences between languages cause expected shifts (Figs. 16–18 in Appendix F.9). While absolute thresholds shift across languages, ROC and AUC remain stable, confirming that MirrorMark’s detectability generalizes well once minimal calibration is applied.
>
> ### **4. Text-quality evaluation**
> - We added GPT-4o LLM-as-judge scores and repetition/diversity metrics (Fig. 13 and Table 10 in Appendix F.7). We also released all generated text and evaluation logs via an anonymous link (https://anonymous.4open.science/r/MirrorMark-abcdefg/text_quality_json.zip) (please click 'view raw' to access the folder) for independent inspection.
> - Results show MirrorMark’s linguistic quality remains nearly identical to non-watermarked text, while distortion-based baselines exhibit noticeable degradation.
>
> ### **5. Comparison with multi-key baseline**
> - We added results comparing MirrorMark with the multi-key baseline (Algorithm 1 of [1]) (Fig. 14 in Appendix F.8). MirrorMark consistently outperforms this baseline.
>
> ### **6. Robustness under edits**
> - We added comprehensive insertion, deletion, and substitution experiments (Tables 6–8 in Appendix F.4). MirrorMark maintains strong detectability even at high edit ratios (e.g., AUC \sim 0.998–0.999 for insertion/substitution at 40% edits), demonstrating that CABS preserves positional consistency under realistic edits.
>
> ### **7. Paraphrasing robustness and TPR reporting**
> - We expanded Table 3 to include TPR@1%FPR, symbol sizes *m*, and zero-bit baselines. Paraphrasing disrupts fine-grained token transitions, reducing bit accuracy. However, MirrorMark’s AUC remains high and consistently exceeds all baselines. The multi-bit decoder also amplifies residual watermark signals, improving detection performance over zero-bit baselines.
>
> ### **8. Theoretical interpretation and empirical verification**
> - We clarified that Theorem 4.1 models the single-position case and added new single-position experiments (Figs. 14–15 in Appendix F.8) that match the theoretical trends.
> - Although Theorem 4.1 does not aim to describe the multi-position setting, the single-position analysis captures the core statistical mechanisms that also shape multi-position behavior. For example, how the mean and variance of the sequence-level score determine the separation between hypotheses $\mathcal{H}_0$ and $\mathcal{H}_1$.
>
> ### **9. CABS parameter ablations**
> - We added full ablations on *f*, *W*, and *max_factor* (Tables 6–8 in Appendix F.4). Across insertion, deletion, and substitution attacks, our default configuration (*f*=3, *W*=4, *max_factor*=1.5) consistently provides the best or near-best robustness and detectability.
>
> ### **10. Instruction-task evaluation**
> - Following the reviewer’s suggestion, we evaluated MirrorMark on instruction tasks (Fig. 19 in Appendix F.10). MirrorMark remains detectable but shows lower TPR due to the reduced sampling entropy of instruction-tuned models, highlighting instruction following as a particularly challenging setting for watermarking.
>
> [1] Three bricks to consolidate watermarks for large language models, Fernandez et al.
>
> In the follow-up message, the reviewer confirmed that most concerns have been addressed and expressed the intention to raise the score once the system allows editing. We truly appreciate this positive update.
>
> ----

---

> > ### Author Response · Authors · 2025-12-03
> > **Summary to the Area Chair (III)**
> >
> > ## **Reviewer 9XrG**
> >
> > This reviewer focused on robustness under realistic edits, the relationship between watermarking and steganography, the interpretation of MirrorMark’s distortion-free design, the mathematical presentation, and the run-time overhead. Many of these concerns—especially robustness and the effect of edits—were already addressed comprehensively in our response to Reviewer GKSn through substantial new experiments. We summarize the key points below.
> >
> > ### **1. Robustness under realistic edits**
> > - The robustness concerns raised here overlap with those discussed by Reviewer GKSn. In the revised manuscript, we added extensive evaluations of insertion, deletion, and substitution attacks, including full CABS parameter ablations (Tables 6–8 in Appendix F.4). These experiments consistently show that CABS localizes edit impact and preserves positional structure, allowing MirrorMark to maintain high detectability even under substantial edits (e.g., AUC $\approx$ 0.998–0.999 at a 40% edit rate).
> > - Robustness to paraphrasing is also evaluated in Table 3. While bit accuracy decreases due to syntactic rewrites (a known limitation across all watermarking methods), AUC remains high, and multi-bit decoding amplifies the surviving signal. Overall, MirrorMark consistently performs best among all baselines under paraphrasing.
> >
> > ### **2. Clarifying watermarking vs. steganography**
> > - We clarified that MirrorMark is fundamentally a watermarking scheme. Although it can serve steganographic purposes due to its distribution-preserving design, mod-1 mirroring introduces structured statistical separation that enables effective multi-bit detection without degrading text quality.
> > - Reduced robustness under high-quality paraphrasing is a shared limitation of all existing methods and does not imply steganographic behavior.
> > - We explicitly identify improving robustness under semantic rewriting as an important direction for future work.
> >
> > ### **3. Reducing mathematical presentation**
> > - Following the reviewer’s suggestion, we streamlined the theoretical exposition by removing non-essential notation, introducing intuitive explanations earlier, and adding Fig. 3 to visually summarize how detectability scales with key statistical quantities.
> > - We clarified that Theorem 4.1 characterizes the single-position setting and added matching single-position experiments (Figs. 14–15), which closely follow the predicted trends, making the theory more accessible while preserving rigor.
> >
> > ### **4. On the feasibility of a “soft version” of MirrorMark**
> > - We clarified that the proposed “soft” mirroring variant—interpolating between original and mirrored \(u\)-values—would violate the measure-preserving property and introduce distributional distortion, degrading text quality.
> >
> > ### **5. Runtime overhead**
> > - We evaluated the wall-clock cost for both watermarking and detection. Watermarked generation adds only about 3 seconds for a 400-token sample (due to per-step random values generation), and detection remains extremely lightweight (about 0.30 seconds), effectively identical to non-watermarked text. This confirms that MirrorMark introduces minimal computational overhead.
> >
> > ----
> >
> > ## **Reviewer Auzr**
> >
> > This reviewer provided a positive borderline-accept rating. Most of the concerns, such as clarity of Section 3, ablation study of CABS, expanded robustness evaluation, and the sensitivity of MirrorMark with varying symbol size $m$, overlap significantly with the issues raised by Reviewers GKSn and 9XrG. All of these points were already addressed in our responses to Reviewers GKSn and 9XrG through substantial improvements in presentation, robustness analysis, and ablation studies.
> >
> > ----
> >
> > ## **Reviewer 9mcw**
> > This reviewer’s only concern was that the initial submission did not strictly follow the official ICLR template. We thank the reviewer for pointing this out. The revised manuscript now fully adheres to the official template and fits within the page limit. Although this reviewer did not evaluate the technical contributions, the revised version also incorporates the substantial experimental additions and clarifications provided in response to the other reviewers’ detailed technical feedback.
> >
> > ----

---

### Note · Program_Chairs · 2026-01-05
**Submission Desk Rejected by Program Chairs**

The original submitted PDF (reviewed by reviewers) had manipulated margins, violating ICLR formatting guidelines. Consequently the submission must be desk rejected.